# A Sensitivity Analysis of State-Space Models on Graphs

## Abstract

The recent success of State-Space Models (SSMs) in sequence modeling has inspired their extension to graphs, giving rise to Graph State-Space Models (GSSMs). While effective, existing approaches often rely on sequentializations or spectral decompositions that lack permutation equivariance, message-passing compatibility, and computational efficiency. Moreover, they typically target either static or temporal graphs in isolation and, crucially, provide only loose or qualitative results on information propagation, offering no exact guarantees on challenges such as vanishing gradients and over-squashing. In this work, we revisit the design of GSSMs through the lens of sensitivity analysis. We introduce a principled integration of modern SSM computation into the Message-Passing Neural Network framework, yielding a unified architecture that is computationally efficient, permutation equivariant, and supports fast parallelism. Our formulation admits closed-form Jacobian computations, enabling an exact sensitivity analysis of node-to-node dependencies and rigorous lower bounds on information flow, contrasting sharply with prior heuristic approaches. These theoretical insights clarify when and how stable long-range propagation can be achieved. Finally, we validate our model across a wide range of benchmarks, including node classification, graph property prediction, long-range reasoning, and spatiotemporal forecasting, where it achieves strong empirical performance while preserving the simplicity of message passing.

## 1 Introduction

Graph Neural Networks (GNNs) (Wu et al., 2020a; Gravina & Bacciu, 2024), and in particular Message-Passing Neural Networks (MPNNs) (Gilmer et al., 2017), have become a standard tool for learning from graph-structured data. Yet, traditional MPNNs such as GCNs (Kipf & Welling, 2016) struggle to propagate information across distant nodes due to phenomena like over-squashing (Alon & Yahav, 2021; Topping et al., 2022; Di Giovanni et al., 2023) and, more generally, vanishing gradients (Di Giovanni et al., 2023; Pascanu et al., 2013; Arroyo et al., 2025), which limit their effectiveness on tasks requiring long-range dependency modeling (Dwivedi et al., 2022b). While a variety of remedies have been proposed, ranging from rewiring techniques (Topping et al., 2022; Karhadkar et al., 2023; Gutteridge et al., 2023), to transformers (Kreuzer et al., 2021b; Ying & Leskovec, 2021; Rampášek et al., 2022; Dwivedi et al., 2021; 2022a), to regularization strategies in weight space (Gravina et al., 2023; 2025), these often rely on heavy architectural modifications and typically do not extend cleanly to standard MPNNs like GCNs.

In parallel, State-Space Models (SSMs) have recently emerged as a powerful paradigm in sequence modeling, with architectures such as LRU (Orvieto et al., 2023), S4 (Gu et al., 2021), and subsequent extensions (Smith et al., 2022; Gupta et al., 2022; Poli et al., 2023; Fu et al., 2022), culminating in advanced designs like Mamba (Gu & Dao, 2023), Griffin (De et al., 2024), and xLSTM (Beck et al., 2024). These models rely on linear recurrent dynamics interleaved with nonlinear projections, a design that enables efficient training, stable gradient flow, universal approximation, and robust long-range dependency modeling (Pipiras & Taqqu, 2017; Voelker et al., 2019; Orvieto et al., 2024; Muca Cirone et al., 2024). Inspired by this progress, several works have attempted to adapt SSMs to graph learning. Current approaches, however, either enforce sequentializations of the graph (Tang et al., 2023; Wang et al., 2024a; Behrouz & Hashemi, 2024) or adopt spectral decompositions (Huang et al., 2024; Zhao et al., 2024), which may compromise permutation equivariance (Bronstein et al., 2021), distort graph topology, or rely on non-unique modes (Lim et al., 2023). Moreover, while these

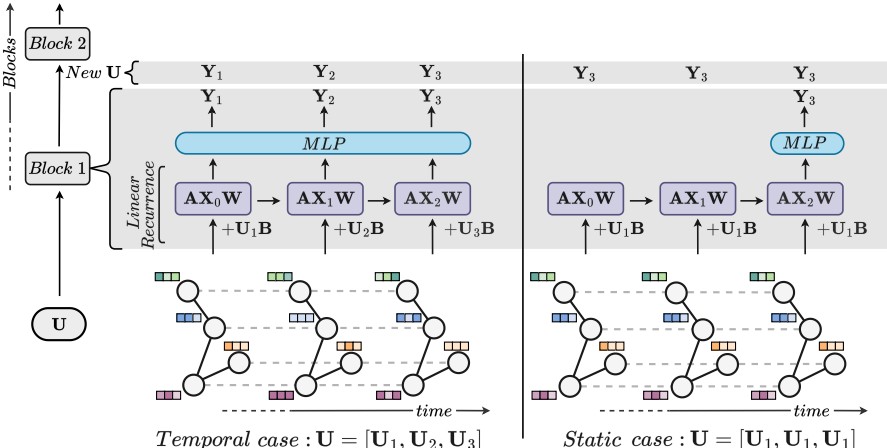

Figure 1: Illustration of our MP-SSM for temporal and static cases, considering a recurrence time $k + 1 = 3$. The temporal case (left) incorporates dynamic updates to node embeddings over time steps, represented as $\mathbf{U} = [\mathbf{U}_1, \mathbf{U}_2, \mathbf{U}_3]$, while the static case (right) uses fixed node embeddings $\mathbf{U} = [\mathbf{U}_1, \mathbf{U}_1, \mathbf{U}_1]$. An MP-SSM block comprises a linear recurrence followed by a multilayer perceptron (MLP). Multiple MP-SSM blocks are stacked to construct a deep MP-SSM architecture.

methods improve propagation in practice, they provide at best loose guarantees on sensitivity, leaving fundamental questions about stability and information flow unanswered.

This paper revisits Graph State-Space Models (GSSMs) through the lens of sensitivity analysis, i.e., by studying how the state of a node depends on information from distant nodes. We propose a principled integration of SSM computation into the MPNN framework that not only preserves permutation equivariance and computational efficiency, but also admits *exact* sensitivity analysis. This allows us to rigorously quantify node-to-node dependencies, derive precise lower bounds for vanishing gradients and over-squashing, and identify unfavorable topologies that exacerbate propagation bottlenecks. In contrast to prior work that relies on approximations or heuristic arguments, our analysis provides concrete and informative characterizations of how information flows in the deep regime.

Our contributions can be summarized as follows:

1. **Principled integration of SSMs and MPNNs through sensitivity analisys.** We introduce a simple yet general framework, namely Message-Passing State-Space Model (MP-SSM), that embeds linear state-space dynamics directly into message passing. This design unifies static and temporal graphs while preserving permutation equivariance and graph topology. It enables stable long-range information propagation and supports fast parallel implementation.

2. **Exact sensitivity analysis.** Our formulation enables closed-form Jacobian computations, yielding exact characterizations of local and global sensitivities, i.e., the model's information transfer capacity. We provide lower bounds that directly link architectural design choices to the alleviation of vanishing gradients and over-squashing.

3. **Empirical validation.** Across 15 benchmarks including synthetic and real-world long-range tasks, heterophilic node classification, and spatiotemporal forecasting, our approach consistently matches or outperforms state-of-the-art baselines, demonstrating both its versatility and effectiveness.

## 2 MESSAGE-PASSING STATE-SPACE MODEL

In this section, we present our framework, called *Message-Passing State-Space Model* (MP-SSM), which integrates state-space modeling into the message-passing paradigm. The theoretical analysis that guided the design of MP-SSM and ensures principled information propagation across the graph is detailed in Section 3.

**Graph and Shift Operator.** We represent a graph as $G = (V, E)$ where $V$ is a set of $n$ nodes, and $E \subseteq V \times V$ is a set of $m$ undirected edges. The adjacency matrix $\tilde{\mathbf{A}} \in \{0,1\}^{n \times n}$ encodes edge presence with $(\tilde{\mathbf{A}})_{ij} = 1$ if $(i,j) \in E$, and zero otherwise. To enable message passing, we use the graph shift operator (GSO) defined as the symmetrically normalized adjacency with self-loops (Kipf & Welling, 2016):

$$\mathbf{A} = \mathbf{D}^{-\frac{1}{2}}(\tilde{\mathbf{A}} + \mathbf{I})\mathbf{D}^{-\frac{1}{2}}, \tag{1}$$

where $\tilde{\mathbf{A}}$ is the adjacency matrix, and $\mathbf{D}$ is the degree matrix of $\tilde{\mathbf{A}} + \mathbf{I}$, with $(\mathbf{D}^{-\frac{1}{2}})_{ii} = \left(1 + \sum_{j=1}^{n}(\tilde{\mathbf{A}})_{ij}\right)^{-\frac{1}{2}}$. We emphasize that, although we adopt the GSO in Eq. (1) for its simplicity and widespread use, our framework is compatible with any choice of GSO.

**Linear State-Space Recurrence on Graphs.** We denote the graph data as a sequence of input node features $[\mathbf{U}_t]_{t=1}^{T}$, with $\mathbf{U}_t \in \mathbb{R}^{n \times c'}$, with $c'$ being the dimensionality of the input features. We note that, for static graphs the sequence consists of a single element, i.e., $\mathbf{U}_1$ (as shown in the bottom-right of Fig. 1). We embed the sequence of input states, obtaining a sequence of hidden states $[\mathbf{X}_t]_{t=1}^{T}$, where $\mathbf{X}_t \in \mathbb{R}^{n \times c}$, via a linear message-passing scheme and channel mixing with learnable weight matrices $\mathbf{W} \in \mathbb{R}^{c \times c}, \mathbf{B} \in \mathbb{R}^{c' \times c}$, as follows:

$$\mathbf{X}_{t+1} = \mathbf{A}\mathbf{X}_t\mathbf{W} + \mathbf{U}_{t+1}\mathbf{B}. \tag{2}$$

Eq. (2) represents the linear state-space recurrence on graphs. Note that the message-passing mechanism of many popular GNN models in the literature can be expressed through the form of of this equation, including methods like GCN (Kipf & Welling, 2016), GAT (Veličković et al., 2018), and GIN (Xu et al., 2019). *A key distinction from such models lies in the use of a purely linear recurrent equation.* This design choice is consistent with modern SSM approaches and, crucially, enables both an exact sensitivity analysis (discussed in Section 3) and an efficient parallel implementation. Specifically, MP-SSM can be parallelised by unrolling the linear recurrence and computing a closed-form solution in a single step. In Appendix B, we describe our fast implementation, discussing both its advantages and limitations, and provide a runtime comparison with a standard GCN, showing that MP-SSM can achieve up to a 1000× speedup.

**MP-SSM Block.** A block of our MP-SSM is designed to propagate information between nodes that are $k$ hops away from each other, where $k$ can also be large, as discussed in Section 3. Each block is composed of $k$ iterations of the linear recurrence described in Eq. (2), followed by a learnable graph-agnostic nonlinear mapping. Setting the initial state $\mathbf{X}_0 = \mathbf{0} \in \mathbb{R}^{n \times c}$, we define our MP-SSM block as:

$$\mathbf{X}_{t+1} = \mathbf{A}\mathbf{X}_t\mathbf{W} + \mathbf{U}_{t+1}\mathbf{B}, \qquad t = 0, \ldots, k, \tag{3}$$
$$\mathbf{Y}_{t+1} = \text{MLP}(\mathbf{X}_{t+1}), \tag{4}$$

where MLP denotes a nonlinear multilayer perceptron of 2 dense layers with a nonlinearity in between them, and $k$ a hyperparameter defining the depth of each MP-SSM block. Eqs. (3) and (4) define the state-space representation on graphs, which forms the foundation of our proposed MP-SSM. Our framework is inspired by SSMs, which are naturally suited for sequential data. In temporal graph settings, the input naturally consists of a sequence of graphs (e.g., with time-varying features). Given an input sequence $\mathbf{U} = [\mathbf{U}_1, \mathbf{U}_2, \ldots, \mathbf{U}_{k+1}]$, we apply the same MLP decoder of Eq. (4), shared across all time steps, to the corresponding embeddings $[\mathbf{X}_1, \mathbf{X}_2, \ldots, \mathbf{X}_{k+1}]$, producing an output sequence $[\mathbf{Y}_1, \mathbf{Y}_2, \ldots, \mathbf{Y}_{k+1}]$ of the same length. For static graphs, however, we must construct a sequence from a single input instance $\mathbf{U}_1$. As illustrated in Figure 1, we unify the treatment of temporal and static settings by generating a constant input sequence $\mathbf{U} = [\mathbf{U}_1, \mathbf{U}_1, \ldots, \mathbf{U}_1]$ of length $k + 1$ for the static case. We note that this design induces a skip connection in the recurrence. In the static setting, the MLP decoder is applied solely to the final embedding after $k + 1$ steps. Consequently, both the input and output sequences are constant: $[\mathbf{U}_1, \mathbf{U}_1, \ldots, \mathbf{U}_1]$ as input and $[\mathbf{Y}_{k+1}, \mathbf{Y}_{k+1}, \ldots, \mathbf{Y}_{k+1}]$ as output. Fig. 1 illustrates and summarizes the modes of operation described above. In Appendix E, we clarify the originality of our framework in relation to recent state-space modeling approaches for temporal graphs, i.e., GGRNN (Ruiz et al., 2020) and GraphSSM (Li et al., 2024), and static graphs, i.e., S4G (Song et al., 2024). A key feature that distinguishes MP-SSM from these approaches is the absence of nonlinearity in the graph diffusion dynamics. In fact, the only nonlinearity in the entire MP-SSM block resides within the MLP decoder. This property is crucial both for enabling exact sensitivity analysis and for supporting an efficient parallel implementation of the recurrence, as detailed in Appendix B.

**The Deep MP-SSM Architecture.** Following principles established in modern SSMs (Gu et al., 2021; Orvieto et al., 2023), we build a hierarchy of representations by constructing a deep MP-SSM architecture composed of stacked MP-SSM blocks. We use the output of an MP-SSM block as an input for the next one. We visually summarize this concept at the top of Figure 1. We note that, stacking multiple MP-SSM blocks allows the model to increase its effective aperture, aggregating information from further nodes. Specifically, the embedding $\mathbf{Y}_{k+1}$ encodes information aggregated up to the $k$-th hop. Therefore, stacking $s$ MP-SSM blocks, each of depth $k$, allows to aggregate information from up to $sk$ hops away. In Appendix F, we provide a multi-hop interpretation of a deep MP-SSM architecture, in the static case.

**MP-SSM generalizes its corresponding MPNN backbone.** We note that our MP-SSM can implement its backbone MPNN, an important property that allows it to retain desired or known behavior from existing MPNNs while also generalizing it and allowing for improved information transfer, as discussed in Section 3. To show that our model can implement its backbone MPNN, which in our case is based on GCN via the chosen GSO (as shown in Eq. (1)), we consider the static case, i.e., an input sequence $[\mathbf{U}_1, \ldots, \mathbf{U}_1]$, under the assumption that the MLP is a nonlinear activation $\sigma$ function. We note that this can be obtained if the weights within the MLP decoder are the identity matrices, i.e., $\mathrm{MLP}(\cdot) = \sigma(\cdot)$. Then an MP-SSM block with $k = 1$ yields a GCN layer. In fact, if $k = 1$ then Eqs. (3) and (4) read:

$$\mathbf{X}_1 = \mathbf{U}_1\mathbf{B} \quad \Rightarrow \quad \mathbf{X}_2 = \mathbf{A}\mathbf{U}_1\mathbf{B}\mathbf{W} + \mathbf{U}_1\mathbf{B} = \mathbf{A}\mathbf{X}_1\mathbf{W} + \mathbf{X}_1 \quad \Rightarrow \quad \mathbf{Y}_2 = \sigma(\mathbf{A}\mathbf{X}_1\mathbf{W} + \mathbf{X}_1),$$

which implements a GCN with a residual connection. Then $\mathbf{Y}_2$ is passed as an input to the next MP-SSM block, which yields a similar update rule, effectively constructing a deep GCN. However, we note that if $k \geq 2$, then an MP-SSM block deviates from the standard GCN processing.

**Final SSM heuristics.** If the GSO is the identity matrix ($\mathbf{A} = \mathbf{I}$), then our MP-SSM resutls in a multi-input multi-output SSM. This architecture is graph-agnostic, and it can be made resilient to vanishing and exploding gradient issues through standard deep learning heuristics such as residual connections (He et al., 2016) and normalization layers (Vaswani et al., 2017b), with dropout being employed as a regularization technique to support the learning of robust hierarchical representations (Srivastava et al., 2014). In our deep MP-SSM architecture, we apply these heuristics between MP-SSM blocks, following established practices in SSMs (Gu et al., 2021; Gu & Dao, 2023). Appendix J.1 presents an ablation study tracing the incremental impact of each SSM heuristic on graph representation learning, progressing from a plain GCN to a deep MP-SSM architecture. Finally, we discuss the complexity and runtimes of MP-SSM in Appendix I.

## 3 SENSITIVITY ANALYSIS

We conduct a sensitivity analysis of MP-SSM via the spectral norm of the Jacobian of node features, as in (Topping et al., 2022). We provide an exact characterization of MP-SSM's gradient flow through the graph, identify unfavourable topological structures that intensify oversquashing effects, and quantitatively assess the impact of removing nonlinearities at each recurrent step of graph diffusion, particularly in alleviating vanishing gradients in the deep regime.

*Remark* 3.1. As discussed in Section 2, MP-SSM extends graph-agnostic SSMs, for which established deep learning heuristics are known to effectively address vanishing/exploding gradient issues. This observation motivates our focus for sensitivity analysis on the linear recurrent equation within an MP-SSM block, as it encapsulates the core dynamics relevant to information propagation on graphs. Notably, all the other operations within a deep MP-SSM are independent of the graph structure. Thus, **if the linear recurrent equation supports effective information transfer, then this property naturally extends across the full MP-SSM architecture**, which is fundamentally an SSM-inspired stack of such linear recurrences.

Let $\mathbf{X}_s^{(j)}$ and $\mathbf{X}_t^{(i)}$ denote the embeddings of nodes $j$ and $i$ at time steps $s \leq t$. We define:

**Definition 3.2** (Local sensitivity). The *local sensitivity* of the $i$-th node features to features of the $j$-th node, after $t - s$ applications of message-passing steps, is defined as the following spectral norm:

$$\mathcal{S}_{ij}(t - s) = \left\| \frac{\partial \mathbf{X}_t^{(i)}}{\partial \mathbf{X}_s^{(j)}} \right\|. \tag{5}$$

Equation (5) measures the influence of node $j$'s features at time $s$ on node $i$ at time $t$.

*Remark* 3.3. If the local sensitivity between two nodes increases exponentially with $t - s$, then the learning dynamics of the MPNN are unstable; that is the typical case for linear MPNNs using the adjacency matrix without any normalization or feature normalization. Therefore, **upper bounds on local sensitivity are linked with stable message propagation, in the deep regime**.

The linearity of the recurrence of an MP-SSM block allows an exact computation of the Jacobian between two nodes $j, i$ at different times $s, t$, in terms of the powers of the GSO, as expressed by Equation (6) in Theorem 3.4 (for the proof, see Appendix C.2).

**Theorem 3.4** (Exact Jacobian computation in MP-SSM). *The Jacobian of the linear recurrent equation of an MP-SSM block, from node $j$ at layer $s$ to node $i$ at layer $t \geq s$, can be computed exactly, and it has the following form:*

$$\frac{\partial \mathbf{X}_t^{(i)}}{\partial \mathbf{X}_s^{(j)}} = \underbrace{(\mathbf{A}^{t-s})_{ij}}_{scalar} \underbrace{(\mathbf{W}^\top)^{t-s}}_{matrix}. \tag{6}$$

Consequently, GSOs that yield a bounded outcome under iterative multiplication promote stable MP-SSM dynamics, as highlighted in Remark 3.3. In Lemma 3.5, we formally prove (see Appendix C.1) that the GSO defined in Equation (1) exhibits this stability property, along with additional characteristics[1] that support our theoretical analysis.

**Lemma 3.5** (Powers of symmetrically normalized adjacency with self-loops). *Assume an undirected graph. The spectrum of the powers of the symmetric normalized adjacency matrix $\mathbf{A} = \mathbf{D}^{-\frac{1}{2}}(\tilde{\mathbf{A}} + \mathbf{I})\mathbf{D}^{-\frac{1}{2}}$ is contained in the interval $[-1, 1]$. The largest eigenvalue of $\mathbf{A}^t$ has absolute value of $1$ with corresponding eigenvector $\mathbf{d} = diag(\mathbf{D}^{\frac{1}{2}})$, for all $t \geq 1$. In particular, the sequence of powers $[\mathbf{A}^t]_{t \geq 1}$ does not diverge or converge to the null matrix.*

Thus, Lemma 3.5 implies that the symmetrically normalized adjacency with self-loops, i.e., Equation (1), serves as a GSO that ensures stable dynamics when performing a large number of message-passing operations in the MP-SSM's framework. Moreover, for such a particular GSO, we can derive a precise approximation of the local sensitivity in the deep regime, as stated in Theorem 3.6 and proved in Appendix C.3.

**Theorem 3.6** (Approximation deep regime). *Assume a connected graph, and the GSO defined in Equation (1). Then, for large values of $t - s$, the Jacobian of the linear recurrent equation of an MP-SSM block, from node $j$ at layer $s$ to node $i$ at layer $t \geq s$, admits the following approximation:*

$$\frac{\partial \mathbf{X}_t^{(i)}}{\partial \mathbf{X}_s^{(j)}} \approx \frac{\sqrt{(1 + d_i)(1 + d_j)}}{|V| + 2|E|} (\mathbf{W}^\top)^{t-s}, \tag{7}$$

*where $d_l = \sum_{j=1}^n (\tilde{\mathbf{A}})_{lj}$ is the degree of the $l$-th node.*

For the case of the GSO of Equation (1), we can find a precise lower bound for the minimum local sensitivity among all possible pairs of nodes in the graph, in the deep regime (proof in Appendix C.4).

**Corollary 3.7** (Lower bound minimum sensitivity). *Assume a connected graph, and the GSO of Equation (1). Then, for large values of $t - s$, the following lower bound for the minimum local sensitivity of the linear recurrent equation of an MP-SSM block holds:*

$$\frac{2}{|V| + 2|E|} ||\mathbf{W}^{t-s}|| \leq \min_{i,j} \mathcal{S}_{ij}(t - s). \tag{8}$$

The minimum local sensitivity is realized for pairs of nodes among which the transfer of information is the most critical due to the structure of the graph. Therefore, **lower bounds on the minimum local sensitivity are linked to the alleviation of over-squashing**. Rewiring techniques are known to help combating this phenomenon (Di Giovanni et al., 2023). Corollary 3.7 proves that, without rewiring, MP-SSM can deal with over-squashing by increasing the norm of the recurrent weight matrix. In Remark 3.8, we construct an example of a topology that approaches the lower bound of Eq. (8), thus realising a worst case scenario due to over-squashing.

---

[1]Similar characteristics of the GSO in Equation (1) have also been discussed in (Oono & Suzuki, 2019).

*Remark* 3.8 (Bottleneck Topologies). A chain of $m$ cliques of order $d$ represents a topology realising a bad scenario for Eq. (7), since local sensitivity can reach values as low as $\frac{1}{md^2}$, scaling on long chains and large cliques, see Appendix C.3.1 for details. This effect is intrinsically tied to the specific topology of the graph, and it aligns with prior studies that emphasize the challenges of learning on graphs with bottleneck structures (Topping et al., 2022).

To assess the overall gradient information flow across the entire graph in the deep regime, we define:

**Definition 3.9** (Global sensitivity). The *global sensitivity* of node features of the overall graph after $t - s$ hops of message aggregation is defined as:

$$\mathcal{S}(t - s) = \max_{i,j} \mathcal{S}_{ij}(t - s). \tag{9}$$

*Remark* 3.10. The local sensitivity between two far-apart nodes can be physiologically small due to the particular topology of the graph (e.g. bottlenecks), or it can be even 0 if two nodes are not connected by any walk. However, if the local sensitivity converges to 0, in the deep regime of large $t - s$, for all the pairs of nodes, i.e., if the global sensitivity converges to 0 regardless of the particular topology of the graph, then it means that the MPNN model is characterized by a vanishing information flow. Therefore, **lower bounds on global sensitivity are linked to the alleviation of vanishing gradient issues, in the deep regime**.

For connected graphs, we can leverage the exact Jacobian computation of Theorem 3.4 to prove the following lower bound on the global sensitivity, see Appendix C.5 for the proof.

**Theorem 3.11** (Lower bound global sensitivity). *Assume a connected graph. The global sensitivity of the linear recurrent equation of an MP-SSM block is lower bounded as follows:*

$$\frac{\rho(\mathbf{A})^{t-s}}{|V|} ||\mathbf{W}^{t-s}|| \leq \mathcal{S}(t - s), \tag{10}$$

*where $\rho(\mathbf{A})$ is the spectral radius of the GSO. Thus, for the GSO of Eq. (1), it holds the lower bound* $\frac{1}{|V|}||\mathbf{W}^{t-s}|| \leq \mathcal{S}(t - s)$.

This theoretical result demonstrates that MP-SSM ensures values of the global sensitivity strictly greater than zero, for any depth $t - s$ and for connected graphs with any number of nodes. This result cannot be guaranteed in a standard MPNN, as the nonlinearity applied at each time step increasingly contributes to vanish information as the depth increases, with more discussions in Appendix D.

*Remark* 3.12. Note that both results of Eqs. (6) and (10) hold for any GSO. However, for the particular case of the symmetrically normalized adjacency with self-loops, we can provide more precise approximations and bounds.

From Section 2, we know that MP-SSM generalizes its backbone MPNNs, and the GCN architecture in particular when using Eq. (1) as GSO. In Theorem 3.13, we provide an estimation of the vanishing effect caused by the application at each time step of a ReLU nonlinearity in a standard GCN compared with our MP-SSM, in the deep regime, as we prove in Appendix C.6.

**Theorem 3.13** (GCN vanishes more than MP-SSM). *Let us consider a GCN that aggregates information from $k$ hops away, i.e., with $k$ layers, equipped with the ReLU activation function. Then, the GCN vanishes information at a $2^{-\frac{k}{2}}$ faster rate than our MP-SSM block with $k$ linear recurrent steps.*

## 4 EXPERIMENTS

We evaluate MP-SSM on standard benchmarks for both static and temporal graphs. For static graphs, we assess long-range propagation via synthetic shortest-path tasks (Section 4.1) and heterophilic node classification (Section 4.2). For temporal graphs, we test spatio-temporal forecasting performance (Section 4.3). Additionally, we benchmark MP-SSM on long-range real-world benchmarks in Appendix H and on a temporal graph benchmark with dynamic topology in Appendix L. MP-SSM is compared against state-of-the-art MPNNs, multi-hop GNNs, graph transformers, graph SSMs, and spatio-temporal models (details in Appendix G.1). The closest baselines are MPNNs and graph SSMs. Datasets statistics and hyperparameter settings are described in Appendices G.2 and G.3,

respectively. Code will be released upon acceptance. We emphasize that, unlike most state-of-the-art graph models, MP-SSM runs at a speed comparable to that of a standard GCN (see runtime and complexity analyses in Appendix I), even without leveraging the optimized implementation discussed in Appendix B.

## 4.1 GRAPH PROPERTY PREDICTION

**Setup.** We evaluate MP-SSM on three synthetic tasks from Gravina et al. (2023): predicting graph diameter, single-source shortest paths (SSSP), and node eccentricity. These tasks require long-range information flow, making them suitable benchmarks for evaluating propagation depth. We follow the original setup, data, and hyperparameters.

**Results.** Table 1 reports results using $log_{10}$(MSE). MP-SSM outperforms all baselines, with an average gain of 2.4 points. On the eccentricity task, it improves over A-DGN by 3.4 points, despite A-DGN being tailored for long-range tasks, and exceeds GCN (its backbone model) by over 4 points on average, despite both using the same GSO. This highlights MP-SSM's superior ability to propagate information across distant nodes. An additional ablation on multiple GSOs is reported in Appendix J.2.

Table 1: Mean test set $log_{10}$(MSE)($\downarrow$) and std averaged on 4 random weight initializations on Graph Property Prediction tasks. The lower, the better. **First**, **second**, and **third** best results for each task are color-coded.

| Model | Diameter | SSSP | Eccentricity |
|---|---|---|---|
| **MPNNs** | | | |
| A-DGN | $-0.5188_{\pm 0.1812}$ | $-3.2417_{\pm 0.0751}$ | $0.4296_{\pm 0.1003}$ |
| DGC | $0.6028_{\pm 0.0050}$ | $-0.1483_{\pm 0.0231}$ | $0.8261_{\pm 0.0032}$ |
| GAT | $0.8221_{\pm 0.0752}$ | $0.6951_{\pm 0.1499}$ | $0.7909_{\pm 0.0222}$ |
| GCN | $0.7424_{\pm 0.0466}$ | $0.9499_{\pm 0.0001}$ | $0.8468_{\pm 0.0028}$ |
| GCNII | $0.5287_{\pm 0.0570}$ | $-1.1329_{\pm 0.0135}$ | $0.7640_{\pm 0.0355}$ |
| GIN | $0.6131_{\pm 0.0990}$ | $-0.5408_{\pm 0.4193}$ | $0.9504_{\pm 0.0007}$ |
| GRAND | $0.6715_{\pm 0.0490}$ | $-0.0942_{\pm 0.3897}$ | $0.6602_{\pm 0.1393}$ |
| GraphCON | $0.0964_{\pm 0.0620}$ | $-1.3836_{\pm 0.0092}$ | $0.6833_{\pm 0.0074}$ |
| GraphSAGE | $0.8645_{\pm 0.0401}$ | $0.2863_{\pm 0.1843}$ | $0.7863_{\pm 0.0207}$ |
| **Transformers** | | | |
| GPS | $-0.5121_{\pm 0.0426}$ | $-3.5990_{\pm 0.1949}$ | $0.6077_{\pm 0.0282}$ |
| **Ours** | | | |
| MP-SSM | $-3.2353_{\pm 0.1735}$ | $-4.6321_{\pm 0.0779}$ | $-2.9724_{\pm 0.0271}$ |

## 4.2 HETEROPHILIC BENCHMARK

**Setup.** We evaluate MP-SSM on five heterophilic benchmarks from Platonov et al. (2023): Roman-empire, Amazon-ratings, Minesweeper, Tolokers, and Questions. These tasks test the model's ability to capture complex interactions between dissimilar nodes. We follow the original data and experimental settings.

**Results.** Table 2 shows that MP-SSM consistently performs well, achieving the highest average ranking across all tasks (see Appendix K). It improves GCN by up to 17% and surpasses transformer- and SSM-based GNNs, including methods tailored for heterophilic graphs, demonstrating strong adaptability to complex, non-homophilic structures.

Table 2: Mean test set score and std averaged over 4 random weight initializations on heterophilic tasks. **First**, **second**, and **third** best results.

| Model | Roman-empire Acc $\uparrow$ | Amazon-ratings Acc $\uparrow$ | Minesweep. AUC $\uparrow$ | Tolokers AUC $\uparrow$ | Questions AUC $\uparrow$ |
|---|---|---|---|---|---|
| **MPNNs** | | | | | |
| CO-GNN | $91.57_{\pm 0.32}$ | $54.17_{\pm 0.37}$ | $97.31_{\pm 0.41}$ | $84.45_{\pm 1.17}$ | $80.02_{\pm 0.86}$ |
| GAT | $80.87_{\pm 0.30}$ | $49.09_{\pm 0.63}$ | $92.01_{\pm 0.68}$ | $83.70_{\pm 0.47}$ | $77.43_{\pm 1.20}$ |
| Gated-GCN | $74.46_{\pm 0.54}$ | $43.00_{\pm 0.32}$ | $87.54_{\pm 1.22}$ | $77.31_{\pm 1.14}$ | $76.61_{\pm 1.13}$ |
| GCN | $73.69_{\pm 0.74}$ | $48.70_{\pm 0.63}$ | $89.75_{\pm 0.52}$ | $83.64_{\pm 0.67}$ | $76.09_{\pm 1.27}$ |
| SAGE | $85.74_{\pm 0.67}$ | $53.63_{\pm 0.39}$ | $93.51_{\pm 0.57}$ | $82.43_{\pm 0.44}$ | $76.44_{\pm 0.62}$ |
| **Graph Transformers** | | | | | |
| Exphormer | $89.03_{\pm 0.37}$ | $53.51_{\pm 0.46}$ | $90.74_{\pm 0.53}$ | $83.77_{\pm 0.78}$ | $73.94_{\pm 1.06}$ |
| GOAT | $71.59_{\pm 1.25}$ | $44.61_{\pm 0.50}$ | $81.09_{\pm 1.02}$ | $83.11_{\pm 1.04}$ | $75.76_{\pm 1.66}$ |
| GPS | $82.00_{\pm 0.61}$ | $53.10_{\pm 0.42}$ | $90.63_{\pm 0.67}$ | $83.71_{\pm 0.48}$ | $71.73_{\pm 1.47}$ |
| GT | $86.51_{\pm 0.73}$ | $51.17_{\pm 0.66}$ | $91.85_{\pm 0.76}$ | $83.23_{\pm 0.64}$ | $77.95_{\pm 0.68}$ |
| **Heterophily-Designated GNNs** | | | | | |
| FAGCN | $65.22_{\pm 0.56}$ | $44.12_{\pm 0.30}$ | $88.17_{\pm 0.73}$ | $77.75_{\pm 1.05}$ | $77.24_{\pm 1.26}$ |
| FSGNN | $79.92_{\pm 0.56}$ | $52.74_{\pm 0.83}$ | $90.08_{\pm 0.70}$ | $82.76_{\pm 0.61}$ | $78.86_{\pm 0.92}$ |
| GBK-GNN | $74.57_{\pm 0.47}$ | $45.98_{\pm 0.71}$ | $90.85_{\pm 0.58}$ | $81.01_{\pm 0.67}$ | $74.47_{\pm 0.86}$ |
| GPR-GNN | $64.85_{\pm 0.27}$ | $44.88_{\pm 0.34}$ | $86.24_{\pm 0.61}$ | $72.94_{\pm 0.97}$ | $55.48_{\pm 0.91}$ |
| H2GCN | $60.11_{\pm 0.52}$ | $36.47_{\pm 0.23}$ | $89.71_{\pm 0.31}$ | $73.35_{\pm 1.01}$ | $63.59_{\pm 1.46}$ |
| **Graph SSMs** | | | | | |
| GMN | $87.69_{\pm 0.50}$ | $54.07_{\pm 0.31}$ | $91.01_{\pm 0.23}$ | $84.52_{\pm 0.21}$ | $-$ |
| GPS+Mamba | $83.10_{\pm 0.28}$ | $45.13_{\pm 0.97}$ | $89.93_{\pm 0.54}$ | $83.70_{\pm 1.05}$ | $-$ |
| **Ours** | | | | | |
| MP-SSM | $90.91_{\pm 0.48}$ | $53.65_{\pm 0.71}$ | $95.33_{\pm 0.72}$ | $85.26_{\pm 0.93}$ | $78.18_{\pm 1.34}$ |

### 4.3 SPATIO-TEMPORAL FORECASTING

**Setup.** We evaluate MP-SSM on five forecasting datasets: Metr-LA, PeMS-Bay (Li et al., 2018), Chickenpox Hungary, PedalMe London, and Wikipedia math (Rozemberczki et al., 2021). The goal is to predict future node values from time-series data. The first two focus on traffic, while the others involve public health, delivery demand, and web activity. We follow the original settings for each dataset.

**Results.** MP-SSM outperforms existing temporal GNNs across all datasets (Tables 3 and 4), highlighting its ability to model both spatial and temporal dependencies. These results confirm MP-SSM's versatility across static and temporal graph domains. Notably, MP-SSM significantly outperforms GGRNN (Ruiz et al., 2020) and GraphSSM (Li et al., 2024), see Table 4, two related state-space approaches for temporal graphs, thus highlighting the originality and effectiveness of our approach (see Appendix E for an extended discussion).

Table 3: Average MSE and standard deviation (↓) of 10 experimental repetitions. Baseline results are reported from Rozemberczki et al. (2021); Errica et al. (2023); Eliasof et al. (2024) . **First**, **second**, and **third** best methods for each task are color-coded.

| Model | Chickenpox Hungary | PedalMe London | Wikipedia Math |
|---|---|---|---|
| **Temporal GNNs** | | | |
| A3T-GCN | $1.114_{\pm 0.008}$ | $1.469_{\pm 0.027}$ | $0.781_{\pm 0.011}$ |
| AGCRN | $1.120_{\pm 0.010}$ | $1.469_{\pm 0.030}$ | $0.788_{\pm 0.011}$ |
| CDE | $\mathbf{0.848}_{\pm 0.020}$ | $\mathbf{0.810}_{\pm 0.063}$ | $0.694_{\pm 0.028}$ |
| DCRNN | $1.124_{\pm 0.015}$ | $1.463_{\pm 0.019}$ | $0.679_{\pm 0.020}$ |
| DyGrAE | $1.120_{\pm 0.021}$ | $1.455_{\pm 0.031}$ | $0.773_{\pm 0.009}$ |
| DynGESN | $0.907_{\pm 0.007}$ | $1.528_{\pm 0.063}$ | $0.610_{\pm 0.003}$ |
| EGCN-O | $1.124_{\pm 0.009}$ | $1.491_{\pm 0.024}$ | $0.750_{\pm 0.014}$ |
| GConvGRU | $1.128_{\pm 0.011}$ | $1.622_{\pm 0.032}$ | $0.657_{\pm 0.015}$ |
| GC-LSTM | $1.115_{\pm 0.014}$ | $1.455_{\pm 0.023}$ | $0.779_{\pm 0.023}$ |
| GRAND | $1.068_{\pm 0.021}$ | $1.557_{\pm 0.049}$ | $0.798_{\pm 0.034}$ |
| GREAD | $0.983_{\pm 0.027}$ | $1.291_{\pm 0.055}$ | $0.704_{\pm 0.016}$ |
| HMM4G | $0.939_{\pm 0.013}$ | $1.769_{\pm 0.370}$ | $0.542_{\pm 0.008}$ |
| MPNN LSTM | $1.116_{\pm 0.023}$ | $1.485_{\pm 0.028}$ | $0.795_{\pm 0.010}$ |
| TDE-GNN | $0.787_{\pm 0.018}$ | $0.714_{\pm 0.051}$ | $\mathbf{0.565}_{\pm 0.017}$ |
| T-GCN | $1.117_{\pm 0.011}$ | $1.479_{\pm 0.012}$ | $0.764_{\pm 0.011}$ |
| **Ours** | | | |
| MP-SSM | $0.748_{\pm 0.011}$ | $0.647_{\pm 0.062}$ | $0.509_{\pm 0.008}$ |

## 5 RELATED WORKS

**Learning Long-Range Dependencies on Graphs.** While GNNs effectively model local structures via message passing, they struggle with long-range dependencies due to over-squashing and vanishing gradients (Alon & Yahav, 2021; Di Giovanni et al., 2023; Arroyo et al., 2025). Standard models like GCN (Kipf & Welling, 2016), GraphSAGE (Hamilton et al., 2017), and GIN (Xu et al., 2019) suffer from degraded performance on tasks requiring global context (Baek et al., 2021; Dwivedi et al., 2022b), especially in heterophilic graphs (Luan et al., 2024; Wang et al., 2024b). Solutions include graph rewiring (Topping et al., 2022; Karhadkar et al., 2023), weight-space regularization (Gravina et al., 2023; 2025), and physics-inspired dynamics (Heilig et al., 2025). Graph Transformers (GTs) like SAN (Kreuzer et al., 2021b), Graphormer (Ying & Leskovec, 2021), and GPS (Rampášek et al., 2022) enhance expressivity using structural encodings (Dwivedi et al., 2021; 2022a), but suffer from quadratic complexity. Scalable alternatives include sparse and linearized attention mechanisms (Zaheer et al., 2020; Choromanski et al., 2020; Shirzad et al., 2023; 2024; Wu et al., 2023; Deng et al., 2024), though simple MPNNs often remain competitive (Tönshoff et al., 2023).

**Learning Spatio-Temporal Interactions on Graphs.** Temporal GNNs often combine GNNs with RNNs to model spatio-temporal dynamics (Gravina & Bacciu, 2024). Some adopt stacked architectures that separate spatial and temporal processing (Seo et al., 2018; Pareja et al., 2020; Panagopoulos et al., 2021; Bai et al., 2021; Cini et al., 2023a), while others integrate GNNs within RNNs for joint modeling (Li et al., 2018; 2019; Chen et al., 2022; Cini et al., 2023b; Ruiz et al., 2020). Our approach follows the latter, but goes further by embedding modern SSM principles directly into the GNN architecture, unifying spatial and temporal reasoning through linear recurrence. This contrasts with GGRNN (Ruiz et al., 2020), which employs a more elaborate message-passing scheme involving nonlinear aggregation over multiple powers of the graph shift operator at each recurrent step.

**Casting State-Space Models into Graph Learning.** Several recent models adopt SSMs for static graphs by imposing sequential orderings, e.g., via degree-based sorting (Wang et al., 2024a) or

Table 4: Multivariate time series forecasting on the Metr-LA and PeMS-Bay datasets for Horizon 12. First, second, and **third** best results for each task are color-coded. Baseline results are reported from (Shao et al., 2022; Liu et al., 2023; Gao et al., 2024; Fan et al., 2024; Zhang et al., 2024).

| Model | Metr-LA | | | PeMS-Bay | | |
|---|---|---|---|---|---|---|
| | MAE ↓ | RMSE ↓ | MAPE ↓ | MAE ↓ | RMSE ↓ | MAPE ↓ |
| **Graph Agnostic** | | | | | | |
| HA | 6.99 | 13.89 | 17.54% | 3.31 | 7.54 | 7.65% |
| FC-LSTM | 4.37 | 8.69 | 14.00% | 2.37 | 4.96 | 5.70% |
| SVR | 6.72 | 13.76 | 16.70% | 3.28 | 7.08 | 8.00% |
| VAR | 6.52 | 10.11 | 15.80% | 2.93 | 5.44 | 6.50% |
| **Temporal GNNs** | | | | | | |
| AdpSTGCN | 3.40 | 7.21 | **9.45%** | 1.92 | 4.49 | 4.62% |
| ASTGCN | 6.51 | 12.52 | 11.64% | 2.61 | 5.42 | 6.00% |
| DCRNN | 3.60 | 7.60 | 10.50% | 2.07 | 4.74 | 4.90% |
| GMAN | 3.44 | 7.35 | 10.07% | 1.86 | **4.32** | 4.37% |
| Graph WaveNet | 3.53 | 7.37 | 10.01% | 1.95 | 4.52 | 4.63% |
| GTS | 3.46 | 7.31 | 9.98% | 1.95 | 4.43 | 4.58% |
| MTGNN | 3.49 | 7.23 | 9.87% | 1.94 | 4.49 | 4.53% |
| RGDAN | **3.26** | **7.02** | 9.73% | 1.82 | **4.20** | 4.28% |
| STAEformer | **3.34** | **7.02** | 9.70% | 1.88 | 4.34 | 4.41% |
| STD-MAE | 3.40 | 7.07 | **9.59%** | **1.77** | **4.20** | **4.17%** |
| STEP | 3.37 | **6.99** | 9.61% | **1.79** | **4.20** | **4.18%** |
| STGCN | 4.59 | 9.40 | 12.70% | 2.49 | 5.69 | 5.79% |
| STSGCN | 5.06 | 11.66 | 12.91% | 2.26 | 5.21 | 5.40% |
| **Temporal Graph SSMs** | | | | | | |
| GGRNN | 3.88 | 8.14 | 10.59% | 2.34 | 5.14 | 5.21% |
| GraphSSM-S4 | 3.74 | 7.90 | 10.37% | 1.98 | 4.45 | 4.77% |
| **Ours** | | | | | | |
| MP-SSM | **3.17** | **6.86** | **9.21%** | **1.62** | **4.22** | **4.05%** |

random walks (Behrouz & Hashemi, 2024), often sacrificing permutation-equivariance. Spectral methods (Huang et al., 2024) offer alternatives but are computationally demanding and prone to over-squashing (Di Giovanni et al., 2023). In the temporal graph setting, GraphSSM (Li et al., 2024) applies the diffusive dynamics of a GNN backbone first, followed by an SSM as a post-processing module. In contrast, our approach embeds the core principles of modern SSMs directly into the graph learning process, yielding a unified framework designed through the lens of sensitivity analysis that seamlessly supports both static and temporal graph modeling, while maintaining permutation equivariance, computational efficiency, and supporting parallel implementation.

# 6 CONCLUSIONS

In this work, we revisited Graph State-Space Models (GSSMs) through the lens of sensitivity analysis. While prior GSSM approaches have demonstrated empirical improvements, they typically rely on techniques that compromise core graph properties and offer only loose theoretical guarantees on information flow.

We propose a general framework called Message-Passing State-Space Model (MP-SSM), whose formulation preserves permutation equivariance, unifies static and temporal graphs, allows for fast implementation and crucially enables *exact* sensitivity analysis. This allows us to rigorously characterize node-to-node dependencies, derive precise lower bounds on vanishing gradients and over-squashing, and identify structural conditions under which information propagation is guaranteed to remain stable.

In addition to these theoretical contributions, our framework remains empirically competitive, achieving strong results across long-range, heterophilic, and spatiotemporal forecasting tasks. We believe this perspective positions sensitivity analysis as a principled foundation for the design of future graph state-space models.

ETHICS STATEMENT

The research conducted in this paper conforms in every aspect with the ICLR Code of Ethics. Our study does not involve human subjects, sensitive personal data, or applications with foreseeable harmful consequences. All experiments were conducted on publicly available datasets, and no ethical concerns are anticipated regarding data usage, methodology, or findings.

REPRODUCIBILITY STATEMENT

We provide all necessary details to implement our MP-SSM in Section 2 and Appendix B, and describe the setup of each experiment in Section 4 and Appendix G, thereby ensuring sufficient information to reproduce our results. Furthermore, all experiments are conducted on open-source datasets available online. We pledge to openly release the full codebase upon acceptance.

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

## A  LLMs Usage

Large Language Models (LLMs) were used as general-purpose assistive tools to improve the writing quality of this paper. Specifically, we used LLMs to help with grammar correction, rephrasing for clarity, and suggesting some improvements to the overall structure of the text. All LLM-generated text was carefully reviewed and edited by the authors to ensure that it accurately reflects the authors' intentions and scientific content. No LLMs were used to generate scientific content, including but not limited to research direction, hypothesis formulation, experimental design, data analysis, or interpretation of results.

## B  Fast Parallel Implementation

We describe all the details to derive and implement a fast parallel implementation for the computation of an MP-SSM block.

The unfolded recurrence of an MP-SSM block gives the following closed-form solution:

$$\mathbf{X}_{k+1} = \mathbf{A}^k \mathbf{U}_1 \mathbf{B} \mathbf{W}^k + \mathbf{A}^{k-1} \mathbf{U}_2 \mathbf{B} \mathbf{W}^{k-1} + \ldots + \mathbf{A} \mathbf{U}_k \mathbf{B} \mathbf{W} + \mathbf{U}_{k+1} \mathbf{B}. \qquad (11)$$

Therefore the equation of an MP-SSM block reads:

$$\mathbf{X}_{k+1} = \sum_{i=0}^{k} \mathbf{A}^i \mathbf{U}_{k+1-i} \mathbf{B} \mathbf{W}^i, \qquad (12)$$

$$\mathbf{Y}_{k+1} = \text{MLP}(\mathbf{X}_{k+1}), \qquad (13)$$

The closed-form solution of an MP-SSM block tells us that we could implement the whole recurrence in one shot. However, the computation of the powers of both the GSO, $\mathbf{A}$, and the recurrent weights, $\mathbf{W}$, can be extremely expensive for generic matrices and large values of $k$. On the other hand, the powers of diagonal matrices are fairly easy to compute, since they are simply the powers of their diagonal entries. Below, we show how to reduce a generic dense real-valued MP-SSM block to an equivalent diagonalised complex-valued MP-SSM block.

Assume the following diagonalisation of the shift operator: $\mathbf{A} = \mathbf{P} \mathbf{\Lambda} \mathbf{P}^{-1}$. If undirected graph, i.e., $\mathbf{A}$ is symmetric, then by spectral theorem the $\mathbf{P}$ is a real orthogonal matrix (i.e. $\mathbf{P}^{-1} = \mathbf{P}^{\top}$) and $\mathbf{\Lambda}$ is real.

Assume the following diagonalisation of the weights: $\mathbf{W} = \mathbf{V} \mathbf{\Sigma} \mathbf{V}^{-1}$. If using dense real matrices as weights, then their diagonalisation is possible only assuming complex matrices of eigenvectors $\mathbf{V}$ and complex eigenvalues $\mathbf{\Sigma}$. Also, note that the set of defective matrices (i.e. non-diagonalizable in $\mathbb{C}$) has zero Lebesgue measure (Golub & Van Loan, 2013).

Assume the following MLP equations with 2 layers: $\text{MLP}(\mathbf{X}) = \phi(\mathbf{X} \mathbf{W}_1) \mathbf{W}_2$, where $\phi$ is a nonlinearity, and $\mathbf{W}_1, \mathbf{W}_2$ real dense matrices.

With the above assumptions, the MP-SSM block equations can be equivalently written as:

$$\mathbf{X}_{k+1} = \sum_{i=0}^{k} \mathbf{P}\mathbf{\Lambda}^i \mathbf{P}^{-1}\mathbf{U}_{k+1-i}\mathbf{B}\mathbf{V}\mathbf{\Sigma}^i\mathbf{V}^{-1}, \tag{14}$$

$$\mathbf{Y}_{k+1} = \phi(\mathbf{X}_{k+1}\mathbf{W}_1)\mathbf{W}_2, \tag{15}$$

which we can write as:

$$\mathbf{X}_{k+1} = \mathbf{P}\left(\sum_{i=0}^{k} \mathbf{\Lambda}^i \mathbf{P}^{-1}\mathbf{U}_{k+1-i}\mathbf{B}\mathbf{V}\mathbf{\Sigma}^i\right)\mathbf{V}^{-1}, \tag{16}$$

$$\mathbf{Y}_{k+1} = \phi(\mathbf{X}_{k+1}\mathbf{W}_1)\mathbf{W}_2, \tag{17}$$

Multiply on the left side both terms by $\mathbf{P}^{-1}$ and on the right side both terms by $\mathbf{V}$

$$\mathbf{P}^{-1}\mathbf{X}_{k+1}\mathbf{V} = \sum_{i=0}^{k} \mathbf{\Lambda}^i \mathbf{P}^{-1}\mathbf{U}_{k+1-i}\mathbf{B}\mathbf{V}\mathbf{\Sigma}^i \tag{18}$$

If we change coordinate reference to $\mathbf{Z}_{k+1} = \mathbf{P}^{-1}\mathbf{X}_{k+1}\mathbf{V}$, then we can write:

$$\mathbf{Z}_{k+1} = \sum_{i=0}^{k} \mathbf{\Lambda}^i \mathbf{P}^{-1}\mathbf{U}_{k+1-i}\mathbf{B}\mathbf{V}\mathbf{\Sigma}^i, \tag{19}$$

$$\mathbf{Y}_{k+1} = \phi(\mathbf{P}\mathbf{Z}_{k+1}\mathbf{V}^{-1}\mathbf{W}_1)\mathbf{W}_2, \tag{20}$$

Equations (19) and (20) give the same exact dynamics of the equations (12) and (13).

The matrix of complex eigenvectors $\mathbf{V}$ in (19) can be merged into the real matrix of weights $\mathbf{B}$ in equation (21). Therefore, we can call $\hat{\mathbf{B}}$ a complex matrix of weights that accounts for the term $\mathbf{B}\mathbf{V}$. Similarly, the matrix eigenvectors $\mathbf{V}^{-1}$ in (20) can be merged into the matrix of weights $\mathbf{W}_1$ in equation (22), that we call $\hat{\mathbf{W}}_1$. To get an exact equivalence, we should exactly multiply by $\mathbf{V}$ and $\mathbf{V}^{-1}$, but merging these into learnable complex-valued matrices $\hat{\mathbf{B}}$ and $\hat{\mathbf{W}}_1$ then we get similar performance.

With these new notations, we can write the equivalent diagonalised complex-valued MP-SSM block:

$$\mathbf{Z}_{k+1} = \sum_{i=0}^{k} \mathbf{\Lambda}^i \hat{\mathbf{U}}_{k+1-i}\hat{\mathbf{B}}\mathbf{\Sigma}^i, \tag{21}$$

$$\mathbf{Y}_{k+1} = \phi(\mathbf{P}\mathbf{Z}_{k+1}\hat{\mathbf{W}}_1)\mathbf{W}_2, \tag{22}$$

where, in summary:

- input is pre-processed as $\hat{\mathbf{U}}_{k+1-i} = \mathbf{P}^{-1}\mathbf{U}_{k+1-i}$,
- $\mathbf{\Lambda}$ is the diagonal matrix of the eigenvalues of the GSO,
- learnable recurrent weights are $\hat{\mathbf{B}}$ (complex and dense), and $\mathbf{\Sigma}$ (complex and diagonal)
- learnable readout weights are $\hat{\mathbf{W}}_1$ (complex and dense), and $\mathbf{W}_2$ (real and dense)

Equations (21)-(22) tell us that we can implement the whole recurrence efficiently in a closed-form solution that only involves powers of diagonal matrices.

We emphasize that the feasibility of a graph-native, fast, and parallel implementation stems naturally as a byproduct of the MP-SSM's design choices, which unify graph diffusion and SSM-like linear recurrence within a single update equation. This differs from previous GraphSSM models that treat diffusion-based GNNs and SSMs as separate components, or that first linearize the graph into sequences prior to applying SSM layers, as detailed in the Introduction (Section 1) and Related Work (Section 5).

We provide in Algorithm 1, the pytorch-like implementation of the fast MP-SSM, provided the input sequence $(\hat{\mathbf{U}}_1, \ldots, \hat{\mathbf{U}}_{k+1})$, computes in parallel the whole output sequence $(\mathbf{Y}_1, \ldots, \mathbf{Y}_{k+1})$. We

---

**Algorithm 1** MP-SSM fast implementation

---

**Require:** the input features $x \in \mathbb{C}^{\text{num\_steps} \times n \times C}$ (if temporal), else $x \in \mathbb{C}^{n \times C}$; the number of iterations (i.e., k+1) `num_steps`; the diagonal complex-valued weight matrix $W \in \mathbb{C}^{\text{hidden\_dim}}$; the complex-valued matrix $B \in \mathbb{C}^{C \times \text{hidden\_dim}}$; the eigenvalues of the GSO `eigenvals` $\in \mathbb{C}^n$

**Ensure:** `out` $\in \mathbb{C}^{\text{num\_steps} \times n \times \text{hidden\_dim}}$

1: $\text{powers} = \text{torch.arange}(\texttt{num\_steps})$

2: $\Lambda_{\text{powers}} = \texttt{eigenvals.unsqueeze}(-1).\text{pow}(\text{powers})$ ▷ shape: $(n, \text{num\_steps})$

3: $\Sigma_{\text{powers}} = \texttt{W.unsqueeze}(-1).\text{pow}(\text{powers})$ ▷ shape: $(\text{hidden\_dim}, \text{num\_steps})$

4: **if not** temporal **then**

5: $\quad x = x.\text{repeat}(\texttt{num\_steps}, 1, 1)$ ▷ shape: $(\text{num\_steps}, n, C)$, static case

6: **end if**

7: $x_{\text{flipped}} = \text{torch.flip}(x, \text{dims} = [0])$ ▷ shape: $(\text{num\_steps}, n, C)$

8: $x_{\text{complex}} = x_{\text{flipped}}.\text{to}(\text{torch.cfloat})$

9: $x_B = \text{torch.matmul}(x_{\text{complex}}, B)$ ▷ shape: $(\text{num\_steps}, n, \text{hidden\_dim})$

10: $\Lambda_{\text{powers}} = \Lambda_{\text{powers}}.\text{permute}(2, 0, 1)$ ▷ shape: $(\text{num\_steps}, n, 1)$

11: $\Sigma_{\text{powers}} = \Sigma_{\text{powers}}.\text{transpose}(1, 0).\text{unsqueeze}(1)$ ▷ shape: $(\text{num\_steps}, 1, \text{hidden\_dim})$

12: $\text{scaled\_x\_B} = \Lambda_{\text{powers}} \cdot x_B \cdot \Sigma_{\text{powers}}$

13: $\texttt{out} = \text{scaled\_x\_B.cumsum}(\text{dim} = 0)$ ▷ shape: $(\text{num\_steps}, n, \text{hidden\_dim})$

14: $d_1, d_2, d_3 = \texttt{out.shape}$

15: $x_{\text{agg}} = \texttt{out.permute}(1, 2, 0).\text{reshape}(n, -1)$ ▷ shape: $(n, \text{num\_steps} \cdot \text{hidden\_dim})$

16: $x_{\text{agg}} = \text{matmul}($
$\quad\quad x = x_{\text{agg}},$
$\quad\quad \text{edge\_index} = \text{matrix\_p\_edge\_index},$
$\quad\quad \text{edge\_weight} = \text{matrix\_p\_edge\_weight}$
$\quad)$

17: $x_{\text{agg}} = x_{\text{agg}}.\text{reshape}(d_2, d_3, d_1).\text{permute}(2, 0, 1)$

18: $\texttt{out} = \text{mlp}(x_{\text{agg}}, \text{batch})$

---

acknowledge that there is no free lunch: we achieve a one-shot parallel implementation trading off GPU memory usage, since the whole tensor of shape (num_steps, $n$, hidden_dim), in line 9 of Algorithm 1, must fit into the GPU. However, with sufficient GPU memory, the entire MP-SSM block computation occurs in $10^{-3}$ seconds, see Figure 2. The figure also shows that MP-SSM scales

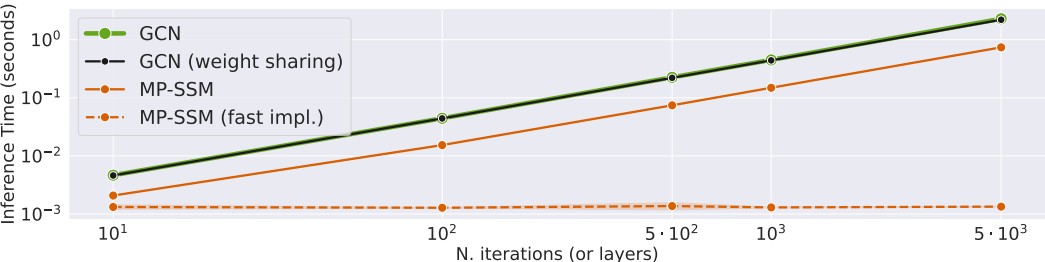

Figure 2: Inference time on a graph of $n = 100$ nodes (with number of edges 3058), input dimension $C = 1$, hidden_dim $= 32$, and increasing lengths $k = 10, 100, 500, 1000, 5000$. GCN is a standard GCN with `tanh` without residual with $k$ layers. GCN (weight sharing) is the same, but just one layer iterated $k$ times. MP-SSM baselines use both 1 block.

similarly to GCN and GCN (weight sharing), whose lines are overlapping, but it is slightly faster, owing to the lack of nonlinearity in the recurrence. This benefit grows with more iterations. On the other hand, the fast implementation of MP-SSM maintains constant runtime, provided enough GPU memory.

We observe that, a practical solution for memory-constrained GPUs is to chunk the computation for large number of recurernce steps. Assuming that $S$ is the maximum number of time steps a GPU can accommodate, the parallel implementation can be divided into $NC = \text{num\_steps}/S$ of chunks. These $NC$ chunks can then be processed sequentially and their results combined. This approach increases computational time roughly by a factor of $NC$ due to the sequentialization on a single GPU. Alternatively, if $NC$ GPUs are available, each chunk can be processed in parallel on a different GPU, since each intermediate computation depends on the current input (i.e., $\mathbf{U}_t$) and powers of the GSO and the learnable weights (i.e., $\mathbf{A}^{k-t-1}$ and $\mathbf{W}^{k-t-1}$) as shown in Equation (11). Then the results of each GPU can be merged to obtain the next prediction $\mathbf{X}_{t+1}$, greatly mitigating the slowdown.

Finally, we note that, unlike standard SSM models such as S4 and Mamba, which follow a Single-Input-Single-Output strategy (computing a separate SSM for each input channel and then mixing the results), our implementation in Algorithm 1 adopts a Multiple-Input-Multiple-Output strategy, enabling native handling of multivariate inputs.

## C PROOFS OF THE THEORETICAL STATEMENTS

Here, we provide all the proofs of lemmas, theorems, and corollaries stated in the main text.

### C.1 PROOF OF LEMMA 3.5

**Lemma.** Assume an undirected graph. The spectrum of the powers of the symmetric normalized adjacency matrix $\mathbf{A} = \mathbf{D}^{-\frac{1}{2}}(\tilde{\mathbf{A}} + \mathbf{I})\mathbf{D}^{-\frac{1}{2}}$ is contained in the interval $[-1, 1]$. The largest eigenvalue of $\mathbf{A}^t$ has absolute value of 1 with corresponding eigenvector $\mathbf{d} = \text{diag}(\mathbf{D}^{\frac{1}{2}})$, for all $t \geq 1$. In particular, the sequence of powers $[\mathbf{A}^t]_{t \geq 1}$ does not diverge or converge to the null matrix.

*Proof.* $\mathbf{A}^t = \left(\mathbf{D}^{-\frac{1}{2}}(\tilde{\mathbf{A}} + \mathbf{I})\mathbf{D}^{-\frac{1}{2}}\right)\left(\mathbf{D}^{-\frac{1}{2}}(\tilde{\mathbf{A}} + \mathbf{I})\mathbf{D}^{-\frac{1}{2}}\right)\ldots\left(\mathbf{D}^{-\frac{1}{2}}(\tilde{\mathbf{A}} + \mathbf{I})\mathbf{D}^{-\frac{1}{2}}\right) = \mathbf{D}^{-\frac{1}{2}}(\tilde{\mathbf{A}} + \mathbf{I})\left(\mathbf{D}^{-1}(\tilde{\mathbf{A}} + \mathbf{I})\right)^{t-1}\mathbf{D}^{-\frac{1}{2}}$. Now, $\mathbf{D}^{-1}(\tilde{\mathbf{A}} + \mathbf{I})$ is a stochastic matrix, and so also its powers are stochastic matrices. Therefore, $\mathbf{D}^{-\frac{1}{2}}\mathbf{A}^t\mathbf{D}^{\frac{1}{2}} = \left(\mathbf{D}^{-1}(\tilde{\mathbf{A}} + \mathbf{I})\right)^t$ is a stochastic matrix. The eigenvalues of a stochastic matrix are contained in the closed unitary disk (Meyer, 2023; Banerjee & Mehatari, 2016). Let, $\lambda_1, \ldots, \lambda_n$ all the eigenvalues (not necessarily distinct) of such a stochastic matrix, with corresponding eigenvectors $\mathbf{v}_1, \ldots, \mathbf{v}_n$. Thus, $\mathbf{D}^{-\frac{1}{2}}\mathbf{A}^t\mathbf{D}^{\frac{1}{2}}\mathbf{v}_l = \lambda_l\mathbf{v}_l$, from which it follows, multiplying both sides by $\mathbf{D}^{\frac{1}{2}}$, that $\mathbf{A}^t\mathbf{D}^{\frac{1}{2}}\mathbf{v}_l = \lambda_l\mathbf{D}^{\frac{1}{2}}\mathbf{v}_l$. This means that the eigenvalues of $\mathbf{A}^t$ are exactly the same of those of the stochastic matrix $\mathbf{D}^{-\frac{1}{2}}\mathbf{A}^t\mathbf{D}^{\frac{1}{2}}$ with eigenvectors $\mathbf{D}^{\frac{1}{2}}\mathbf{v}_1, \ldots, \mathbf{D}^{\frac{1}{2}}\mathbf{v}_n$, for all $t$. In particular, the assumption of undirected graph implies $\mathbf{A}$ is a symmetric matrix, thus we get that all eigenvalues of $\mathbf{A}^t$ are real and contained inside $[-1, 1]$, for all $t$. Since the spectral radius of a stochastic matrix is 1, and the vector $\mathbf{1}$ with all components equal to 1 is necessarily an eigenvector due to the row-sum being 1 for a stochastic matrix, then it follows that the largest eigenvalue of $\mathbf{A}^t$ is 1 and $\mathbf{d} = \text{diag}(\mathbf{D}^{\frac{1}{2}})$ is an eigenvector corresponding to eigenvalue 1, for all $t$. To see why the sequence of powers $[\mathbf{A}^t]_{t \geq 1}$ does not diverge or converge to the null matrix, we observe that, since $\mathbf{A}$ is symmetric, the Spectral Theorem implies we can diagonalize in $\mathbb{R}$ the matrix $\mathbf{A} = \mathbf{Q}\mathbf{\Lambda}\mathbf{Q}^\top$ with $\mathbf{Q}$ orthogonal matrix and $\mathbf{\Lambda}$ diagonal matrix of real eigenvalues. Powers of $\mathbf{A}$ can be written as $\mathbf{A}^t = (\mathbf{Q}\mathbf{\Lambda}\mathbf{Q}^\top)(\mathbf{Q}\mathbf{\Lambda}\mathbf{Q}^\top)\ldots(\mathbf{Q}\mathbf{\Lambda}\mathbf{Q}^\top) = \mathbf{Q}\mathbf{\Lambda}^t\mathbf{Q}^\top$. Thus the eigenvalues of $\mathbf{A}^t$ are $\lambda_l^t$, for $l = 1, \ldots, n$. We already proved that the eigenvalues $\lambda_n \leq \ldots \leq \lambda_1$ are contained in the real interval $[-1, 1]$. Hence, this ensures that the sequence of powers cannot diverge. On the other hand, we can spectrally decompose symmetric matrices as follows (Haykin, 2009), $\mathbf{A}^t = \sum_{l=1}^n \lambda_l^t\mathbf{q}_l\mathbf{q}_l^\top$, where $\mathbf{q}_l$ is the eigenvector corresponding to the eigenvalue $\lambda_l$. Thus, for large values of $t$, the spectral components corresponding to eigenvalues strictly less than 1 in absolute value vanish, so the matrix $\mathbf{A}^t$ approaches the sum of terms corresponding to eigenvalues with absolute value equal to 1. This proves that the sequence of powers cannot converge to the null matrix. $\square$

## C.2 PROOF OF THEOREM 3.4

**Theorem.** The Jacobian of the linear recurrent equation of an MP-SSM block, from node $j$ at layer $s$ to node $i$ at layer $t \geq s$, can be computed exactly, and it has the following form:

$$\frac{\partial \mathbf{X}_t^{(i)}}{\partial \mathbf{X}_s^{(j)}} = \underbrace{(\mathbf{A}^{t-s})_{ij}}_{\text{scalar}} \underbrace{(\mathbf{W}^\top)^{t-s}}_{\text{matrix}}.$$

*Proof.* In this proof we use the notation $(\mathbf{M})_{ij}$ to denote the $(i,j)$ entry of a matrix $\mathbf{M}$, and $\mathbf{M}^{(i)}$ to denote the $i$-th row of a matrix $\mathbf{M}$. Let us start with the recurrent equation $\mathbf{X}_{t+1} = \mathbf{A}\mathbf{X}_t\mathbf{W} + \mathbf{U}_{t+1}\mathbf{B}$. Therefore, the $i$-th node features are updated as follows: $\mathbf{X}_{t+1}^{(i)} = \sum_{l=1}^n (\mathbf{A})_{il}\mathbf{X}_t^{(l)}\mathbf{W} + \mathbf{U}_{t+1}^{(i)}\mathbf{B}$. Now, the only term involving $\mathbf{X}_t^{(j)}$ is $(\mathbf{A})_{ij}\mathbf{X}_t^{(j)}\mathbf{W}$. Therefore, the Jacobian reads $\dfrac{\partial \mathbf{X}_{t+1}^{(i)}}{\partial \mathbf{X}_t^{(j)}} =$

$\dfrac{\partial}{\partial \mathbf{X}_t^{(j)}}\left((\mathbf{A})_{ij}\mathbf{X}_t^{(j)}\mathbf{W}\right)$. Now, given a row vector $\mathbf{x} \in \mathbb{R}^c$ and a square matrix $\mathbf{M}$, then the function $\mathbf{f}(\mathbf{x}) = \mathbf{x}\mathbf{M}$, whose $i$-th component is $f_i = \sum_{l=1}^c x_l(\mathbf{M})_{li}$, has derivatives $\frac{\partial f_i}{\partial \mathbf{x}_j} = \frac{\partial}{\partial x_j}(x_j(\mathbf{M})_{ji}) = (\mathbf{M})_{ji}$. Hence, the Jacobian is $\frac{\partial \mathbf{f}}{\partial \mathbf{x}} = \mathbf{M}^\top$. Therefore, it holds $\dfrac{\partial \mathbf{X}_{t+1}^{(i)}}{\partial \mathbf{X}_t^{(j)}} = (\mathbf{A})_{ji}\mathbf{W}^\top$. For the case of non-consecutive time steps, we can unfold the recurrent equation $\mathbf{X}_{t+1} = \mathbf{A}\mathbf{X}_t\mathbf{W} + \mathbf{U}_{t+1}\mathbf{B}$ between any two time steps $s \leq t$, as follows:

$$\mathbf{X}_t = \mathbf{A}^{t-s}\mathbf{X}_s\mathbf{W}^{t-s} + \sum_{l=0}^{t-s-1} \mathbf{A}^l \mathbf{U}_{t-l}\mathbf{B}\mathbf{W}^i. \tag{23}$$

From the unfolded recurrent equation (23) of a MP-SSM we can see that the only term involving $\mathbf{X}_s$ is $\mathbf{A}^{t-s}\mathbf{X}_s\mathbf{W}^{t-s}$. Thus, the Jacobian reads $\dfrac{\partial \mathbf{X}_t^{(i)}}{\partial \mathbf{X}_s^{(j)}} = \dfrac{\partial}{\partial \mathbf{X}_s^{(j)}}\left((\mathbf{A}^{t-s}\mathbf{X}_s\mathbf{W}^{t-s})^{(i)}\right) = \dfrac{\partial}{\partial \mathbf{X}_s^{(j)}}\left((\mathbf{A}^{t-s})_{ij}\mathbf{X}_s^{(j)}\mathbf{W}^{t-s}\right) = (\mathbf{A}^{t-s})_{ij}(\mathbf{W}^\top)^{t-s}$. $\qquad\square$

## C.3 PROOF OF THEOREM 3.6

**Theorem.** Assume a connected graph, and the GSO defined in Eq. (1). Then, for large values of $t - s$, the Jacobian of the linear recurrent equation of an MP-SSM block, from node $j$ at layer $s$ to node $i$ at layer $t \geq s$, admits the following approximation:

$$\frac{\partial \mathbf{X}_t^{(i)}}{\partial \mathbf{X}_s^{(j)}} \approx \frac{\sqrt{(1+d_i)(1+d_j)}}{|V| + 2|E|}(\mathbf{W}^\top)^{t-s},$$

where $d_l = \sum_{j=1}^n (\tilde{\mathbf{A}})_{lj}$ is the degree of the $l$-th node.

*Proof.* We provide an estimation of the term $(\mathbf{A}^{t-s})_{ij}$ for the case of large values of $t - s$, and assuming a connected graph. We use the decomposition $\mathbf{A}^{t-s} = \sum_{l=1}^n \lambda_l^{t-s}\mathbf{q}_l\mathbf{q}_l^\top$, where $\mathbf{q}_l$ is the unitary eigenvector corresponding to the eigenvalue $\lambda_l$. As discussed in the proof of Lemma 3.5, for large values of $t - s$, all the spectral components corresponding to eigenvalues strictly less than 1 (in absolute value) tend to converge to 0. Moreover, by the Perron–Frobenius theorem for irreducible non-negative matrices (Horn & Johnson, 2012), since the graph is connected and with self-loops, there is only one simple eigenvalue equal to 1, and $-1$ cannot be an eigenvalue. Thus it holds the approximation $\mathbf{A}^{t-s} \approx \mathbf{q}_1\mathbf{q}_1^\top$. Now thanks to Lemma 3.5, we know that $\mathbf{q}_1$ must be the vector $\mathbf{d} = \text{diag}(\mathbf{D}^{\frac{1}{2}})$ normalised to be unitary, and $\mathbf{D}$ is the degree matrix of $\tilde{\mathbf{A}} + \mathbf{I}$. Thus, $\mathbf{q_1} = \dfrac{(\sqrt{1+d_1}, \ldots, \sqrt{1+d_n})}{\sqrt{\sum_{l=1}^n (1+d_l)}}$, where $d_l = \sum_{j=1}^n (\tilde{\mathbf{A}})_{lj}$ is the degree of the $l$-th node. Therefore,

$(\mathbf{q}_1\mathbf{q}_1^\top)_{ij} = \dfrac{\sqrt{(1+d_i)(1+d_j)}}{n + \sum_{l=1}^n d_l} = \dfrac{\sqrt{(1+d_i)(1+d_j)}}{|V| + 2|E|}$. $\qquad\square$

### C.3.1 EXAMPLE OF A BAD SCENARIO FOR EQ. (7)

Fig. 3 illustrates an example of a bad scenario for Eq. (7), i.e., a chain of $m$ cliques of order $d$ connected via bridge-nodes of degree 2 (the minimum to connect them). In the Figure, we consider $m = 6$ and $d = 10$. The pair of bridge nodes $i$ and $j$ depicted in red in Fig. 3 are 12 hops apart, so it can be considered a relatively long-term interaction.

In the long-term approximation given by Eq. (7), the local sensitivity between two bridge nodes of this topology scales as $\frac{1}{md^2}$, for long chains ($m$ large) and big cliques ($d$ large). In fact, in such a graph the vast majority of nodes has degree approximately $d - 1$, thus $\sum_{l=1}^{n} d_l \approx n(d-1)$. Specifically, there are exactly $m - 1$ nodes of degree 2 (bridge nodes), and $md$ nodes with degree approximately $d - 1$. Now, $n = m - 1 + md \approx md$, therefore $n + \sum_{l=1}^{n} d_l \approx n + n(d-1) = nd \approx md^2$. Scaling to long chains and large cliques, this approximation becomes more accurate, and so the local sensitivity between two bridge nodes is rescaled by the term $\frac{\sqrt{(1+d_i)(1+d_j)}}{n+\sum_{l=1}^{n} d_l} \approx \frac{3}{md^2}$.

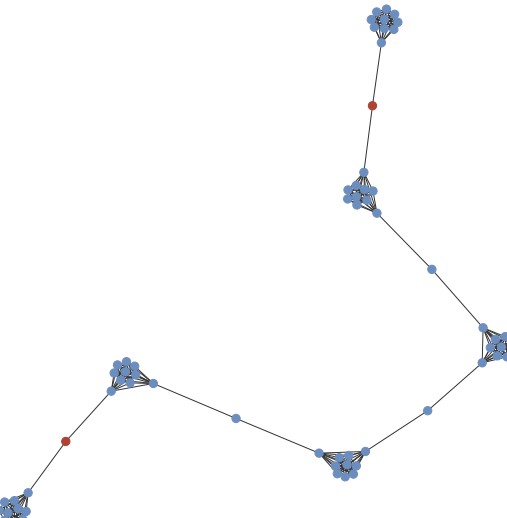

Figure 3: A chain of six cliques (containing ten nodes each) connected via bridge-nodes of degree 2. The pair of red nodes is a pair of nodes that minimizes the quantity in Eq. (7). Note that the red nodes are 12 hops apart, so it can be considered long-term.

### C.4 PROOF OF COROLLARY 3.7

**Corollary.** Assume a connected graph, and the GSO of Eq. (1). Then, for large values of $t - s$, the following lower bound for the minimum local sensitivity of the linear recurrent equation of an MP-SSM block holds:

$$\frac{2}{|V| + 2|E|} ||\mathbf{W}^{t-s}|| \leq \min_{i,j} \mathcal{S}_{ij}(t - s). \tag{24}$$

*Proof.* In the deep regime, we can use the approximation of Eq. (7) of $\frac{\partial \mathbf{X}_t^{(i)}}{\partial \mathbf{X}_s^{(j)}} \approx \frac{\sqrt{(1+d_i)(1+d_j)}}{|V| + 2|E|} (\mathbf{W}^{\top})^{t-s}$. Therefore, we have:

$$\min_{i,j} \left\| \frac{\partial \mathbf{X}_t^{(i)}}{\partial \mathbf{X}_s^{(j)}} \right\| \approx \frac{1}{|V| + 2|E|} \left\| (\mathbf{W}^{\top})^{t-s} \right\| \min_{i,j} \sqrt{(1+d_i)(1+d_j)} \geq \frac{2}{|V| + 2|E|} \left\| (\mathbf{W}^{\top})^{t-s} \right\|,$$

where the last inequality holds since the minimum degree value of a node in a connected graph is 1. Thus, we conclude that $\min_{i,j} \mathcal{S}_{ij}(t-s) \geq \frac{2}{|V|+2|E|}||(\mathbf{W}^\top)^{t-s}|| = \frac{2}{|V|+2|E|}||\mathbf{W}^{t-s}||$, noticing that $||\mathbf{W}^\top|| = ||\mathbf{W}||$. □

## C.5 PROOF OF THEOREM 3.11

**Theorem.** Assume a connected graph. The global sensitivity of the linear recurrent equation of an MP-SSM block is lower bounded as follows:

$$\mathcal{S}(t-s) \geq \frac{\rho(\mathbf{A})^{t-s}}{|V|}||\mathbf{W}^{t-s}||,$$

where $\rho(\mathbf{A})$ is the spectral radius of the GSO. Thus, for the GSO of Eq. (1), it holds the lower bound $\mathcal{S}(t-s) \geq \frac{1}{|V|}||\mathbf{W}^{t-s}||$.

*Proof.* By Eqs. (5), (6) and (9), we get $\mathcal{S}(t-s) = \max_{i,j} |(\mathbf{A}^{t-s})_{ij}|||(\mathbf{W}^\top)^{t-s}|| = \max_{i,j} |(\mathbf{A}^{t-s})_{ij}|||\mathbf{W}^{t-s}||$. Let us define $n = |V|$ the number of nodes. The square of the maximum entry of an $(n,n)$ matrix $\mathbf{M}$ is always greater than the arithmetic mean of all the square coefficients, in other words, $\frac{||\mathbf{M}||_F^2}{n^2} \leq \max_{i,j} \mathbf{M}_{i,j}^2$, where $||\mathbf{M}||_F$ denotes the Frobenius norm. Therefore, $\frac{||\mathbf{M}||_F}{n} \leq \max_{i,j} |\mathbf{M}_{i,j}|$. Now, the symmetry of $\mathbf{A}$ implies there are $\lambda_1, \ldots, \lambda_n$ real eigenvalues with corresponding orthonormal eigenvectors $\mathbf{q}_1, \ldots, \mathbf{q}_n$ so that we can decompose $\mathbf{A}^{t-s} = \sum_{l=1}^n \lambda_l^{t-s} \mathbf{q}_l \mathbf{q}_l^\top$. Thus, the Frobenius norm is $||\mathbf{A}^{t-s}||_F = \sqrt{\sum_{l=1}^n \lambda_l^{2(t-s)} ||\mathbf{q}_l||^2} = \sqrt{\sum_{l=1}^n \lambda_l^{2(t-s)}} \geq |\lambda_1|^{t-s}$, where $|\lambda_1|$ is the largest in absolute value between all the eigenvalues, i.e. the spectral radius $\rho(\mathbf{A})$.

$$\max_{i,j} |(\mathbf{A}^{t-s})_{ij}| \geq \frac{||\mathbf{A}^{t-s}||_F}{n} \geq \frac{\rho(\mathbf{A})^{t-s}}{n}, \tag{25}$$

from which we get the thesis

$$\mathcal{S}(t-s) = \max_{i,j} |(\mathbf{A}^{t-s})_{ij}|\, ||\mathbf{W}^{t-s}|| \geq \frac{\rho(\mathbf{A})^{t-s}}{n}||\mathbf{W}^{t-s}||.$$

For the particular case of GSO given by Eq. (1), the spectral radius $\rho(\mathbf{A})$ is exactly 1 due to Lemma 3.5. □

## C.6 PROOF OF THEOREM 3.13

**Theorem.** Let us consider a GCN network that aggregates information from $k$ hops away, i.e., with $k$ layers, equipped with the ReLU activation function. Then, the GCN vanishes information at a $2^{-\frac{k}{2}}$ faster rate than our MP-SSM block with a number $k$ of linear recurrent steps.

*Proof.* The state-update equation of a GCN with a residual connection is $\mathbf{X}_{t+1} = \sigma(\mathbf{A}\mathbf{X}_t\mathbf{W} + \mathbf{X}_t)$. Therefore, the features of $i$-th node at time $t+1$ are updated as $\mathbf{X}_{t+1}^{(i)} = \sigma\left(\sum_{l=1}^n (\mathbf{A})_{il}\mathbf{X}_t^{(l)}\mathbf{W} + \mathbf{X}_t^{(i)}\right)$. Similarly to the proof of theorem 3.4, we can write

$$\frac{\partial \mathbf{X}_{t+1}^{(i)}}{\partial \mathbf{X}_t^{(j)}} = \frac{\partial}{\partial \mathbf{X}_t^{(j)}}\left(\sigma\left((\mathbf{A})_{ij}\mathbf{X}_t^{(j)}\mathbf{W}\right)\right) =$$

$$= \text{diag}\left(\sigma'\left((\mathbf{A})_{ij}\mathbf{X}_t^{(j)}\mathbf{W}\right)\right)(\mathbf{A})_{ij}\mathbf{W}^\top,$$

where we assumed that $i \neq j$, so that the residual connection term does not appear in the derivative w.r.t. $\mathbf{X}_t^{(j)}$. Since we are considering $\sigma = \text{ReLU}$, the diagonal entries $\sigma'\left((\mathbf{A})_{ij}\mathbf{X}_t^{(j)}\mathbf{W}\right)$ are either 0 or 1. Let's assume that the components of the vector $\sigma'\left((\mathbf{A})_{ij}\mathbf{X}_t^{(j)}\mathbf{W}\right)$ are independent and identically distributed (i.i.d.) Bernoulli random variables, each with probability $\frac{1}{2}$ of taking the value

0. Now, let's consider a walk $\{(i_t, j_t)\}_{t=0}^{k-1}$ of length $k$ connecting the $j$-th node at a reference time $t = 0$ to the $i$-th node at time $t = k$. Then, the Jacobian of GCN along such a walk reads:

$$\frac{\partial \mathbf{X}_k^{(i)}}{\partial \mathbf{X}_0^{(j)}} = \prod_{t=0}^{k-1} \mathbf{P}_t \mathbf{M}_t,$$

where $\mathbf{P}_t = \text{diag}\left(\sigma'\left((\mathbf{A})_{i_t j_t} \mathbf{X}_t^{(j_t)} \mathbf{W}\right)\right)$, and $\mathbf{M}_t = (\mathbf{A})_{i_t j_t} \mathbf{W}^\top$. On the other hand, the Jacobian of the linear recurrent equation (3) of an MP-SSM block, in the static case with a number $k$ of linear recurrent steps computed along the same walk reads:

$$\frac{\partial \mathbf{X}_k^{(i)}}{\partial \mathbf{X}_0^{(j)}} = \prod_{t=0}^{k-1} \mathbf{M}_t.$$

We aim to prove that, for a generic vector $\mathbf{x}$ with entries i.i.d. random variables distributed symmetrically about zero (e.g. according to a Normal distribution with zero mean), it holds the approximation $||\prod_{t=0}^{k-1} \mathbf{P}_t \mathbf{M}_t \mathbf{x}|| \approx 2^{-\frac{k}{2}} ||\prod_{t=0}^{k-1} \mathbf{M}_t \mathbf{x}||$. We prove the thesis using a recursive argument. First, we observe that, denoting $\mathbf{y} = \mathbf{M}_0 \mathbf{x}$, then we can write

$$||\mathbf{P}_0 \mathbf{M}_0 \mathbf{x}||^2 = ||\mathbf{P}_0 \mathbf{y}||^2 = (p_1 y_1)^2 + \ldots + (p_n y_n)^2. \tag{26}$$

Now, since the $p_i$ are assumed i.i.d. Bernoulli random variables, each with probability $\frac{1}{2}$ of taking the value 0, in the sum of (26), roughly a portion of half of the contributions from $\mathbf{y}$ are zeroed-out due to action of $\mathbf{P}_0$. Therefore,

$$||\mathbf{P}_0 \mathbf{M}_0 \mathbf{x}||^2 = ||\mathbf{P}_0 \mathbf{y}||^2 \approx \frac{1}{2} ||\mathbf{y}||^2 = \frac{1}{2} ||\mathbf{M}_0 \mathbf{x}||^2. \tag{27}$$

Note that the larger the dimension of the graph $n$, the more accurate the approximation of (27). Therefore, we conclude that $||\mathbf{P}_0 \mathbf{M}_0 \mathbf{x}|| \approx 2^{-\frac{1}{2}} ||\mathbf{M}_0 \mathbf{x}||$. Now, we proceed recursively by denoting $\tilde{\mathbf{x}}_t = \mathbf{P}_{t-1} \mathbf{M}_{t-1} \ldots \mathbf{P}_0 \mathbf{M}_0 \mathbf{x}$, and defining the scalars $c_t = \frac{||\mathbf{M}_t \tilde{\mathbf{x}}_t||}{||\tilde{\mathbf{x}}_t||} > 0$, for all $t = 1, \ldots, k-1$. Then, we can write

$$||\mathbf{P}_{k-1} \mathbf{M}_{k-1} \mathbf{P}_{k-2} \mathbf{M}_{k-2} \ldots \mathbf{P}_0 \mathbf{M}_0 \mathbf{x}|| =$$
$$= ||\mathbf{P}_{k-1} \mathbf{M}_{k-1} \tilde{\mathbf{x}}_{k-1}|| \approx$$
$$\approx 2^{-\frac{1}{2}} ||\mathbf{M}_{k-1} \tilde{\mathbf{x}}_{k-1}|| =$$
$$= 2^{-\frac{1}{2}} c_{k-1} ||\tilde{\mathbf{x}}_{k-1}|| =$$
$$= 2^{-\frac{1}{2}} c_{k-1} ||\mathbf{P}_{k-2} \mathbf{M}_{k-2} \tilde{\mathbf{x}}_{k-2}|| \approx$$
$$\approx 2^{-\frac{1}{2}} c_{k-1} 2^{-\frac{1}{2}} c_{k-2} ||\tilde{\mathbf{x}}_{k-2}|| \approx \ldots$$
$$\approx 2^{-\frac{k}{2}} c_{k-1} c_{k-2} \ldots c_0 ||\mathbf{x}||.$$

On the other hand, for the case of MP-SSM, it reads:

$$||\mathbf{M}_{k-1} \mathbf{M}_{k-2} \ldots \mathbf{M}_0 \mathbf{x}|| = c_{k-1} ||\mathbf{M}_{k-2} \ldots \mathbf{M}_0 \mathbf{x}|| =$$
$$= c_{k-1} c_{k-2} ||\mathbf{M}_{k-3} \ldots \mathbf{M}_0 \mathbf{x}|| = \ldots$$
$$= c_{k-1} c_{k-2} \ldots c_0 ||\mathbf{x}||.$$

This proves that a standard GCN vanishes information $2^{-\frac{k}{2}}$ faster than MP-SSM.
We assumed weight sharing in the GCN, but the same proof holds assuming different weights $\mathbf{W}_1, \ldots, \mathbf{W}_k$ at each GCN layer, by simply using the same exact weight matrices for the linear equation of MP-SSM. □

## D   THE VANISHING GRADIENT TENDENCY IN NONLINEAR MPNNS

Let us consider a highly connected graph without bottlenecks, such that the transfer of messages from any node to any other node is not affected by issues due to structural properties of the graph.

However, in the deep regime, the presence of a nonlinearity at each time step can lead the global sensitivity (as defined in Eq. (9)) to be vanishing small.

For an MP-SSM block, the local sensitivity $\mathcal{S}_{ij}(t-s)$ of the features of the $i$-th node to features of the $j$-th node after $t-s$ applications of message-passing aggregations, is exactly the norm of the Jacobian of Eq. (6), i.e. the norm of the product of the $(i,j)$-entry of $\mathbf{A}^{t-s}$ and the matrix $(\mathbf{W}^{\top})^{t-s}$. For standard MPNN approaches, the local sensitivity has a more complicated expression due to nonlinearities at each aggregation step, but usually there are 3 key contributors: one from several multiplications of the shift operator (akin to $\mathbf{A}^{t-s}$ in our MP-SSM), one from several multiplications of the weights (akin to $(\mathbf{W}^{\top})^{t-s}$ in our MP-SSM), and one from several multiplications of the derivative of the nonlinearity evaluated on the sequence of embeddings $\mathbf{D}(s), \mathbf{D}(s+1), .., \mathbf{D}(t)$. Usually the nonlinearity is pointwise, so $\mathbf{D}(t)$ is a diagonal matrix with entries usually in $[0,1]$, thus contributing to vanishing the gradient more and more at each time step. Hence, if the subsequent multiplications of weights and nonlinearity-based terms tend to vanish, while the powers of the shift operator $\mathbf{A}$ are bounded (as it is for the case of the symmetrically normalized adjacency with self-loops, proved in Lemma 4.5) then the local sensitivity tends to vanish *for all pair of nodes*, for $t-s$ large enough. This will be reflected in the global sensitivity, which also will tend to vanish, for $t-s$ large enough. This demonstrates that global sensitivity effectively quantifies the severity of vanishing gradient issues in MPNN models plagued by this problem. Note further that the local sensitivity of the linear recurrence in each block of our MP-SSM has the exact form of $||(\mathbf{A}^{t-s})_{ij}(\mathbf{W}^{\top})^{t-s}||$, while for standard MPNN approaches with nonlinearities at each time step the vanishing effect will be stronger, as we formally proved for the case of GCN in Theorem 3.13.

### D.1 Additional Vanishing Effects Beyond Variance-Preserving Scaling.

The analysis in Theorem 3.13 quantifies the contraction introduced by pointwise nonlinearities through a Bernoulli$(0.5)$ model of the ReLU derivative masks. One may attempt to counteract this contraction by rescaling the activation function (e.g., using $\sqrt{2}\,\mathrm{ReLU}$) or by similar adopting variance-preserving initialization schemes such as He initialization He et al. (2015), which are designed to maintain stable signal norms in feedforward architectures. These techniques effectively compensate for the *expected* shrinkage caused by the diagonal derivative matrices. Nonetheless, when nonlinear transformations are repeatedly applied through the *same* weight matrix, as in weight-sharing settings or recurrent message-passing, the i.i.d. assumptions underlying variance-preserving theory no longer hold. As observed in prior work on deep and recurrent networks Sussillo & Abbott (2014), repeated application of a fixed operator induces directional compounding of contractions or expansions along its singular directions, leading to a substantially stronger vanishing effect than predicted by independent-layer analyses. In the remainder of this subsection, we discuss this phenomenon in more detail and provide an empirical illustration showing that, even after compensating for Bernoulli contraction via $\sqrt{2}$-scaling, significant gradient decay persists at large depths.

To illustrate this phenomenon, we report a simple controlled experiment in an RNN-like setting (no graph structure) over $k = 1000$ recurrent steps. We compare (i) a ReLU RNN with weight sharing of equation $\mathbf{x}_{t+1} = \mathrm{ReLU}(\mathbf{W}\,\mathbf{x}_t + \mathbf{B}\mathbf{u}_{t+1})$, (ii) the same model with $\sqrt{2}$-scaled ReLU (as in variance-preserving scheme), i.e. $\mathbf{x}_{t+1} = \sqrt{2}\,\mathrm{ReLU}(\mathbf{W}\,\mathbf{x}_t + \mathbf{B}\mathbf{u}_{t+1})$, and (iii) a linear RNN, i.e. $\mathbf{x}_{t+1} = \mathbf{W}\,\mathbf{x}_t + \mathbf{B}\mathbf{u}_{t+1}$. For each setting we measure the spectral norm of the Jacobian $\left\|\frac{\partial \mathbf{x}_k}{\partial \mathbf{x}_0}\right\|$ over five independent trials. The recurrent matrix $\mathbf{W} \in \mathbb{R}^{128 \times 128}$ is sampled from a standard normal distribution and rescaled to have spectral radius 1; $\mathbf{B} \in \mathbb{R}^{128 \times 10}$ and the inputs $\mathbf{u}_t \in \mathbb{R}^{10}$ are sampled i.i.d. from a standard normal distribution for each time step.

| Scenario | $\sqrt{2}$ scaling | Mean Jacobian norm over 5 trials |
|---|---|---|
| Shared $W$, ReLU | No | $5.52 \times 10^{-174}$ |
| Shared $W$, ReLU | Yes | $9.74 \times 10^{-26}$ |
| Shared $W$, Linear | No | $2.025$ |

Table 5: Jacobian norm $\left\|\frac{\partial \mathbf{x}_k}{\partial \mathbf{x}_0}\right\|$ for $k = 1000$ recurrent steps under different nonlinearities and initialization schemes.

The results in Table 5 show that, even if $\sqrt{2}$-scaling is devised to preserve layerwise variance, the repeated application of the same $\mathbf{W}$ causes contractions or expansions to compound along the same singular directions of $\mathbf{W}$. By contrast, in the feedforward / independent-weights case each random matrix $\mathbf{W}_t$ rotates and redistributes contraction across directions, which mitigates consistent compounding. For this reason, the Bernoulli-based estimate of $2^{-k/2}$ of Theorem 3.13 should be interpreted as a best-case contraction rate. Indeed, for $k = 1000$, the Bernoulli assumption in our theorem predicts $2^{-500} \approx 10^{-150}$, yet our empirical results show an even stronger vanishing effect (around $10^{-175}$). The empirical gap is fully consistent with theory: it reflects the additional contraction created by repeatedly applying the same operator, deviating from the i.i.d. assumption. This geometric compounding effect is not captured by the Bernoulli analysis alone, yet it accumulates on top of it and becomes significant at large depths. Thus, the vanishing gap between MP-SSM and opportunely rescaled nonlinear models still emerges empirically at large depths.

## E  RELATION TO OTHER GNNS BASED ON STATE-SPACE MODELING

**Static Graph Modelling.** In the recent literature, we can find GNNs that leverage the state-space model formalism. An example is that of S4G (Song et al., 2024). Despite both S4G and our MP-SSM leverage the same formalism, there are key differences that distinguish the two models. (i) Our MP-SSM operates natively on graphs, while S4G requires graph-to-sequence conversion. S4G, like other attempts to incorporate SSMs into graph learning (e.g., Wang et al. (2024a); Behrouz & Hashemi (2024)), chooses to first extract sequences from a static graph and then apply an SSM module, which in the case of S4G is the S4 model (Gu et al., 2021). We note that this process compresses graph neighborhoods into linear sequences and thus may not fully retain the original structural relationships. In contrast, our MP-SSM maintains the graph structure and operates on it directly. Moreover, differently from S4G, our MP-SSM framework seamlessly extends naturally to temporal graphs as well as time-varying topologies. (ii) To create a sequence, S4G collapses the $k$-hop neighborhood of a root node into a single embedding at step $k$ of a surrogate input sequence. In other words, a node in the $k$-hop shell contributes only to step $k$, regardless of the richer set of longer or alternative paths through which information could propagate. MP-SSM instead aggregates information along all walks (including cycles) in accordance with the powers of the GSO of choice, leading to a fundamentally different inductive bias that more faithfully reflects graph structure. (iii) S4G uses a single-input single-output (SISO) sequential architecture, while MP-SSM is inherently a multi-input multi-output (MIMO) model. In modern state-space models, MIMO architectures have already been shown to provide strictly greater expressive capacity than SISO variants (Smith et al., 2022). In our setting of spatiotemporal learning, this advantage is further supported by graph-based studies demonstrating that jointly modeling multiple time series through a relational structure yields more informative representations and superior predictive performance compared to treating each series independently (Cini et al., 2023b; Spadon et al., 2022). Finally, (iv) our MP-SSM provides a theoretical general analysis in which the sensitivity directly depends on the graph topology and the chosen GSO. Specifically, our analysis enables a quantitative understanding of graph learning issues like vanishing gradients and oversquashing without considering the collapsed surrogate of a graph, i.e., a sequence. This analysis is possible thanks to the precise computations in our theoretical framework, which, to the best of our knowledge, are not present in the existing literature.

**Temporal Graph Modelling.** In the recent literature, we can find temporal graph models that leverage the state-space approach. The MP-SSM presents a simplified yet effective recurrent architecture for temporal graph modeling, offering clear advantages in architectural design when compared to alternatives such as GGRNN (Ruiz et al., 2020) or GraphSSM (Li et al., 2024). The MP-SSM recurrent dynamics are governed by a simple linear diffusion on the graph:

$$\mathbf{X}_{t+1} = \mathbf{A}\mathbf{X}_t\mathbf{W} + \mathbf{U}_{t+1}\mathbf{B}. \tag{28}$$

In contrast, the GGRNN recurrent equation (in its simplest form, without gating mechanisms) adopts a more elaborate design:

$$\mathbf{X}_{t+1} = \sigma\left(\sum_{j=0}^{K-1} \mathbf{A}^j\mathbf{X}_t\mathbf{W}_j + \sum_{j=0}^{K-1} \mathbf{A}^j\mathbf{U}_{t+1}\mathbf{B}_j\right), \tag{29}$$

where multiple powers of the shift operator, $\mathbf{A}$, are used to aggregate information from both previous embedding $\mathbf{X}_t$ and current input features $\mathbf{U}_{t+1}$, weighted with several learnable matrices, $\mathbf{W}_j$ and $\mathbf{B}_j$, which are applied for different $j$ values, and finally, applying a nonlinearity *at each time step*.

The key distinguishing feature of MP-SSM is the *absence of nonlinearity in the recurrent update*, with the only nonlinear transformation appearing in a downstream MLP decoder, typically composed of two dense layers with an activation function in between. This feature also allows for a fast implementation of the recurrence, since it can be unfolded to get a closed-form solution, see Appendix B. Moreover, in an MP-SSM block, the same weights, $\mathbf{W}, \mathbf{B}$ and MLP parameters, are shared across all time steps, ensuring *strict weight sharing throughout the sequence*. Moreover, our methodology implements a stack of MP-SSM blocks to build richer representations, differently from GGRNN where only one layer of recurrent computation is performed.

On the other hand, the GraphSSM model (Li et al., 2024) adopts a strategy of stacking several GraphSSM blocks similar to MP-SSM, but their building blocks are fundamentally different from our MP-SSM block. In fact, a GraphSSM block processes the spatio-temporal input sequence $[\mathbf{U}_t]$ in three main stages, see Appendix D.2 of Li et al. (2024). First, a GNN backbone is applied to the input sequence, generating a corresponding sequence of node embeddings $\mathbf{X}_t$. Next, each embedding is mixed with the one from the previous time step $\mathbf{X}_{t-1}$, producing a smoothed temporal embedding $\mathbf{H}_t$. This mixed sequence $[\mathbf{H}_t]$ is then treated as a multivariate time series and passed through an SSM layer (such as S4, S5, or S6) to yield the final sequence $[\mathbf{Y}_t]$ as the output of a GraphSSM block. Our approach is conceptually simpler, as it integrates both the GNN diffusive dynamics and sequence-based processing within a unified linear recurrence (Eq. (28)) followed by a shared MLP applied across time steps. In this sense, MP-SSM embeds the core principles behind modern SSMs, which are the very principles that have driven the success of sequential modeling, directly into the graph processing framework. In contrast, GraphSSM merely combines GNN and SSM backbones in a modular fashion to address temporal graph tasks, without deeply integrating their underlying mechanisms.

In Table 6, we provide a direct comparison between MP-SSM, GGRNN, and GraphSSM, on the Metr-LA and PeMS-Bay datasets. To ensure a fair and comprehensive comparison, we computed MAE, RMSE, and MAPE for all three models: MP-SSM, GGRNN, and GraphSSM. We used GGRNN without gating mechanisms, as it achieved the best performance on Metr-LA according to (Ruiz et al., 2020, Table IV), and GraphSSM-S4, since the authors reported in Li et al. (2024) that their experiments were primarily conducted using the S4 architecture. As the results show, our method consistently and significantly outperforms both GGRNN and GraphSSM across all three metrics on both datasets.

Table 6: Multivariate time series forecasting on the Metr-LA and PeMS-Bay datasets for Horizon 12. **Best** results for each task are in bold.

| Model | Metr-LA | | | PeMS-Bay | | |
|---|---|---|---|---|---|---|
| | MAE ↓ | RMSE ↓ | MAPE ↓ | MAE ↓ | RMSE ↓ | MAPE ↓ |
| GGRNN | 3.88 | 8.14 | 10.59% | 2.34 | 5.14 | 5.21% |
| GraphSSM-S4 | 3.74 | 7.90 | 10.37% | 1.98 | 4.45 | 4.77% |
| MP-SSM (ours) | **3.17** | **6.86** | **9.21%** | **1.62** | **4.22** | **4.05%** |

## F   Multi-hop Interpretation of a Deep MP-SSM Architecture

MP-SSM is fundamentally different from multi-hop GNNs approaches: it operates through strictly 1-hop message passing at each iteration and does not perform aggregation from far-away hops by design. Nonetheless, to better understand its behavior in deeper architectures, we explore how a multi-hop perspective can be used for interpretation, drawing contrasts with a representative multi-hop model, Drew (Gutteridge et al., 2023). For this purpose, let us consider the static case, with the input being the sequence $[\mathbf{U}_1, \ldots, \mathbf{U}_1]$. The linearity of the recurrent equation of an MP-SSM block

allows us to unfold the recurrent equation as follows:

$$\mathbf{X}_{k+1} = \mathbf{A}^{k+1}\mathbf{X}_0\mathbf{W}^{k+1} + \sum_{i=0}^{k}\mathbf{A}^i\mathbf{U}_1\mathbf{B}\mathbf{W}^i. \tag{30}$$

Therefore, assuming a zero initial state and including the MLP into the equation, we have the following expression in the output of the first MP-SSM block:

$$\mathbf{Y}_{k+1} = \mathrm{MLP}\Big(\sum_{i=0}^{k}\mathbf{A}^i\mathbf{U}_1\mathbf{B}\mathbf{W}^i\Big). \tag{31}$$

Due to the various powers of the shift operator $\mathbf{I}, \mathbf{A}, \mathbf{A}^2, \ldots, \mathbf{A}^k$, we can interpret Eq. (31) as a $k$-hop aggregation of the input graph $\mathbf{U}_1$. Now, the sequence $[\mathbf{Y}_{k+1}, \ldots, \mathbf{Y}_{k+1}]$ is the input to the second MP-SSM block. Therefore, stacking the second MP-SSM block, and considering a residual connection from the first MP-SSM block, we have the following expression in the output of the second MP-SSM block:

$$\mathbf{Y}_{2(k+1)} = \mathbf{Y}_{k+1} + \mathrm{MLP}\Big(\sum_{i=0}^{k}\mathbf{A}^i\mathbf{Y}_{k+1}\mathbf{B}_2\mathbf{W}_2^i\Big), \tag{32}$$

where $\mathbf{B}_2, \mathbf{W}_2$, are the shared weights of the second MP-SSM block. In general, in a deep MP-SSM architecture of $s$ blocks, we have the following expression in the output of the $s$-th MP-SSM block:

$$\mathbf{Y}_{s(k+1)} = \mathbf{Y}_{(s-1)(k+1)} + \mathrm{MLP}\Big(\sum_{i=0}^{k}\mathbf{A}^i\mathbf{Y}_{(s-1)(k+1)}\mathbf{B}_s\mathbf{W}_s^i\Big). \tag{33}$$

To reveal the multi-hop view, we denote $\hat{\mathbf{Y}}^{(s)} = \mathbf{Y}_{s(k+1)}$, $\hat{\mathbf{W}}_i^{(s)} = \mathbf{B}_s\mathbf{W}_s^i$, and describe the deep MP-SSM architecture at the granularity of its blocks, as follows:

$$\hat{\mathbf{Y}}^{(s)} = \hat{\mathbf{Y}}^{(s-1)} + \mathrm{MLP}\Big(\sum_{i=0}^{k}\mathbf{A}^i\hat{\mathbf{Y}}^{(s-1)}\hat{\mathbf{W}}_i^{(s)}\Big). \tag{34}$$

This multi-hop interpretation of a deep MP-SSM architecture resembles the DRew-GCN architecture (Gutteridge et al., 2023), a multi-hop MPNN employing a dynamically rewired message passing strategy with delay. In fact, the recurrent equation of DRew-GCN, rephrased in our MP-SSM notation for ease of comparison, is defined as:

$$\mathbf{Y}^{(s+1)} = \mathbf{Y}^{(s)} + \sigma\left(\sum_{i=1}^{s+1}\mathbf{A}(i)\mathbf{Y}^{(s-\tau_\nu(i))}\mathbf{W}_i^{(s)}\right), \tag{35}$$

where $\mathbf{A}(i)$ is the degree-normalised shift operator that considers all the neighbors at an *exact* $i$ hops from each respective root node, $\mathbf{W}_i^{(s)}$ are weight matrices, and $\tau_\nu(i)$ is a positive integer (the *delay*) defining the temporal window for the aggregation of past embeddings. Comparing Eq. (34) and Eq. (35) we can summarize the following differences:

- DRew aggregates information using $\mathbf{A}(i)$, a function of the GSO that counts neighbors at an *exact* $i$ hops distance, while MP-SSM considers the powers of the GSO, $\mathbf{A}^i$, thus accounting for all the possible walks of length $i$. Similarly, the learnable weights in MP-SSM reflect the architectural bias induced by the recurrence, as they are structured through powers of a base matrix, specifically following the form $\hat{\mathbf{W}}_i^{(s)} = \mathbf{B}_s\mathbf{W}_s^i$.
- DRew nonlinearly aggregates information via a pointwise nonlinearity $\sigma$, while MP-SSM employs a more expressive 2-layers MLP.
- MP-SSM uses the same features for multi-hop aggregation (corresponding to $\tau_\nu(i) \equiv 0$), whereas DRew aggregates features from previous layers with a delay $\tau_\nu(i) = \max(0, i - \nu)$, effectively introducing a temporal rewiring of the graph.

Although the unfolding of MP-SSM yields expressions involving powers of the GSO, this resemblance to multi-hop architectures such as DRew (Gutteridge et al., 2023) is purely superficial. Unlike models

that aggregate information from distant nodes within a single layer, MP-SSM performs strictly 1-hop message passing at each iteration. The higher-order GSO terms emerge naturally from the recurrence, not from an architectural bias toward multi-hop aggregation. This formulation, grounded in first principles, preserves the original graph topology and constitutes a structurally distinct approach. We provide in Table 7 a comparison of DRew-GCN (results taken from Gutteridge et al. (2023)) with our MP-SSM on the Peptides-func and Peptides-struct from the LRGB task (Dwivedi et al., 2022b). Notably, MP-SSM outperforms DRew-GCN on the Peptides-struct task, suggesting that the structural architectural bias introduced by the recurrence, combined with MLP adaptivity, offers a stronger advantage than aggregating information via rewired connections from delayed past features. In contrast, on the Peptides-func task, the performance of the two models falls within each other's standard deviation, indicating no statistically significant difference between DRew-GCN, despite its dynamic rewiring strategy with delay, and MP-SSM. In Appendix H we report an extended evaluation on the LRGB benchmark.

Table 7: Results for Peptides-func and Peptides-struct averaged over 3 training seeds. DRew-GCN results are taken from Gutteridge et al. (2023). The **best** scores are in bold.

| Model | Peptides-func AP ↑ | Peptides-struct MAE ↓ |
|---|---|---|
| DRew-GCN | $\mathbf{69.96}_{\pm 0.76}$ | $0.2781_{\pm 0.0028}$ |
| MP-SSM (ours) | $69.93_{\pm 0.52}$ | $\mathbf{0.2458}_{\pm 0.0017}$ |

# G EXPERIMENTAL DETAILS

## G.1 EMPLOYED BASELINES

In our experiments, the performance of our method is compared with various state-of-the-art GNN baselines from the literature. Specifically, we consider:

- classical MPNN-based methods, i.e., GCN (Kipf & Welling, 2016), GraphSAGE (Hamilton et al., 2017), GAT (Veličković et al., 2018), GatedGCN (Bresson & Laurent, 2018), GIN (Xu et al., 2019), ARMA (Bianchi et al., 2021), GINE (Hu et al., 2020), GCNII (Chen et al., 2020), and CoGNN (Finkelshtein et al., 2024);

- heterophily-specific models, i.e., H2GCN (Zhu et al., 2020), CPGNN (Zhu et al., 2021), FAGCN (Bo et al., 2021), GPR-GNN (Chien et al., 2021), FSGNN (Maurya et al., 2022), GloGNN (Li et al., 2022), GBK-GNN (Du et al., 2022), and JacobiConv (Wang & Zhang, 2022);

- physics-inspired MPNNs, i.e., DGC (Wang et al., 2021), GRAND (Chamberlain et al., 2021), GraphCON (Rusch et al., 2022), A-DGN (Gravina et al., 2023), GREAD (Choi et al., 2023), CDE (Zhao et al., 2023), and TDE-GNN (Eliasof et al., 2024);

- Graph Transformers, i.e., Transformer (Vaswani et al., 2017a; Dwivedi & Bresson, 2021), GT (Shi et al., 2021), SAN (Kreuzer et al., 2021a), GPS (Rampášek et al., 2022), GOAT (Kong et al., 2023), Exphormer (Shirzad et al., 2023), NAGphormer (Chen et al., 2023), GRIT (Ma et al., 2023), and GraphViT (He et al., 2023);

- Higher-Order DGNs, i.e., DIGL (Gasteiger et al., 2019), MixHop (Abu-El-Haija et al., 2019), DRew (Gutteridge et al., 2023), and GRED (Ding et al., 2024).

- SSM-based GNN, i.e., Graph-Mamba (Wang et al., 2024a), GMN (Behrouz & Hashemi, 2024), GPS+Mamba (Behrouz & Hashemi, 2024), GGRNN (Ruiz et al., 2020), and GraphSSM (Li et al., 2024).

- Graph-agnostic temporal predictors, i.e., Historical Average (AV), SVR (Smola & Schölkopf, 2004), and FC-LSTM (Sutskever et al., 2014), and VAR (Lu et al., 2016);

- Spatio-temporal GNNs, i.e., DCRNN (Li et al., 2018), GConvGRU (Seo et al., 2018), Graph WaveNet (Wu et al., 2019b), ASTGCN (Guo et al., 2019), STSGCN (Song et al., 2020), GMAN (Zheng et al., 2020), MTGNN (Wu et al., 2020b), AGCRN (Bai et al., 2020), T-GCN (Zhao et al., 2020), DyGrAE (Taheri & Berger-Wolf, 2020), EGCN-O (Pareja

et al., 2020), A3T-GCN (Bai et al., 2021), MPNN LSTM (Panagopoulos et al., 2021), GTS (Shang et al., 2021), STEP (Shao et al., 2022), GC-LSTM (Chen et al., 2022), Dyn-GESN (Micheli & Tortorella, 2022), HMM4G (Errica et al., 2023), STAEformer (Liu et al., 2023), RGDAN (Fan et al., 2024), AdpSTGCN (Zhang et al., 2024), and STD-MAE (Gao et al., 2024).

## G.2 DATASETS STATISTICS

In our experiments, we compute the performance of our MP-SSM on widely used benchmarks for both static and temporal graphs. Specifically, we consider:

- long-range propagation tasks, i.e., the three graph property prediction tasks proposed by Gravina et al. (2023) ("Diameter", "SSSP", and "Eccentricity") and the "Peptide-func" and "Peptide-struct" tasks from the long-range graph benchmark (Dwivedi et al., 2022b);
- heterophilic tasks, i.e., "Roman-empire", "Amazon-ratings", "Minesweeper", "Tolokers", and "Questions" (Platonov et al., 2023);
- temporal tasks, i.e., "Metr-LA" and "PeMS-Bay" for traffic forecasting (Li et al., 2018), and the "Chickenpox Hungary", "PedalMe London", and "Wikipedia math" forecasting tasks introduced by Rozemberczki et al. (2021).

In Table 8, we report the statistics of the employed datasets.

Table 8: Dataset statistics

|  | Task | Nodes | Edges | Graphs (or Timesteps) | Frequency |
|---|---|---|---|---|---|
| Static | Diameter | 25 - 35 | 22 - 553 | 7,040 | – |
|  | SSSP | 25 - 35 | 22 - 553 | 7,040 | – |
|  | Eccentricity | 25 - 35 | 22 - 553 | 7,040 | – |
|  | Peptide-func | 150.94 (avg) | 307.30 (avg) | 15,535 | – |
|  | Peptide-struct | 150.94 (avg) | 307.30 (avg) | 15,535 | – |
|  | Roman-empire | 22,662 | 32,927 | 1 | – |
|  | Amazon-ratings | 24,492 | 93,050 | 1 | – |
|  | Minesweeper | 10,000 | 39,402 | 1 | – |
|  | Tolokers | 11,758 | 519,000 | 1 | – |
|  | Questions | 48,921 | 153,540 | 1 | – |
| Temporal | Metr-LA | 207 | 1,515 | 34,272 | 5 mins |
|  | PeMS-Bay | 325 | 2,369 | 52,116 | 5 mins |
|  | Chickenpox Hungary | 20 | 102 | 512 | Weekly |
|  | PedalMe London | 15 | 225 | 15 | Weekly |
|  | Wikipedia math | 731 | 27,079 | 1,068 | Daily |

## G.3 HYPERPARAMETER SPACE

In Table 9, we report the grid of hyperparameters employed in our experiments by our method on all the considered benchmarks.

## H RESULTS ON THE LONG-RANGE GRAPH BENCHMARK

To further evaluate the performance of our MP-SSM, we consider two tasks of the Long-Range Graph Benchmark (LRGB) (Dwivedi et al., 2022b).

**Setup.** We evaluate MP-SSM on the Peptides-func and Peptides-struct tasks from the LRGB benchmark, which involve predicting functional and structural properties of peptides that require modeling long-range dependencies. We follow the original experimental setup and 500k parameter budget.

**Results.** As shown in Table 10, MP-SSM outperforms standard MPNNs, transformer-based GNNs, and most multi-hop and SSM-based models. It achieves the highest average ranking across tasks

Table 9: The grid of hyperparameters employed during model selection for the graph property prediction tasks (*GPP*), Long Range Graph Benchmark (*LRGB*), heterophilic benchmarks (*Hetero*), and spatio-temporal benchmarks (*Temporal*).

| Hyperparameters | Values | | | |
|---|---|---|---|---|
| | *GPP* | *LRGB* | *Hetero* | *Temporal* |
| Optimizer | Adam | AdamW | AdamW | AdamW |
| Learning rate | 0.003 | 0.001, 0.0005, 0.0001 | 0.001, 0.0005 ,0.0001 | 0.005, 0.001, 0.0005 ,0.0001 |
| Weight decay | $10^{-6}$ | 0, 0.0001, 0.001 | 0, 0.0001, 0.001 | 0, 0.0001, 0.001 |
| Dropout | 0 | 0, 0.5 | 0, 0.4, 0.5, 0.6, | 0, 0.5 |
| N. recurrences | 1, 5, 10, 20 | 1, 2, 4, 8, 16 | 1, 2, 4, 8, 16 | 1, 2, 4, 8, 16 |
| Embedding dim | 10, 20, 30 | 32,64,128,256 | 32,64,128,256 | 32,64,128,256 |
| N. Blocks | 1, 2 | 1, 2, 4, 8, 16 | 1, 2, 4, 8, 16 | 1, 2, 4, 8, 16 |
| Structure of $\mathbf{U}$ | | $\mathbf{U} = [\mathbf{U}_1, \dots, \mathbf{U}_1]$ | | $\mathbf{U} = [\mathbf{U}_1, \mathbf{U}_2, \dots]$ |

without relying on global attention or graph rewiring. Compared to other graph SSMs, MP-SSM delivers strong performance while preserving permutation-equivariance.

Table 10: Results for Peptides-func and Peptides-struct averaged over 3 training seeds. Re-evaluated methods employ the 3-layer MLP readout proposed in Tönshoff et al. (2023). Note that all MPNN-based methods include structural and positional encoding. The **first**, **second**, and **third** best scores are colored. Baseline results are reported from Dwivedi et al. (2022b); Gutteridge et al. (2023); Tönshoff et al. (2023); He et al. (2023); Ding et al. (2024); Gravina et al. (2025). ‡ means 3-layer MLP readout and residual connections are employed.

| Model | Peptides-func AP ↑ | Peptides-struct MAE ↓ | avg. Rank ↓ |
|---|---|---|---|
| **MPNNs** | | | |
| A-DGN | $59.75_{\pm 0.44}$ | $0.2874_{\pm 0.0021}$ | 26.0 |
| GatedGCN | $58.64_{\pm 0.77}$ | $0.3420_{\pm 0.0013}$ | 29.0 |
| GCN | $59.30_{\pm 0.23}$ | $0.3496_{\pm 0.0013}$ | 29.5 |
| GCNII | $55.43_{\pm 0.78}$ | $0.3471_{\pm 0.0010}$ | 30.5 |
| GINE | $54.98_{\pm 0.79}$ | $0.3547_{\pm 0.0045}$ | 32.0 |
| GRAND | $57.89_{\pm 0.62}$ | $0.3418_{\pm 0.0015}$ | 29.0 |
| GraphCON | $60.22_{\pm 0.68}$ | $0.2778_{\pm 0.0018}$ | 24.0 |
| SWAN | $67.51_{\pm 0.39}$ | $0.2485_{\pm 0.0009}$ | 12.5 |
| **Multi-hop GNNs** | | | |
| DIGL+MPNN | $64.69_{\pm 0.19}$ | $0.3173_{\pm 0.0007}$ | 25.0 |
| DIGL+MPNN+LapPE | $68.30_{\pm 0.26}$ | $0.2616_{\pm 0.0018}$ | 16.5 |
| DRew-GatedGCN | $67.33_{\pm 0.94}$ | $0.2699_{\pm 0.0018}$ | 19.5 |
| DRew-GatedGCN+LapPE | $69.77_{\pm 0.26}$ | $0.2539_{\pm 0.0007}$ | 12.0 |
| DRew-GCN | $69.96_{\pm 0.76}$ | $0.2781_{\pm 0.0028}$ | 14.0 |
| DRew-GCN+LapPE | $71.50_{\pm 0.44}$ | $0.2536_{\pm 0.0015}$ | 8.0 |
| DRew-GIN | $69.40_{\pm 0.74}$ | $0.2799_{\pm 0.0016}$ | 17.5 |
| DRew-GIN+LapPE | $71.26_{\pm 0.45}$ | $0.2606_{\pm 0.0014}$ | 9.5 |
| GRED | $70.85_{\pm 0.27}$ | $0.2503_{\pm 0.0019}$ | 7.0 |
| MixHop-GCN | $65.92_{\pm 0.36}$ | $0.2921_{\pm 0.0023}$ | 23.0 |
| MixHop-GCN+LapPE | $68.43_{\pm 0.49}$ | $0.2614_{\pm 0.0023}$ | 15.5 |
| **Transformers** | | | |
| GraphGPS+LapPE | $65.35_{\pm 0.41}$ | $0.2500_{\pm 0.0005}$ | 15.5 |
| Graph ViT | $69.42_{\pm 0.75}$ | $0.2449_{\pm 0.0016}$ | 5.5 |
| GRIT | $69.88_{\pm 0.82}$ | $0.2460_{\pm 0.0012}$ | 5.0 |
| Transformer+LapPE | $63.26_{\pm 1.26}$ | $0.2529_{\pm 0.0016}$ | 19.5 |
| SAN+LapPE | $63.84_{\pm 1.21}$ | $0.2683_{\pm 0.0043}$ | 22.0 |
| **Modified and Re-evaluated‡** | | | |
| DRew-GCN+LapPE | $69.45_{\pm 0.21}$ | $0.2517_{\pm 0.0011}$ | 11.0 |
| GatedGCN | $67.65_{\pm 0.47}$ | $0.2477_{\pm 0.0009}$ | 11.0 |
| GCN | $68.60_{\pm 0.50}$ | $0.2460_{\pm 0.0007}$ | 7.5 |
| GINE | $66.21_{\pm 0.67}$ | $0.2473_{\pm 0.0017}$ | 12.0 |
| GraphGPS+LapPE | $65.34_{\pm 0.91}$ | $0.2509_{\pm 0.0014}$ | 17.0 |
| **Graph SSMs** | | | |
| GMN | $70.71_{\pm 0.83}$ | $0.2473_{\pm 0.0025}$ | 4.5 |
| Graph-Mamba | $67.39_{\pm 0.87}$ | $0.2478_{\pm 0.0016}$ | 12.5 |
| **Ours** | | | |
| MP-SSM | $69.93_{\pm 0.52}$ | $0.2458_{\pm 0.0017}$ | 4.0 |

**Ablations.** As discussed in Section 2, the MLP in Equation (4) is implemented as a standard MLP with 2 linear layers and a nonlinearity in between. To better understand the role of the nonlinearity and MLP's depth, we ablate in Table 11 on the performance of ReLU, GELU, and ELU functions, and in Table 12 we ablate over MLP depths of 1, 2, and 3. Our results show that the performance of the tested nonlinearities are statistically similar, while that a two-layer MLP is a good balance between computational demand and performance.

Table 11: Results for Peptides-func and Peptides-struct averaged over 3 training seeds for three different nonlinearities in the MLP of Equation (4), i.e., ReLU, GELU, and ELU.

| Model | Peptides-func | Peptides-struct |
|---|---|---|
| | AP ↑ | MAE ↓ |
| MP-SSM w/ ReLU | $69.93_{\pm 0.52}$ | $0.2458_{\pm 0.0017}$ |
| MP-SSM w/ GELU | $69.88_{\pm 0.49}$ | $0.2456_{\pm 0.0018}$ |
| MP-SSM w/ ELU | $69.95_{\pm 0.60}$ | $0.2459_{\pm 0.0011}$ |

Table 12: Results for Peptides-func and Peptides-struct averaged over 3 training seeds for three different depth of the MLP in Equation (4).

| Model | Peptides-func | Peptides-struct |
|---|---|---|
| | AP ↑ | MAE ↓ |
| MP-SSM w/ 1 layer MLP | $69.12_{\pm 0.43}$ | $0.2461_{\pm 0.0009}$ |
| MP-SSM w/ 2 layers MLP | $69.93_{\pm 0.52}$ | $0.2458_{\pm 0.0017}$ |
| MP-SSM w/ 3 layers MLP | $69.91_{\pm 0.57}$ | $0.2451_{\pm 0.0014}$ |

## I  COMPLEXITY AND RUNTIMES

We discuss the theoretical complexity of our method, followed by a comparison of runtimes with other methods.

**Complexity Analysis.** Our MP-SSM consists of a stack of blocks. Each of them performs a linear recurrence of $k$ iterations followed by the application of a nonlinear map, as defined in Eqs. (3) and (4). Note that $k$ is either the length of the temporal graph sequence or a hyperparameter. Given the similarities between the linear recurrence in MP-SSM and standard MPNNs, described in Section 2, the recurrence retains the complexity of standard MPNNs. Therefore, the Eq. (3) is linear in the number of node $|V|$ and edges $|E|$, achieving a time complexity of $\mathcal{O}(k \cdot (|V| + |E|))$, with $k$ the number of iterations. Considering $\mathcal{O}(m)$ the time complexity of the MLP in Eq. (4), then the final time complexity of one MP-SSM block is $\mathcal{O}(k \cdot (|V| + |E|) + m)$ in the static case and $\mathcal{O}(k \cdot (|V| + |E| + m))$ in the temporal case.

**Runtimes.** We provide runtimes for MP-SSM and compare it with other methods, such as Graph GPS and GCN, in Table 13. In all cases, we use a model with 256 hidden dimensions and a varying depth effective by changing the number of recurrences from 2 to 16 in our MP-SSM with 2 MP-SSM blocks, and the number of layers is the depth for other methods. We report the training and inference times in milliseconds, as well as the downstream performance performance obtained on the Roman-Empire dataset. As shown in the table, MP-SSM delivers stronger performance than graph transformers at only a fraction of their computational cost, i.e., maintaining a runtime comparable to GCN, which scales linearly with the graph size. Notably, our MP-SSM achieves better performance than GCN and GPS, and maintains its performance as depth increases, different than GCN. All runtimes are measured on an NVIDIA A6000 GPU with 48GB of memory.

Table 13: Training and Inference Runtime (milliseconds) and obtained node classification accuracy (%) on the Roman-Empire dataset.

| Metrics | Method | Depth | | | |
|---|---|---|---|---|---|
| | | 4 | 8 | 16 | 32 |
| Training (ms) | | 18.38 | 33.09 | 61.86 | 120.93 |
| Inference (ms) | GCN | 9.30 | 14.64 | 27.95 | 53.55 |
| Accuracy (%) | | 73.60 | 61.52 | 56.86 | 52.42 |
| Training (ms) | | 1139.05 | 2286.96 | 4545.46 | OOM |
| Inference (ms) | GPS | 119.10 | 208.26 | 427.89 | OOM |
| Accuracy (%) | | 81.97 | 81.53 | 81.88 | OOM |
| Training (ms) | | 1179.08 | 2304.77 | 4590.26 | OOM |
| Inference (ms) | GPS$_{GAT+Performer}$ (RWSE) | 120.11 | 209.98 | 429.03 | OOM |
| Accuracy (%) | | 84.89 | 87.01 | 86.94 | OOM |
| Training (ms) | | 23.19 | 41.44 | 72.09 | 141.82 |
| Inference (ms) | MP-SSM | 10.93 | 18.87 | 38.87 | 67.59 |
| Accuracy (%) | | 85.73 | 88.02 | 90.82 | 90.91 |

## J  ABLATIONS

### J.1  IMPACT OF SSM HEURISTIC ON GRAPH REPRESENTATION LEARNING

We perform an ablation study to isolate the incremental contribution of each SSM heuristic to the performance gains in reconstructing graph-structural information that depends on learning long-range dependencies; specifically for computing quantities like the diameter of a graph, the single-source-shortest-paths (SSSP), and the eccentricity of a node, see Section 4.1 for more details on these tasks. Results of this ablation are reported in Table 14.

Table 14: Architecture ablation study. Mean test $log_{10}(MSE)$ and std averaged on 4 random weight initialization on Graph Property Prediction tasks (Section 4.1). The lower, the better. The evaluation include: a nonlinear multilayer GCN (`GCN`), a linear multilayer GCN (`Linear GCN`), a linear multilayer GCN with weight sharing (`Linear GCN (ws)`), Linear GCN (ws) followed by an MLP (`1 Block Linear GCN`), a stack of multiple 1 Block Linear GCN (`Multi-Blocks Linear GCN`), and our `MP-SSM`, which represent a multi-blocks linear GCN with standard deep learning heuristics such as residual connections and normalisation layers between blocks.

| Model | Diameter ↓ | SSSP ↓ | Eccentricity ↓ |
|---|---|---|---|
| GCN | $0.7424_{\pm0.0466}$ | $0.9499_{\pm0.0001}$ | $0.8468_{\pm0.0028}$ |
| Linear GCN | $-2.1255_{\pm0.0984}$ | $-1.5822_{\pm0.0002}$ | $-2.1424_{\pm0.0014}$ |
| Linear GCN (ws) | $-2.2678_{\pm0.1277}$ | $-1.5823_{\pm0.0001}$ | $-2.1447_{\pm0.001}$ |
| 1 Block Linear GCN | $-2.2734_{\pm0.1513}$ | $-1.5836_{\pm0.0025}$ | $-2.1869_{\pm0.0058}$ |
| Multi-Blocks Linear GCN | $-2.3531_{\pm0.3183}$ | $-1.5821_{\pm0.0001}$ | $-2.1861_{\pm0.0066}$ |
| MP-SSM | $\mathbf{-3.2353}_{\pm0.1735}$ | $\mathbf{-4.6321}_{\pm0.0779}$ | $\mathbf{-2.9724}_{\pm0.0271}$ |

We devised an ablation aimed to incrementally add/remove components starting from a plain GCN and ending with the full MP-SSM architecture. We first remove the nonlinearity from a GCN (second row in Table 14), then add weight sharing to obtain a linear recurrence (third row), then introduce a shared MLP over the recurrent steps to obtain an MP-SSM block (fourth row), next stack multiple MP-SSM blocks (fifth row), and finally add residual connections and normalization between blocks (last row).

The ablation conducted reveals that removing the nonlinearity from GCN yields a significant performance improvement. Introducing weight sharing, effectively incorporating recurrence into the linear graph diffusion process, yields a slight performance boost while considerably reducing the number of parameters. Appending an MLP at the last time step of this linear recurrent architecture does not result in statistically significant gains, except marginally for the Eccentricity task. Likewise, constructing a hierarchical block structure does not noticeably enhance performance. These limited improvements

suggest that, for the three tasks considered, the linear recurrence mechanism alone, provided a long enough recurrence, is sufficient to capture meaningful representations to reconstruct graph's structural information. Finally, incorporating standard deep learning heuristics further strengthens the full MP-SSM architecture, consistently improving performance across all tasks. These results also highlight that the components contributing most to MP-SSM's performance vary across tasks. For example, in the Diameter task, the linear propagation alone yields the largest gains, whereas in SSSP the residual connections and normalization provide the main performance boost. Overall, MP-SSM's effectiveness stems from the synergy of its core basic components: (i) linear recurrent propagation, which propagates information across the graph while avoiding the accumulation of nonlinear distortions, (ii) universal approximation power of MLPs, enabling expressive feature transformations on representations that have been progressively aggregated over many recurrent diffusion steps, and (iii) stacked deep residual blocks, allowing hierarchical representation learning while promoting stable gradients.

### J.2 ON THE INFLUENCE OF DIFFERENT GSOs ON THE GRAPH PROPERTY PREDICTION TASKS

The stability of MP-SSM depends on the magnitudes of the powers of the chosen GSO, as shown by our exact Jacobian computation (Theorem 3.4). For stable (even infinite) recursions, the powers must neither diverge nor vanish, motivating the use of the symmetrically normalized adjacency matrix with self-loops (i.e., Equation (1)) as GSO in our MP-SSM framework, as proven in Lemma 3.5. Another suitable GSO candidate is the Random Walk normalized Laplacian (i.e., $\mathbf{L} = \mathbf{I} - \mathbf{D}^{-1}\mathbf{A}$), whose powers possess similar stability guarantees. In Table 15, we have added an experiment on the Graph Property Prediction benchmark (see Section 4.1) task comparing performance using the Random Walk (RW) and the unnormalized Laplacian (i.e., $\mathbf{L} = \mathbf{D} - \mathbf{A}$) GSOs. As can be seen, using the Random Walk GSO leads to performance comparable to our GSO in Equation (1), as both possess similar stability guarantees. In contrast, the unnormalized Laplacian is the least suitable choice. The reason is that the powers of this operator can grow rapidly, and since these powers appear in the gradients (see the Jacobian expression in Equation (6)), such amplification can induce significant training instabilities and ultimately degrade performance. This observation is empirically confirmed in Figure 4: with identical model configurations, the unnormalized Laplacian causes the Jacobian norm to grow exponentially after roughly 10 steps.

Table 15: Mean test set $log_{10}(\text{MSE})(\downarrow)$ and std averaged on 4 random weight initializations on Graph Property Prediction tasks for different GSOs in MP-SSM. We consider three GSOs: symmetrically normalized adjacency matrix with self-loops (*Eq.* (1)); the Random Walk normalized Laplacian (*RW*), $\mathbf{L} = \mathbf{I} - \mathbf{D}^{-1}\mathbf{A}$; and the unnormalized Laplacian (*L*), $\mathbf{L} = \mathbf{D} - \mathbf{A}$. The lower, the better.

| Model | Diameter $\downarrow$ | SSSP $\downarrow$ | Eccentricity $\downarrow$ |
|---|---|---|---|
| MP-SSM (*Eq.* (1)) | $-3.2353_{\pm 0.1735}$ | $-4.6321_{\pm 0.0779}$ | $-2.9724_{\pm 0.0271}$ |
| MP-SSM (*RW*) | $-3.2445_{\pm 0.0481}$ | $-4.3860_{\pm 0.0379}$ | $-3.1326_{\pm 0.1040}$ |
| MP-SSM (*L*) | $-2.3509_{\pm 0.0192}$ | $-3.9729_{\pm 0.8539}$ | $-2.2353_{\pm 0.0138}$ |

## K EXTENDED COMPARISON ON THE HETEROPHILIC BENCHMARK

To further evaluate the performance of MP-SSM, we report a more complete comparison for the heterophilic task in Table 16. Specifically, we include more MPNN-based models, graph transformers, and heterophily-designated GNNs.

In Table 16, we color the top three methods. Different from the main body of the paper, here we also include sub-variants of methods in the highlighted results, providing an additional perspective on the findings. Notably, our MP-SSM achieves the best average ranking across all datasets in the heterophilic benchmarks. We believe that MP-SSM perform strongly on these tasks because of two main reasons: (i) it is well-suited to capture long-range dependencies, and (ii) it can effectively represent low- and high-frequency components. Specifically, as discussed in Platonov et al. (2023), these tasks likely involve long-range dependencies due to their graph structure and dimensionality, and MP-SSM is well-suited to capture such dependencies (as discussed in Section 3), giving it an advantage. Second, unlike standard MPNNs that rely on repeated local nonlinear aggregation,

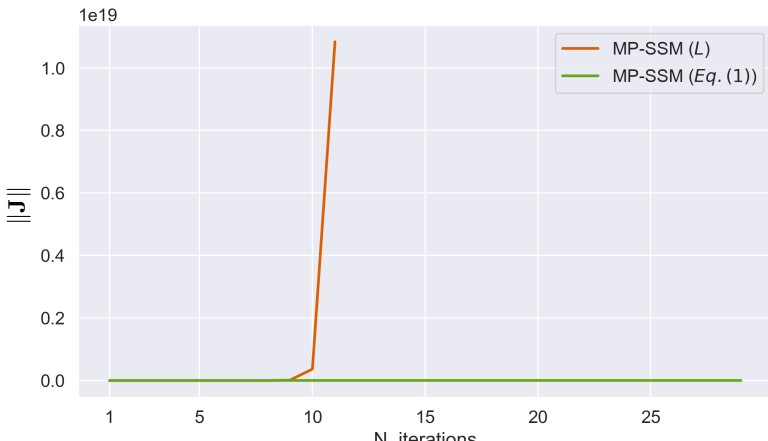

Figure 4: The norm of the Jacobian of a 1 block MP-SSM for different GSOs, measured on the Diameter task (see Section 4.1). We consider two GSOs: symmetrically normalized adjacency matrix with self-loops (*Eq.* (1)), and the unnormalized Laplacian (*L*), $\mathbf{L} = \mathbf{D} - \mathbf{A}$.

MP-SSM uses a linear recurrence. When unfolded, this corresponds to a weighted sum over powers of the graph shift operator (see Equation (11)). Crucially, these weights are learnable and can be negative, allowing the model not only to accumulate signals but also to cancel or invert them. Such behavior is known to be useful for heterophilic settings (Chien et al., 2021; Eliasof et al., 2023). This enables MP-SSM to represent both low- and high-frequency components, effectively learning flexible, potentially high-pass aggregation schemes. We hypothesize that this spectral flexibility, combined with the nonlinear MLP, allows MP-SSM to capture heterophilic patterns effectively.

## L  MP-SSM AND TIME-VARYING TOPOLOGIES

To further evaluate the spatiotemporal performance of our MP-SSM, we consider the Twitter Tennis RG benchmark Rozemberczki et al. (2021), where both node features and edges change over time. Specifically, Twitter Tennis RG is a mention graph in which nodes are Twitter accounts and their labels encode the number of mentions between them. We follow the original experimental setup of Rozemberczki et al. (2021).
As shown in Table 17, MP-SSM outperforms all the baselines, demonstrating its effectiveness also with evolving graph topologies.

## M  ON THE SIMILARITY WITH POLYNOMIAL FILTERS

Although there are similarities with polynomial filters, such as ChebNet (Defferrard et al., 2016) and SGC (Wu et al., 2019a), since MP-SSM gives rise to a polynomial-like expansion when unfolding the recurrence (i.e., Equation (11)), this similarity is only structural. Indeed, ChebNet employs the Chebyshev polynomial, which suffer from instability issues with high-order Chebyshev filters (Hariri et al., 2025). Differently, the polynomial-like behavior of MP-SSM emerges naturally from iterating a 1-hop linear recurrence and its dynamics remains stable even at large recurrent depths, as proven in Lemma 3.5 and Theorem 3.6. Moreover, the recurrence in the MP-SSM block unfolds into a learnable combination of all powers of the GSO, thereby offering a significantly richer propagation scheme than the fixed $k$-hop aggregation used by SGC. To further illustrate these differences empirically, we report in Table 18 the performance on the Graph Property Prediction benchmark (see Section 4.1) comparing our MP-SSM with ChebNet and SGC. We note that the performance gap is large across all tasks, despite the superficial similarity in polynomial structure, highlighting that our method behaves fundamentally differently in practice and better capture long-range dependencies between nodes. Therefore, while one may observe polynomial expressions in an unrolled recurrence, the architectural

Table 16: Mean test set score and std averaged over 4 random weight initializations on heterophilic datasets. The higher, the better. **First**, **second**, and **third** best results for each task are color-coded. Baseline results are reported from Finkelshtein et al. (2024); Behrouz & Hashemi (2024); Platonov et al. (2023); Müller et al. (2024); Luan et al. (2024). "∗" in the rank column means that the average has been computed over less trials.

| Model | Roman-empire Acc ↑ | Amazon-ratings Acc ↑ | Minesweeper AUC ↑ | Tolokers AUC ↑ | Questions AUC ↑ | avg. Rank ↓ |
|---|---|---|---|---|---|---|
| **Luan et al. (2024)** | | | | | | |
| MLP-1 | $64.12_{\pm 0.61}$ | $38.60_{\pm 0.41}$ | $50.59_{\pm 0.83}$ | $71.89_{\pm 0.82}$ | $70.33_{\pm 0.96}$ | 41.0 |
| MLP-2 | $66.04_{\pm 0.71}$ | $49.55_{\pm 0.81}$ | $50.92_{\pm 1.25}$ | $74.58_{\pm 0.75}$ | $69.97_{\pm 1.16}$ | 34.4 |
| SGC-1 | $44.60_{\pm 0.52}$ | $40.69_{\pm 0.42}$ | $82.04_{\pm 0.77}$ | $73.80_{\pm 1.35}$ | $71.06_{\pm 0.92}$ | 38.6 |
| **Graph-agnostic** | | | | | | |
| ResNet | $65.88_{\pm 0.38}$ | $45.90_{\pm 0.52}$ | $50.89_{\pm 1.39}$ | $72.95_{\pm 1.06}$ | $70.34_{\pm 0.76}$ | 37.4 |
| ResNet+adj | $52.25_{\pm 0.40}$ | $51.83_{\pm 0.57}$ | $50.42_{\pm 0.83}$ | $78.78_{\pm 1.11}$ | $75.77_{\pm 1.24}$ | 32.0 |
| ResNet+SGC | $73.90_{\pm 0.51}$ | $50.66_{\pm 0.48}$ | $70.88_{\pm 0.90}$ | $80.70_{\pm 0.97}$ | $75.81_{\pm 0.96}$ | 29.0 |
| **MPNNs** | | | | | | |
| CO-GNN($\Sigma, \Sigma$) | $91.57_{\pm 0.32}$ | $51.28_{\pm 0.56}$ | $95.09_{\pm 1.18}$ | $83.36_{\pm 0.89}$ | $80.02_{\pm 0.86}$ | **8.0** |
| CO-GNN($\mu, \mu$) | $91.37_{\pm 0.35}$ | $54.17_{\pm 0.37}$ | $97.31_{\pm 0.41}$ | $84.45_{\pm 1.17}$ | $76.54_{\pm 0.95}$ | **6.8** |
| GAT | $80.87_{\pm 0.30}$ | $49.09_{\pm 0.63}$ | $92.01_{\pm 0.68}$ | $83.70_{\pm 0.47}$ | $77.43_{\pm 1.20}$ | 18.0 |
| GAT-sep | $88.75_{\pm 0.41}$ | $52.70_{\pm 0.62}$ | $93.91_{\pm 0.35}$ | $83.78_{\pm 0.43}$ | $76.79_{\pm 0.71}$ | 9.8 |
| GAT (LapPE) | $84.80_{\pm 0.46}$ | $44.90_{\pm 0.73}$ | $93.50_{\pm 0.54}$ | $84.99_{\pm 0.54}$ | $76.55_{\pm 0.84}$ | 16.0 |
| GAT (RWSE) | $86.62_{\pm 0.53}$ | $48.58_{\pm 0.41}$ | $92.53_{\pm 0.65}$ | $85.02_{\pm 0.67}$ | $77.83_{\pm 1.22}$ | 11.6 |
| GAT (DEG) | $85.51_{\pm 0.56}$ | $51.65_{\pm 0.60}$ | $93.04_{\pm 0.62}$ | $84.22_{\pm 0.81}$ | $77.10_{\pm 1.23}$ | 12.6 |
| Gated-GCN | $74.46_{\pm 0.54}$ | $43.00_{\pm 0.32}$ | $87.54_{\pm 1.22}$ | $77.31_{\pm 1.14}$ | $76.61_{\pm 1.13}$ | 31.4 |
| GCN | $73.69_{\pm 0.74}$ | $48.70_{\pm 0.63}$ | $89.75_{\pm 0.52}$ | $83.64_{\pm 0.67}$ | $76.09_{\pm 1.27}$ | 25.8 |
| GCN (LapPE) | $83.37_{\pm 0.55}$ | $44.35_{\pm 0.36}$ | $94.26_{\pm 0.49}$ | $84.95_{\pm 0.78}$ | $77.79_{\pm 1.34}$ | 14.6 |
| GCN (RWSE) | $84.84_{\pm 0.55}$ | $46.40_{\pm 0.55}$ | $93.84_{\pm 0.48}$ | $85.11_{\pm 0.77}$ | $77.81_{\pm 1.40}$ | 12.0 |
| GCN (DEG) | $84.21_{\pm 0.47}$ | $50.01_{\pm 0.69}$ | $94.14_{\pm 0.50}$ | $82.51_{\pm 0.83}$ | $76.96_{\pm 1.21}$ | 16.4 |
| SAGE | $85.74_{\pm 0.67}$ | $53.63_{\pm 0.39}$ | $93.51_{\pm 0.57}$ | $82.43_{\pm 0.44}$ | $76.44_{\pm 0.62}$ | 15.6 |
| **Graph Transformers** | | | | | | |
| Exphormer | $89.03_{\pm 0.37}$ | $53.51_{\pm 0.46}$ | $90.74_{\pm 0.53}$ | $83.77_{\pm 0.78}$ | $73.94_{\pm 1.06}$ | 16.6 |
| NAGphormer | $74.34_{\pm 0.77}$ | $51.26_{\pm 0.72}$ | $84.19_{\pm 0.66}$ | $78.32_{\pm 0.95}$ | $68.17_{\pm 1.53}$ | 30.6 |
| GOAT | $71.59_{\pm 1.25}$ | $44.61_{\pm 0.50}$ | $81.09_{\pm 1.02}$ | $83.11_{\pm 1.04}$ | $75.76_{\pm 1.66}$ | 31.2 |
| GPS | $82.00_{\pm 0.61}$ | $53.10_{\pm 0.42}$ | $90.63_{\pm 0.67}$ | $83.71_{\pm 0.48}$ | $71.73_{\pm 1.47}$ | 21.4 |
| GPS$_{GCN+Performer}$ (LapPE) | $83.96_{\pm 0.53}$ | $48.20_{\pm 0.67}$ | $93.85_{\pm 0.41}$ | $84.72_{\pm 0.77}$ | $77.85_{\pm 1.25}$ | 12.8 |
| GPS$_{GCN+Performer}$ (RWSE) | $84.72_{\pm 0.65}$ | $48.08_{\pm 0.85}$ | $92.88_{\pm 0.50}$ | $84.81_{\pm 0.86}$ | $76.45_{\pm 1.51}$ | 16.6 |
| GPS$_{GCN+Performer}$ (DEG) | $83.38_{\pm 0.68}$ | $48.93_{\pm 0.47}$ | $93.60_{\pm 0.47}$ | $80.49_{\pm 0.97}$ | $74.24_{\pm 1.18}$ | 22.6 |
| GPS$_{GAT+Performer}$ (LapPE) | $85.93_{\pm 0.52}$ | $48.86_{\pm 0.38}$ | $92.62_{\pm 0.79}$ | $84.62_{\pm 0.54}$ | $76.71_{\pm 0.98}$ | 14.4 |
| GPS$_{GAT+Performer}$ (RWSE) | $87.04_{\pm 0.58}$ | $49.92_{\pm 0.68}$ | $91.08_{\pm 0.58}$ | $84.38_{\pm 0.91}$ | $77.14_{\pm 1.49}$ | 15.0 |
| GPS$_{GAT+Performer}$ (DEG) | $85.54_{\pm 0.58}$ | $51.03_{\pm 0.60}$ | $91.52_{\pm 0.46}$ | $82.45_{\pm 0.89}$ | $76.51_{\pm 1.19}$ | 20.0 |
| GPS$_{GCN+Transformer}$ (LapPE) | OOM | OOM | $91.82_{\pm 0.41}$ | $83.51_{\pm 0.93}$ | OOM | 33.8 |
| GPS$_{GCN+Transformer}$ (RWSE) | OOM | OOM | $91.17_{\pm 0.51}$ | $83.53_{\pm 1.06}$ | OOM | 34.4 |
| GPS$_{GCN+Transformer}$ (DEG) | OOM | OOM | $91.76_{\pm 0.61}$ | $80.82_{\pm 0.95}$ | OOM | 36.2 |
| GPS$_{GAT+Transformer}$ (LapPE) | OOM | OOM | $92.29_{\pm 0.61}$ | $84.70_{\pm 0.56}$ | OOM | 30.2 |
| GPS$_{GAT+Transformer}$ (RWSE) | OOM | OOM | $90.82_{\pm 0.56}$ | $84.01_{\pm 0.96}$ | OOM | 33.8 |
| GPS$_{GAT+Transformer}$ (DEG) | OOM | OOM | $91.58_{\pm 0.56}$ | $81.89_{\pm 0.85}$ | OOM | 36.0 |
| GT | $86.51_{\pm 0.73}$ | $51.17_{\pm 0.66}$ | $91.85_{\pm 0.76}$ | $83.23_{\pm 0.64}$ | $77.95_{\pm 0.68}$ | 14.4 |
| GT-sep | $87.32_{\pm 0.39}$ | $52.18_{\pm 0.80}$ | $92.29_{\pm 0.47}$ | $82.52_{\pm 0.92}$ | $78.05_{\pm 0.93}$ | 12.6 |
| **Heterophily-Designated GNNs** | | | | | | |
| CPGNN | $63.96_{\pm 0.62}$ | $39.79_{\pm 0.77}$ | $52.03_{\pm 5.46}$ | $73.36_{\pm 1.01}$ | $65.96_{\pm 1.95}$ | 40.0 |
| FAGCN | $65.22_{\pm 0.56}$ | $44.12_{\pm 0.30}$ | $88.17_{\pm 0.73}$ | $77.75_{\pm 1.05}$ | $77.24_{\pm 1.26}$ | 31.0 |
| FSGNN | $79.92_{\pm 0.56}$ | $52.74_{\pm 0.83}$ | $90.08_{\pm 0.70}$ | $82.76_{\pm 0.61}$ | $78.86_{\pm 0.92}$ | 18.2 |
| GBK-GNN | $74.57_{\pm 0.47}$ | $45.98_{\pm 0.71}$ | $90.85_{\pm 0.58}$ | $81.01_{\pm 0.67}$ | $74.47_{\pm 0.86}$ | 28.0 |
| GloGNN | $59.63_{\pm 0.69}$ | $36.89_{\pm 0.14}$ | $51.08_{\pm 1.23}$ | $73.39_{\pm 1.17}$ | $65.74_{\pm 1.19}$ | 41.0 |
| GPR-GNN | $64.85_{\pm 0.27}$ | $44.88_{\pm 0.34}$ | $86.24_{\pm 0.61}$ | $72.94_{\pm 0.97}$ | $55.48_{\pm 0.91}$ | 38.4 |
| H2GCN | $60.11_{\pm 0.52}$ | $36.47_{\pm 0.23}$ | $89.71_{\pm 0.31}$ | $73.35_{\pm 1.01}$ | $63.59_{\pm 1.46}$ | 39.6 |
| JacobiConv | $71.14_{\pm 0.42}$ | $43.55_{\pm 0.48}$ | $89.66_{\pm 0.40}$ | $68.66_{\pm 0.65}$ | $73.88_{\pm 1.16}$ | 36.2 |
| **Graph SSMs** | | | | | | |
| GMN | $87.69_{\pm 0.50}$ | $54.07_{\pm 0.31}$ | $91.01_{\pm 0.23}$ | $84.52_{\pm 0.21}$ | – | 11.0∗ |
| GPS + Mamba | $83.10_{\pm 0.28}$ | $45.13_{\pm 0.97}$ | $89.93_{\pm 0.54}$ | $83.70_{\pm 1.05}$ | – | 25.5∗ |
| **Ours** | | | | | | |
| MP-SSM | $90.91_{\pm 0.48}$ | $53.65_{\pm 0.71}$ | $95.33_{\pm 0.72}$ | $85.26_{\pm 0.93}$ | $78.18_{\pm 1.34}$ | **2.4** |

motivation, stability properties, and empirical behavior of MP-SSM differ sharply from classical polynomial-filter GNNs.

Table 17: Mean test MSE and std averaged over 10 experimental repetitions on Twitter Tennis RG Benchmark. Baseline results are reported from (Rozemberczki et al., 2021).

| DCRNN | GConvLSTM | DyGrAE | EGCN-H | T-GCN | AGCRN | MP-SSM (Ours) |
|---|---|---|---|---|---|---|
| $2.049_{\pm 0.023}$ | $2.049_{\pm 0.024}$ | $2.031_{\pm 0.006}$ | $2.040_{\pm 0.018}$ | $2.045_{\pm 0.027}$ | $2.039_{\pm 0.022}$ | $\mathbf{2.028}_{\pm 0.015}$ |

Table 18: Mean test set $log_{10}(\text{MSE})(\downarrow)$ and std averaged on 4 random weight initializations on Graph Property Prediction tasks. The lower, the better. ChebNet's results are reported from Hariri et al. (2025).

| Model | Diameter | SSSP | Eccentricity |
|---|---|---|---|
| ChebNet | $-0.1517_{\pm 0.0343}$ | $-1.8519_{\pm 0.0539}$ | $-1.2151_{\pm 0.0852}$ |
| SGC | $-2.6497_{\pm 0.0333}$ | $-1.5822_{\pm 0.0001}$ | $-2.3798_{\pm 0.0126}$ |
| MP-SSM | $\mathbf{-3.2353}_{\pm 0.1735}$ | $\mathbf{-4.6321}_{\pm 0.0779}$ | $\mathbf{-2.9724}_{\pm 0.0271}$ |

