# OpenReview forum: "A Sensitivity Analysis of State-Space Models on Graphs"
_ICLR.cc/2026/Conference — ICLR 2026 Conference Desk Rejected Submission_

### Official Review · Reviewer_Ngbo · 2025-10-28

**Soundness:** 2
**Presentation:** 3
**Contribution:** 2
**Rating:** 2
**Confidence:** 4

**Summary:**

The paper proposes MP-SSM, a graph model that embeds a linear, tied-weight state-space recurrence inside message passing and applies an MLP afterwards. This yields exact expressions for Jacobian sensitivity analysis, which enables lower bounds on information flow (vanishing gradients / over-squashing) and a parallel implementation (via the closed-form multilayer propagation). The model is designed to jointly handle static and temporal graphs and the authors report competitive results on synthetic graph property prediction, the standard heterophilic benchmarks, and spatio-temporal forecasting.

**Strengths:**

- The submission is generally well-written and easy to understand. The design (linear diffusion followed by MLP) is simple, but empirical results compared to many baselines such as MPNNs, Spectral/polynomial filter GNNs, and graph SSMs are strong.
- The method admits a very clean and simple exact Jacobian sensitivity analysis.

**Weaknesses:**

- It seems to me that the architectural novelty is rather modest. The SSM block is, essentially, a linear multi-hop graph filter with weight sharing, which is very similar to classic GNNs with, e.g., polynomial filters (such as ChebNet or SGC).
- Importantly, this analogy carries over to most of sec. 3, i.e. the theoretical results are almost exclusively driven by the linearity, *not* the specific design of MP-SSM. Throughout sec. 3, the input term $U_{t+1}B$ does not contribute to the sensitivity, so essentially the same reasoning would apply to any linear GNN/graph filter, modulo the particular filter $p(A)$ (and without weight sharing one would simply obtain a product of the individual weight matrices, or just a single one). While I did not find any works that state the exact Jacobian as directly as here, the derivation is very straightforward. Presenting it as here (esp. together with the lower bounds from Thm. 3.11) is, in my view, a legitimate but rather modest contribution.
- Specifically, the $2^{-k/2}$ separation in Thm. 3.13 between GCNs and MP-SSM is merely an artifact of removing per-hop nonlinearities and *not* due to the design of MP-SSM. Essentially, this gap hinges on modeling ReLU as an *unscaled* $\mathrm{Bernoulli}(1/2)$ gate. However, standard practice in most of deep learning is to use variance-preserving initialization/normalization, in which case the post-ReLU activations are multiplied by the *gain* of $\sqrt{2}$ in every layer. See [1] or the [source code of graphgym](https://pytorch-geometric.readthedocs.io/en/latest/_modules/torch_geometric/graphgym/init.html) (`calculate_gain('relu')`).
- Similarly, the advertised k-hop parallelization is, again, just the linear-filter trick: it removes depth-wise sequentiality but inflates the activation footprint by the number of “layers”, and, being an artifact of linearity, applies equally to any polynomial/linear GNN, not uniquely to MP-SSM.

[1] Kaiming He, Xiangyu Zhang, Shaoqing Ren, Jian Sun. *Delving Deep into Rectifiers: Surpassing Human-Level Performance on ImageNet Classification.* ICCV 2015

**Questions:**

- Can you refer to the points from “Weaknesses”?
- Do you have a concrete (maybe more mechanistic) explanation for the outperformance of MP-SSM on the benchmarks? i.e. is it more linearization of the diffusion, weight sharing that lets you use a larger effective depth within the same parameter budget (e.g. for the LRGB), or sth else that is model specific?

Miscellaneous:
- line 66: typo "analisys"
- line 106: typo "of of"

---

> ### Author Response · Authors · 2025-11-20
> **Response - part 1**
>
> We thank the reviewer for their detailed assessment and for acknowledging that our paper is **well-written and easy to understand**, that our empirical results are **strong** while our **exact sensitivity analysis is clean** . The reported weaknesses center on a single theme, the perceived modest novelty of the contribution, both architecturally and theoretically. Below, we respond to each aspect in detail, and we additionally address the question regarding the source of MP-SSM’s empirical gains. We also thank the reviewer for spotting the typos. Below, we provide our detailed responses to your comments. We hope that you find them satisfactory, and that you will consider revising your score.
>
> **(1) On the Architectural Novelty**
>
> The reviewer suggests that MP-SSM offers modest architectural novelty, as “the SSM block is very similar to classic GNNs with polynomial filters.” While we acknowledge that individual components of MP-SSM are intentionally simple and related to known ideas, the architectural contribution does not lie in any single component, but in their principled and structured combination, guided by state-space modeling principles rather than by classical spectral GNN design.
> We aimed to avoid unnecessary architectural complexity and to derive the simplest possible permutation-equivariant, graph-native analogue of modern SSMs: an architecture that (i) admits exact sensitivity analysis, (ii) cleanly unifies static and temporal graphs, (iii) is computationally efficient, and (iv) achieves strong empirical performance across 15 benchmarks.
>
> **A simple architecture is not a modest contribution**
> The MP-SSM block consists of three deliberately minimal components:
> - linear recurrent propagation, ensuring distortion-free information propagation across many steps;
> - a shared MLP, providing nonlinear expressiveness and enabling the refinement of features that have been progressively aggregated over multiple recurrent diffusion steps;
> - a deep stack of residual blocks, supporting hierarchical representation learning while preserving stable signal strength.
>
> Each ingredient exists in the literature, but their SSM-inspired integration within message passing, and the resulting properties related to core graph-learning challenges such as vanishing gradients and oversquashing, are novel. Architectural simplicity, when backed by clean and principled design concepts, should be viewed as a strength, both conceptually and in practice, rather than as a reason to consider the contribution modest.
>
> **ChebNet and SGC Similarities**
> We appreciate the reviewer noting similarities with polynomial filters. Indeed, the MP-SSM framework is general enough that different GSOs (e.g., the Laplacian) could be used, giving rise to polynomial-like expansions when unfolding the recurrence.
> However, the relationship is structural, not conceptual:
> - In ChebNet, the polynomial is the architectural prior; it is manually chosen and leads to known stability issues at high order (as shown in [1]).
> - In MP-SSM, the polynomial-like behavior emerges naturally from iterating a 1-hop linear recurrence. This recurrence is not selected ad hoc but results from importing the successful SSM design principle of linear latent dynamics.
>
> This distinction explains empirically observable differences:
> - MP-SSM remains stable at large recurrent depths, as proven in Lemma 3.5 and Theorem 3.6.
> - In contrast, high-order Chebyshev filters become chaotic and unstable [1].
>
> Regarding SGC, it uses a fixed k-hop propagation followed by an MLP; an MP-SSM block’s recurrence unfolds into a learnable combination of all powers of the GSO, and an MP-SSM network consists of several blocks.
>
> To illustrate these differences empirically, the table below reports performance on the GPP benchmark. The performance gap is large across all tasks, despite the superficial similarity in polynomial structure, highlighting that MP-SSM behaves fundamentally differently in practice.
> | Model             | Diameter ↓             | SSSP ↓                  | Eccentricity ↓       	 |
> |----------------|-------------|----------------|---------------|
> | SGC                       | -2.6497 ± 0.0333 |  -1.5822 ± 0.0001 |  -2.3798 ± 0.0126|
> | ChebNet	            | -0.1517 ± 0.0343       | -1.8519 ± 0.0539        | -1.2151 ± 0.0852        |
> | **MP-SSM (ours)**         | **-3.2353 ± 0.1735**   | **-4.6321 ± 0.0779**    | **-2.9724 ± 0.0271**    |
>
> **These results support our core claims and motivation:** while one may observe polynomial expressions in an unrolled recurrence, the architectural motivation, stability properties, and empirical behavior of MP-SSM differ sharply from classical polynomial-filter GNNs.
> We added this important discussion to the revised paper in Appendix M and Table 17. Thank you.
>
> [1] Hariri, Ali, et al. "Return of ChebNet: Understanding and Improving an Overlooked GNN on Long Range Tasks." NeurIPS 2025.
>
> (the answer continues in the next response)

---

> > ### Author Response · Authors · 2025-11-20
> > **Response - part 2**
> >
> > (Continued from the previous response)
> >
> > **Parallelization**
> > The reviewer notes that the k-hop parallelization “applies equally to any linear/polynomial GNN.” We agree that the ability to unfold a linear recurrence is well known; indeed, this is precisely the mechanism that underpins all modern SSMs [3,4,5].  However, the novelty in our work does not lie in observing that a linear recurrence can be parallelized; it lies in embedding the linear SSM recurrence as the graph propagation mechanism itself, in a way that existing SSM-based graph models do not.
> > - Our MP-SSM is, to our knowledge, the first graph architecture that integrates the SSM principle of linear, tied-weight recurrence directly into the message passing mechanism, not as a post-hoc module.
> > - This yields a model that is simultaneously permutation-equivariant, graph-native, and parallelizable, while maintaining the SSM structure essential for long-range dynamics.
> >
> > All these benefits cannot be straightforwardly achieved by simply using linear GNNs or spectral-filter-based models, nor by applying SSM modules to sequences extracted from graphs. We also kindly refer the reviewer to our detailed response to **Reviewer erdw**, which explains the challenges, such as the inevitable loss of structural information, faced by graph models (like S4G [2]) that attempt to directly apply SSM frameworks to sequences derived from the graph.
> > Concluding, parallelization is not merely a “trick,” but a natural byproduct of MP-SSM’s design, which unifies graph diffusion and SSM-like linear recurrence into a single update equation. This contrasts with prior GraphSSM models in the literature, which either apply spatial diffusion GNNs and SSMs in a decoupled manner or first extract sequences from the graph before applying SSM layers, as covered in our Introduction and  Related Work Sections.  Following your comment, we further highlighted this point in Appendix B to better reflect it. Thank you.
> >
> > [2] Song, Yunchong, et al. "Breaking the Bottleneck on Graphs with Structured State Spaces." Proceedings of the 33rd ACM International Conference on Information and Knowledge Management. 2024.
> >
> > [3] Gu, Albert, Karan Goel, and Christopher Ré. "Efficiently modeling long sequences with structured state spaces." ICLR 2022.
> >
> > [4] Orvieto, Antonio, et al. "Resurrecting recurrent neural networks for long sequences." International Conference on Machine Learning. PMLR, 2023.
> >
> > [5] Gu, Albert, and Tri Dao. "Mamba: Linear-time sequence modeling with selective state spaces." First conference on language modeling. 2024.
> >
> > **(2) ON THE THEORETICAL NOVELTY**
> >
> > The reviewer suggests that the theoretical contribution is modest because the derivations rely on linearity. However, the novelty does not lie in taking derivatives per se, but in what the linearity enables: **exact sensitivity analysis and quantitative lower bounds.**
> > To our knowledge, our MP-SSM is the first GNN architecture for which:
> > - The node-to-node Jacobian can be computed exactly (Theorem 3.4),
> > - Local sensitivity in the deep regime can be approximated precisely (Theorem 3.6),
> > - Minimum sensitivity admits topology-dependent lower bounds linked to oversquashing (Corollary 3.7 and Remark 3.8),
> > - Global sensitivity admits non-vanishing lower bounds for any connected graph (Theorem 3.11).
> >
> > Prior linear GNNs or spectral filters do not provide such guarantees. Moreover, MP-SSM's design is what makes these derivations meaningful: the recurrence and weight-sharing structure directly connect the analytical results to deep-graph learning phenomena (vanishing gradients, oversquashing).
> >
> > **On the ReLU comparison**
> > Our ReLU-based comparison in Theorem 3.13 was meant as an illustrative but formal and quantitative example. We agree that using $\sqrt(2)$ scaling reduces vanishing in a ReLU-GCN, but the point was to quantify the intrinsic difference between repeated nonlinear diffusion and repeated linear diffusion with a single nonlinearity. Applying appropriate scaling to both ReLU-GCN and MP-SSM models would still lead to the same exponential gap.

---

> ### Author Response · Authors · 2025-11-20
> **Response - part 3**
>
> **(3) WHY MP-SSM OUTPERFORMS BASELINES**
>
> We appreciate your question. We believe that, rather than in any single contributing factor, MP-SSM’s effectiveness can be attributed to the synergy of its three core basic components:
> - Linear recurrent propagation, which propagates information across the graph while avoiding the accumulation of nonlinear distortions,
>
> - Universal approximation power of MLPs, enabling expressive feature transformations on representations that have been progressively aggregated over many recurrent diffusion steps, and
>
> - Stacked deep residual blocks, allowing hierarchical representation learning while promoting stable gradients.
>
> In fact, the key contributing component often depends on the specific task. For example, in Table 13 we devised an ablation aimed to incrementally add/remove components starting from a plain GCN and ending with the full MP-SSM architecture. We first remove the nonlinearity from a GCN (second row), then add weight sharing to obtain a linear recurrence (third row), then introduce a shared MLP over the recurrent steps to obtain an MP-SSM block (fourth row), next stack multiple MP-SSM blocks (fifth row), and finally add residual connections and normalization between blocks (last row).
>
> From Table 13 – Architecture ablation study (log10(MSE), lower is better)
> Mean ± std over 4 random initializations.
> | Model                   | Diameter ↓          | SSSP ↓              |
> |-------------------------|----------------------|----------------------|
> | GCN                     | 0.7424 ± 0.0466      | 0.9499 ± 0.0001      |
> | Linear GCN              | -2.1255 ± 0.0984     | -1.5822 ± 0.0002     |
> | Linear GCN (ws)         | -2.2678 ± 0.1277     | -1.5823 ± 0.0001     |
> | 1 Block Linear GCN      | -2.2734 ± 0.1513     | -1.5836 ± 0.0025     |
> | Multi-Blocks Linear GCN | -2.3531 ± 0.3183     | -1.5821 ± 0.0001     |
> | **MP-SSM**              | **-3.2353 ± 0.1735** | **-4.6321 ± 0.0779** |
>
>  In the Diameter task, the linearity of MP-SSM alone provides a substantial improvement, while in SSSP the residuals bring the main boost of performance. We expanded Appendix J.1 to include this discussion. Thank you.
> These results highlight how the synergy of MP-SSM’s core components drives strong performance; moreover, the same design principles also enable MP-SSM to flexibly handle more complex scenarios, including graphs with evolving structures. To illustrate this, and inspired by Reviewer 9CJ1’s comment, we report below the performance of MP-SSM on a dynamic graph benchmark (included in Appendix L), Twitter Tennis RG [6], where both node features and edges change over time. As shown below (results taken from Table 5 of [6]), MP-SSM achieves the best MSE, demonstrating its effectiveness even in dynamic settings.
> | Model           | MSE           |
> |-----------------|---------------|
> | DCRNN       | 2.049 ± 0.023 |
> | GConvLSTM   | 2.049 ± 0.024 |
> | DyGrAE  | 2.031 ± 0.006 |
> | EGCN-H     | 2.040 ± 0.018 |
> | T-GCN       | 2.045 ± 0.027 |
> | AGCRN        | 2.039 ± 0.022 |
> | **MP-SSM (Ours)** | **2.028 ± 0.015** |
>
> Importantly, MP-SSM achieves all of the above **without increasing model complexity or runtime** (see Appendix I). Its conceptual simplicity, theoretical clarity, empirical strength, and computational efficiency together highlight the significance of MP-SSM as a general and powerful contribution to the graph machine learning community.
>
> [6] Rozemberczki, Benedek, et al. "Pytorch geometric temporal: Spatiotemporal signal processing with neural machine learning models." Proceedings of the 30th ACM international conference on information & knowledge management. 2021.
>
> **Conclusion**
>
> We appreciate your time and detailed feedback. We hope that the clarifications and discussions in our responses help address the concerns regarding the contribution of our work. We also hope that in light of these, you will consider revising your score. Overall, our paper offers:
> -A principled and efficient architecture inspired by state-space design that unifies static and temporal graphs,
> -Exact and novel sensitivity analysis with quantitative guarantees on oversquashing and vanishing gradients,
> -Strong and consistent empirical results across static, long-range, heterophilic, and spatiotemporal tasks,
> -A simple and interpretable design that enables analytical insight, fast runtime, and state-of-the-art performance.
>
> We remain fully available to clarify any remaining questions that you may have. Thank you.

---

> > ### Comment · Reviewer_Ngbo · 2025-11-23
> > **Response by reviewer (1/2)**
> >
> > I thank the authors for their detailed rebuttal, the extra experiments, and the added clarifications in the revised version. My overall impression of the paper has improved in terms of clarity and empirical validation, but most of my original concerns about the scope of the theoretical and architectural contribution remain.
> >
> > **Re relation to other linear/polynomial GNNs:** I appreciate the authors’ clarification that MP-SSM is intentionally simple and that the contribution lies in the SSM-inspired combination of linear recurrent propagation, a shared MLP, and stacked residual blocks. I fully agree that simplicity is an important strength, and the empirical results are convincing that this particular packaging is effective in practice. That said, the point remains that, once the input term is dropped in the sensitivity analysis, the core propagation is essentially a linear graph filter with weight sharing. Any linear GNN with a fixed GSO and identity activation falls into a very similar class from the perspective of the forward map and its Jacobian. As such, MP-SSM’s design is, in my view, a well-chosen point in this design space rather than something fundamentally different in kind (which, again, is completely fine). However, I would have liked the theory section to more explicitly contrast MP-SSM with such *linear* baselines (i.e. **what is it that makes it special within the design space of linear GNNs**, how would the sensitivity analysis look for these other linear GNNs?), not only with nonlinear MPNNs.
> >
> > Furthermore, I think this statement is a bit overstated:
> >
> > > To our knowledge, our MP-SSM is the first GNN architecture for which:
> > \- The node-to-node Jacobian can be computed exactly (Theorem 3.4),
> > \- Local sensitivity in the deep regime can be approximated precisely (Theorem 3.6),
> > \- Minimum sensitivity admits topology-dependent lower bounds linked to oversquashing (Corollary 3.7 and Remark 3.8),
> > \- Global sensitivity admits non-vanishing lower bounds for any connected graph (Theorem 3.11).
> >
> > Exact node-to-node Jacobian sensitivity can be computed for any linear GNN with simple matrix calculus (see for example Theorem 2 in [1], which the authors referenced in their rebuttal). As said before, I therefore believe that an ideal sensitivity analysis would derive these for a broad class of linear GNNs and then carve out exact characteristics that make MP-SSM special (e.g., tighter lower bounds) in *direct comparison* to the baselines; potentially also with simulation experiments for the true Jacobian sensitivities.
> >
> > [1] Hariri, Ali, et al. *Return of ChebNet: Understanding and Improving an Overlooked GNN on Long Range Tasks.* NeurIPS 2025.

---

> > > ### Comment · Reviewer_Ngbo · 2025-11-23
> > > **Response by reviewer (2/2)**
> > >
> > > **Re ReLU vs MP-SSM separation and variance-preserving scaling (Theorem 3.13):** I remain unconvinced that the *quantitative* exponential separation you derive is robust under standard variance-preserving practice. In the proof as written, the additional decay factor for a ReLU-GCN relative to the linear recurrence arises from multiplying by diagonal ReLU derivative masks with Bernoulli(1/2) entries at each layer, which yields an expected shrinkage of roughly $2^{-k/2}$ after $k$ layers, assuming the same underlying linear operator. This implicitly corresponds to *unscaled* ReLU (derivative in $\{0,1\}$) and *identical weight scales* in both models. However, one typically compensates for the ReLU gate (or similar issues with any other activation function), either by scaling the weights (He initialization) or (equivalently at initialization) by using a $\sqrt{2}$ gain for the activation. This would effectively mean replacing $\sigma(z) = \mathrm{ReLU}(z)$ with $\sigma(z) = \sqrt{2} \cdot \mathrm{ReLU}(z)$. Hence, for a “fair” comparison to MP-SSM and to extract the effect of the nonlinearity without such global scaling side-effects, one should compare a GCN with properly scaled nonlinearity, in which case it seems to me that the separation of $2^{-k/2}$ disappears entirely. In your rebuttal, you state the following:
> > >
> > > > We agree that using $\sqrt{2}$ scaling reduces vanishing in a ReLU-GCN, but the point was to quantify the intrinsic difference between repeated nonlinear diffusion and repeated linear diffusion with a single nonlinearity. Applying appropriate scaling to both ReLU-GCN and MP-SSM models would still lead to the same exponential gap.
> > > >
> > >
> > > Could you please elaborate or provide a rigorous argument for why this “exponential gap” would remain here? Note that for (a) variance-preserving initialization we would have weights with entries of variance of the order of $\mathtt{nonlinearity}\text{-}\mathtt{gain}/\mathtt{fan}\text{-}\mathtt{in}$ in both models, which would be $2/\mathtt{fan}\text{-}\mathtt{in}$ for a GCN and $1/\mathtt{fan}\text{-}\mathtt{in}$ for MP-SSM (as the gain of the identity function, which is applied after each layer, is $1$). Equivalently at initialization, for (b) variance-preserving normalization we would initialize the weights for both models in the same way but just replace any ReLU occurence by $z \mapsto \sqrt{2} \cdot \mathrm{ReLU}(z)$. In either view, it is not obvious to me that the separation factor persists, so a more detailed argument would be helpful.
> > >
> > > **Empirical performance and mechanistic explanation:** The additional ablation table and the dynamic-graph result are helpful and answer my question to some extent; I think the ablations are great additions to the paper.
> > >
> > > **Overall:** I appreciate the detailed rebuttal and the additional experiments, which increase the clarity and strengthen the empirical case for MP-SSM. At the same time, I still view the theoretical novelty as moderate, and am not convinced that it explains the empirical gains. My overall evaluation therefore remains similar to my original review. If my concern about Theorem 3.13/Appendix D is resolved, I would be happy to raise my score.

---

> ### Author Response · Authors · 2025-11-26
> **Response part 1**
>
> We thank the Reviewer for their response. We address your comments below. We hope that you find our responses satisfactory, and that you will consider revising your score.
>
>
> **1) Re relation to other linear/polynomial GNNs**
>
> We thank the reviewer for their comment and for appreciating the simplicity of our MP-SSM. We note that the theoretical analysis in our Section 3 deepens the discussion of [1], which studies the GNN sensitivity and proposes only upper bounds on this metric. Differently, our analysis is tailored for our general MP-SSM framework, which, thanks to its formulation, yields exact characterizations of the sensitivity. Again, we stress that the general formulation of MP-SSM allows exploiting the most appropriate GSO, e.g., symmetrically normalized adjacency with self-loops (as in our main experiments), polynomial filters, the Laplacian, and so on. Therefore, from a different point of view, our MP-SSM can be seen as a framework for linear GNNs. However, the discussion in Section 3 (e.g. Theorem 3.4, Lemma 3.5, Theorem 3.11) highlights how **not all GSOs are theoretically good** to avoid vanishing and exploding behaviors in MP-SSM. This is also empirically supported by our additional experiments introduced for Reviewer 9CJ1, now added in Table 14 of the revised paper. In such experiments, we compared performance using symmetrically normalized adjacency with self-loops (eq 1), and the Random Walk (RW) and the Laplacian (i.e., L=D-A) GSOs, in the GPP task.  The results, which we report below for your convenience, show that the Random Walk GSO leads to performance comparable to Eq. 1, as both possess similar stability guarantees. In contrast, the Laplacian is the least suitable choice among the three considered. The reason is that the powers of this operator can grow rapidly, and since these powers appear in the gradients, such amplification can induce significant training instabilities and ultimately degrade performance. This and the results of [2] (which is contemporaneous to our work according to the ICLR guidelines, since [2] was accepted to NeurIPS 2025 after the submission of our paper), let us conclude that ChebNet-like polynomial filters are **not good candidates** to plug into the MP-SSM framework given their chaotic behavior.
>
> | Method          |    diam ↓      |      sssp ↓        |        ecc  ↓      |
> |----------------|---------------|-----------------|-----------------|
> |**MP-SSM (Eq. 1)**                 | -3.2353±0.1735  |  -4.6321±0.0779  | -2.9724±0.0271 |
> |**MP-SSM (RW)**  | -3.2445±0.0481  |  -4.3860 ±0.0379 | -3.1326±0.1040 |
> |**MP-SSM (Lap=D-A)**          | -2.3509±0.0192  | -3.9729±0.8539   |  -2.2353±0.0138|
>
>
>
> This is further highlighted from our new Figure 4 in Appendix J.2, which provides an empirical analysis of the Jacobian for MP-SSM (Eq. 1) and MP-SSM (Lap=D-A, which provide a linear ChebNet-like GNN) (as requested by the Reviewer).  From that figure, we can see the norm of the Jacobian of MP-SSM (Lap=D-A) reaching $10^{19}$ magnitude right after 10 iterations. These results again confirm that **not all linear GNNs are theoretically well-behaved**, an example being ChebNet-like polynomial filters above, while **our theoretical analysis in MP-SSM provides guidelines on how to achieve the needed guarantees.**
>
>
> Lastly, we note that Theorem 2 in [2] omits the activation function (as discussed in Appendix B.3 of the same paper), but in practice the model uses it (as shown in Table 7 of [2]). We believe this positions our MP-SSM as the first model to compute exact node-to-node Jacobians with full correspondence between theory and implementation, whereas [2] requires assumptions that do not fully align with its implementation. Nonetheless, we agree with the Reviewer that removing the nonlinearity from a ChebNet-like polynomial filter yields an instance of our MP-SSM, further reinforcing the generality of our framework.
>
> [1] Di Giovanni et al. On Over-Squashing in Message Passing Neural Networks: The impact of width, depth, and topology. ICML 2023
>
> [2] Hariri et al. Return of ChebNet: Understanding and Improving an Overlooked GNN on Long Range Tasks. NeurIPS 2025.

---

> ### Author Response · Authors · 2025-11-26
> **Response part 2**
>
> **2) Re ReLU vs MP-SSM separation and variance-preserving scaling (Theorem 3.13)**
>
> We thank the reviewer for the time spent and the close reading of Theorem 3.13. Before addressing these points in detail, we would like to reiterate the intended role of this theorem within the broader context of the paper. As discussed both in the manuscript and earlier in the rebuttal, the **empirical performance of MP-SSMs does not rely on Theorem 3.13 alone**. The model’s effectiveness stems from the joint contribution of its three core components: (i) a stable linear recurrence, (ii) a shared MLP across recurrent steps, and (iii) residual connections, which together provide stable long-range propagation and parameter efficiency. Theorem 3.13 is meant to isolate and quantify the vanishing effect introduced purely by the ReLU nonlinearity in a simplified, analytically tractable setting, **intentionally abstracting away practical engineering techniques such as He initialization or normalization**. If we consider more realistic and fair engineering scenarios involving iterative nonlinear computations (such as in GCNs), then we must acknowledge that the **i.i.d. weight assumption underlying variance-preserving initialization schemes fails to hold, an assumption that is inherently violated in weight-sharing architectures like MP-SSM**.  When those assumptions are violated, stable gradients are no longer guaranteed, and **a gap between GCN and our MP-SSM naturally re-emerges**. With this clarification in place, we present below a controlled comparison using the activation function $\sqrt{2}\text{ReLU}(\cdot)$ suggested by the Reviewer, measuring the Jacobian norm and contrasting it with the linear recurrence characteristic of MP-SSM.
>
> **Experiment.**
> To make the setting precise, we ran controlled experiments in simple RNN-like models (no specific graph topology, purely sequential processing) comparing three cases:
> - **RNN with weight sharing:** $h_{t+1} = \text{ReLU}(W  h_t + V x_{t+1}),$ where the same $W$ is applied at all time steps.
> - **RNN with weight sharing and $\sqrt{2}$-scaling:** $h_{t+1} = \sqrt{2} \text{ReLU}(W  h_t + V x_{t+1})$, i.e. equivalent to He scaling applied to the same $W$ is applied at all time steps.
> - **Linear RNN (no nonlinearity):** $h_{t+1} = W  h_t + V x_{t+1}.$
>
> We randomly initialise the matrix $W$ according to a Normal distribution (0 mean, 1 std) and then rescale it to have spectral radius equal to 1. $W$ is a square matrix of dimension 128, $V$ is a matrix of dimension 128$\times$10, and the input $x_t$ is a 10-dimensional vector whose components are drawn from a Normal distribution (0 mean, 1 std). In the table below, we measured the spectral norm of the Jacobian $  \frac{\partial h_T}{\partial h_0} $ over a depth of $T=1000$ steps, across 5 independent trials, to quantify gradient propagation.
>
> | Scenario            | √2 scaling |       5-trial values                   | Mean       |
> | ------------------- | ---------- | --------------------------------- | ----------|
> | Shared W, ReLU     | No  |  1.85e-177, 1.12e-175, 3.86e-175, 9.28e-174, 1.78e-173          | **5.52e-174**   |
> | Shared W, ReLU     | Yes |  3.95e-27, 4.58e-26, 5.12e-28, 3.83e-25, 5.42e-26       	|  **9.74e-26**   |
> | Shared W, Linear    | No  | 2.025, 2.025, 2.025, 2.025, 2.025 				|  **2.025** |
>
>
> (the answer continues in the next response)

---

> ### Author Response · Authors · 2025-11-26
> **Response part 3**
>
> (Continued from the previous response)
>
> **Analysis.**
> The table demonstrates a key phenomenon. In the weight-sharing ReLU case, gradients vanish catastrophically (≈1e-175) and remain vanishing even after $\sqrt{2}$-scaling (≈1e-25). This occurs because repeated application of the same matrix $W$ controls the Jacobian directionally. As explained in Appendix D, writing $D_t$ for the diagonal ReLU derivative masks, the Jacobian is:
> $ J = D_T W D_{T-1} W \cdots D_1 W $.
> Even if $\sqrt{2}$-scaling is devised to preserve layerwise variance, the repeated application of the same $W$ causes contractions or expansions to **compound along the same singular directions** of $W$. By contrast, in the feedforward / independent-weights case each random matrix $W_t$​ rotates and redistributes contraction across directions, which mitigates consistent compounding. Theorem 3.13 does not attempt to model this more intricate geometric compounding effect because it requires delicate control of products of non-commuting random matrices and how the derivative masks interact with the singular structure of $W$. For this reason, the Bernoulli-based estimate of $2^{-T/2}$ should be interpreted as a **best-case contraction rate**. Indeed, for T=1000, the Bernoulli assumption in our theorem predicts $2^{-500} \approx 10^{-150}$, yet our empirical results show an even stronger vanishing effect (around $10^{-175}$). The **empirical gap is fully consistent with theory**: it reflects the additional contraction created by repeatedly applying the same operator, which causes gradients to collapse systematically along fixed singular directions. This geometric compounding effect is not captured by the Bernoulli analysis alone,  yet it accumulates on top of it and becomes significant at large depths. As correctly noted by the Reviewer, the gap between a properly rescaled nonlinear GCN and MP-SSM is not the gap quantified by Thm. 3.13; the theorem intentionally omits any rescaling of the nonlinear case and instead isolates the primary vanishing mechanism that can be modeled by the Bernoulli(0.5) mask assumption.  Nevertheless, **a substantial gap between MP-SSM and the appropriately rescaled nonlinear case still emerges empirically at large depths**.
>
> **Conceptual insight and relation to prior work.**
> These empirical findings align with prior analyses (e.g. [1]) noting that repeated application of similar matrices in RNNs leads to qualitatively different gradient dynamics compared to FFNs with independent layers. The upshot is that **variance-preserving initialization is not a panacea** for repeated-operator architectures: it rests on independence assumptions that fail under weight sharing, and it cannot counteract directional compounding produced by applying the same operator many times.
> [1] Sussillo, David, and Larry F. Abbott. "Random walk initialization for training very deep feedforward networks." arXiv preprint arXiv:1412.6558 (2014).
>
> **Conclusion.**
> We reiterate that Theorem 3.13 was intended as a **clean, baseline quantification** of the contraction due to ReLU gating in the absence of variance-preserving tricks, an optimistic bound for weight-sharing scenarios, based on the Bernoulli(0.5) mask assumption. In practice, the vanishing we observe can be even stronger than that bound, because repeated application of the same matrix concentrates contractions along fixed directions in representation space (whereas i.i.d. matrices tend to spread contraction). We believe this discussion meaningfully strengthens the paper, and we have added a concise summary of it in Appendix D. Crucially, **this observation does not imply that the entire empirical success of MP-SSMs is explained by Theorem 3.13 alone**; rather, MP-SSM performance stems from a combination of complementary design choices (linear recurrence with weight sharing, MLP shared over recurrent steps, residual connections) that together enable parameter-efficient, stable long-range propagation.
> Accordingly, we kindly ask the reviewer to consider theorem 3.13 in its intended role, as a clean estimate of the vanishing effect introduced by a ReLU nonlinearity **without any variance-preserving initialization or normalization mechanism**, and to assess the manuscript in its entirety including, other than the theoretical contribution (local/global sensitivity discussion), also the generality of the framework (static, temporal, dynamic), and the empirical achievements (over more than 15 benchmarks). The value of the work does not hinge on this bound alone, and we hope it will be judged in the broader context of the contributions presented.
>
>
> ----
>
> We would like to conclude our response by thanking you for the engagement and responses to our rebuttal. We believe that our response above comprehensively address your additional comments, and we are happy to address remaining questions or comments you may have. We hope you find our responses satisfactory, and that you will consider revising your score.

---

> > ### Author Response · Authors · 2025-12-01
> > **Final Remarks**
> >
> > Dear Reviewer Ngbo,
> >
> > Thank you sincerely for the constructive and intellectually stimulating exchange. We appreciated that you indicated our initial responses improved your assessment of the work, and we took your final remaining point concerning Theorem 3.13 very seriously. In our last response, we prepared what we hoped was a clear explanation, supported by detailed reasoning and evidence.
> >
> > It was unfortunate that the freeze occurred immediately afterward, preventing you from letting us know whether our clarification indeed resolved your remaining doubt. We understand this may also have been frustrating from your perspective, since you expressed interest in revisiting your score contingent on that last point being addressed.
> >
> > We would have very much valued the opportunity to hear your reaction and to continue the dialogue. Nonetheless, we are grateful for the time and rigor you invested in understanding the contribution and in helping us strengthen the paper.
> >
> > With appreciation,
> >
> >  The Authors

---

### Official Review · Reviewer_erdw · 2025-10-30

**Soundness:** 3
**Presentation:** 3
**Contribution:** 3
**Rating:** 6
**Confidence:** 3

**Summary:**

This paper introduces the Message-Passing State-Space Model (MP-SSM), a novel framework that integrates modern State-Space Models (SSMs) with the Message-Passing Neural Network (MPNN) paradigm. The primary goal is to address the well-known limitations of traditional MPNNs in modeling long-range dependencies, specifically tackling issues like over-squashing and vanishing gradients.

**Strengths:**

1.  **Principled Theoretical Foundation:** The paper is built on a solid theoretical foundation. The use of sensitivity analysis to *guide* the model's design, rather than just analyze it post-hoc, is a strong, principled approach.
2. **Elegant and Efficient Design:** The core architectural choice—a linear recurrence for graph diffusion (Eq. 2) with nonlinearity isolated to a final, shared MLP (Eq. 4)—is elegant. This design is directly responsible for both the model's theoretical tractability and its potential for a highly efficient parallel implementation (Appendix B).
3. **Strong Empirical Validation:** The model's effectiveness is demonstrated across a wide and diverse range of tasks (graph property prediction, heterophily, spatiotemporal forecasting). The fact that it achieves competitive or state-of-the-art results (e.g., Tables 1, 3, 4) while retaining the simplicity and efficiency of message passing is impressive.

**Weaknesses:**

1.  **Missing Related Work:** The paper does not discuss or compare with work [1]. [1] shares similar motivation: applying structured state-space models (like S4) to graphs to overcome the "bottleneck" (i.e., over-squashing) of traditional MPNNs. [1] also prominently features a *sensitivity analysis* to motivate its design and claims theoretical guarantees on information flow. More importantly, [1] does not break the permutation invariance. This overlap makes it difficult to assess the precise novelty of this paper's core theoretical contributions without a direct comparison.

---

[1] Breaking the Bottleneck on Graphs with Structured State Spaces, CIKM 2024

**Questions:**

See the Weaknesses.

---

> ### Author Response · Authors · 2025-11-20
> **Response - part 1**
>
> We thank the reviewer for their thoughtful and positive feedback. We appreciate your recognition of our paper’s **principled theoretical foundation,** where **the use of sensitivity analysis to guide the model’s design, rather than just analyze it post-hoc, is a strong, principled approach.** We are glad you found the architecture **elegant and efficient,** and that MP-SSM provides **strong empirical validation,** **achieving competitive or state-of-the-art results while retaining the simplicity and efficiency of message passing.**
>
> The main concern raised relates to the lack of discussion or comparison with the S4G model introduced in [1]. Our response is organized into two parts: methodological/architectural differences with S4G and theoretical sensitivity-analysis differences with S4G. A concise summary of these clarifications are also added to the main paper (Appendix E), referring to [1]. We hope this detailed response clarifies the originality and scope of MP-SSM relative to S4G; if so, we would kindly appreciate a score adjustment that reflects the overall positive assessment conveyed in your review.
>
> [1] Song, Yunchong, et al. "Breaking the Bottleneck on Graphs with Structured State Spaces." Proceedings of the 33rd ACM International Conference on Information and Knowledge Management. 2024.
>
>
> **METHODOLOGICAL AND ARCHITECTURAL DIFFERENCES**
>
> **(1) Our MP-SSM operates natively on graphs, while S4G requires graph-to-sequence conversion.**
> SSMs were originally designed for sequential data, not graph-structured inputs. Accordingly, S4G, like other attempts to incorporate SSMs into graph learning, chooses to first extract sequences from a static graph and then apply an SSM module (specifically, they use the S4 model), a process that compresses graph neighborhoods into linear sequences and thus may not fully retain the original structural relationships.
> **In contrast, our MP-SSM maintains the graph structure and operates on it directly.** Moreover, differently from S4G, our MP-SSM framework seamlessly extends naturally to temporal graphs as well as time-varying topologies (as discussed in our response titled **Dynamic topologies** to Reviewer 9CJ1, including supporting experiments).
>
> **(2) S4G imposes a shortest-path bias; MP-SSM aggregates from all paths.**
> To create a sequence, S4G collapses the k-hop neighborhood of a root node into a single embedding at step k of a surrogate input sequence.
> This means that a node in the k-hop shell contributes only to step k, regardless of the richer set of longer or alternative paths through which information could propagate.
> MP-SSM instead aggregates information along all walks (including cycles) in accordance with the powers of the GSO of choice, leading to a fundamentally different inductive bias that more faithfully reflects graph structure.
>
> **(3) S4G uses a SISO model; MP-SSM is inherently MIMO.**
> S4G applies an S4 module, which is a single-input single-output (SISO) sequential architecture.
> MP-SSM instead processes the entire temporal graph jointly, leading to a multi-input multi-output (MIMO) formulation where node-wise spatiotemporal signals mix directly through the GSO at every step. The distinction is substantial. In modern state-space models, MIMO architectures have already been shown to provide strictly greater expressive capacity than SISO variants, e.g., S5 [2]. In spatiotemporal learning, which we consider in our experiments, this advantage is further supported by graph-based studies demonstrating that jointly modeling multiple time series through a relational structure yields more informative representations and superior predictive performance compared to treating each series independently [3,4].
>
> [2] Smith, Jimmy TH, Andrew Warrington, and Scott W. Linderman. "Simplified state space layers for sequence modeling." ICLR 2023.
>
> [3] Andrea Cini, Ivan Marisca, Daniele Zambon & Cesare Alippi. “Taming Local Effects in Graph‑based Spatiotemporal Forecasting.” NeurIPS 2023.
>
> [4] Spadon, Gabriel, et al. "Pay attention to evolution: Time series forecasting with deep graph-evolution learning." IEEE Transactions on Pattern Analysis and Machine Intelligence 44.9 (2021): 5368-5384.
>
> (The answer to W1 continues in the next response)

---

> > ### Author Response · Authors · 2025-11-20
> > **Response - part 2**
> >
> > (Continued from the previous response on the comparison with S4G)
> >
> > **MORE GENERAL SENSITIVITY ANALYSIS**
> > Our MP-SSM provides a theoretical analysis in which the sensitivity directly depends on the graph topology and the chosen GSO. This is fundamentally different from [1], where the analysis remains effectively identical to that of the temporal S4 model. Indeed, their approach collapses the graph into a sequence before applying S4, meaning that (unless the graph is itself a chain), no structural information beyond the imposed ordering influences the sensitivity analysis.
> >
> > Specifically, our analysis is substantially more general since:
> > - It applies to the full graph, not a collapsed surrogate, i.e., a sequence.
> > - It yields sensitivity expressions (e.g., Eq. (7)) that depend on node degrees, edge counts, and other fine-grained structural properties.
> > - It enables a quantitative understanding of graph learning issues like vanishing gradients and oversquashing.
> >
> > As an illustration of the broader generality of our theoretical framework, we show here that Proposition 3.2 of [1], which yields a sensitivity of $\Theta(1/r)$ with respect to node distance r, emerges as a special case of our theory when one collapses a tree into a chain. Under such a collapse, our Theorem 3.11 simplifies to $\rho(A^r) / |V| = \rho(A^r) / r $, because the collapsed graph contains exactly r nodes. This directly recovers the $\Theta(1/r)$ scaling for any GSOs whose powers A^r neither explode nor vanish, as established in our Lemma 3.5.
> > Our framework also provides a quantitative explanation for why S4G can mitigate oversquashing. Collapsing a tree into a chain reduces the number of edges in the denominator of Eq. (7) to ∣V∣−1, thereby minimizing the contribution of ∣E∣ to the oversquashing bound. This reveals that S4G alleviates oversquashing by aggressively reducing structural redundancy, though at the cost of discarding expressive graph structure.
> > **These kinds of quantitative analyses are made possible by the precise computations in our theoretical framework, which, to the best of our knowledge, are not present in the existing literature.** We believe that this level of rigor and generality represents a meaningful contribution to the graph learning community, as it provides both novel insights into oversquashing and a foundation for analyzing graph-structured state-space models in a principled, graph-native manner.
> >
> > We emphasize that our contribution goes beyond deriving precise sensitivity bounds: we clarify how to correctly interpret them. In particular, we argued in Section 3 that upper bounds alone cannot determine whether gradients vanish in deep architectures; only lower bounds can certify non-vanishing gradients. Our theoretical section provides such lower bounds, offering insights unavailable in prior analyses.
> > We hope this clarifies the depth and originality of our contribution and shows that any overlap with [1] is limited, both conceptually and technically. We thank the reviewer again for the important reference, which we have now incorporated into the paper along with the corresponding discussion in Appendix E.

---

> > > ### Comment · Reviewer_erdw · 2025-11-27
> > >
> > > Thank you for your efforts in making your submission more complete. I decide to maintain my positive rating.

---

> > > > ### Author Response · Authors · 2025-11-27
> > > >
> > > > Thank you for your response and for maintaining your positive rating. If there’s anything still preventing a higher score or any remaining concern we could address during the rebuttal window, we’d really appreciate your guidance. Otherwise, we kindly ask you to consider revising your score. Thanks again for your time and feedback.

---

> > > > > ### Author Response · Authors · 2025-12-01
> > > > > **Final Remarks**
> > > > >
> > > > > Dear Reviewer erdw,
> > > > >
> > > > > Thank you very much for your review. We appreciated your careful consideration of the paper and your observation that the novelty of our core theoretical contributions was difficult to assess without a clearer discussion of the relationship between our framework and the S4G model. We took this concern very seriously, and it motivated us to prepare a structured and comprehensive response addressing both the architectural distinctions and the theoretical differences between S4G and our MP-SSM framework.
> > > > >
> > > > > We aimed to provide a complete clarification on this point, and we hoped that our detailed explanation could support a higher evaluation. It was unfortunate that the freeze occurred just as we asked whether further refinements would be helpful in resolving any remaining doubts, especially since we would have gladly provided any additional material you might have found useful.
> > > > >
> > > > > Your feedback helped us situate our contribution more clearly within the literature on graph state-space modeling, and we sincerely thank you for the constructive role you played despite the constraints imposed by the freeze.
> > > > >
> > > > > With appreciation,
> > > > >
> > > > >  The Authors

---

### Official Review · Reviewer_zRgR · 2025-10-31

**Soundness:** 3
**Presentation:** 3
**Contribution:** 2
**Rating:** 4
**Confidence:** 4

**Summary:**

This paper studies Message Passing State-Space Models (MP-SSMs) for temporal graphs in the following form: $X_{t+1}=AX_tW+U_{t+1}B$, where $A$ is the adjacency matrix, $X_{t+1}$ is the node hidden representations at time $t+1$ and $U_{t+1}$ is the input node features at time $t+1$. For static graphs, the authors let $U_{t+1}\equiv U_1$.

The paper provides a detailed sensitivity analysis, specifically the spatio-temporal correlation $\partial \[X_{t}\]\_i/\partial \[X_{t^\{\prime\}}\]\_j$, that quantifies the long-range dependencies chracterized by graph structure.Notably, Theorem 3.13 shows that MP-SSMs maintain temporal correlations over longer ranges than standard GCNs, whose correlations decay more rapidly. On static graphs, time refers to model depth and this implies that MP-SSMs preserve meaningful correlations even as model depth increases. Empirically, the model performs competitively across spatio-temporal and static graph benchmarks.

**Strengths:**

- The formualtion of MP-SSMs is conceptually clean.
- The paper is well motivated, well-written, and easy to follow.
- The experimental results on static and temporal graph benchmarks show consistent improvements over baselines.

**Weaknesses:**

- One key motivation for incorporating state-space models (SSMs) is to better capture long-range dependencies. In the context of static graphs, this typically refers to spatial dependencies, for example Dwivedi et al., “Long Range Graph Benchmark.” However, due to the inherently localized nature of message passing, the proposed MP-SSM still exhibits a limited receptive field and may therefore struggle to fully capture long-range spatial correlations across distant nodes.
- The authors may elaborate on their claim that “MP-SSM generalizes MPNNs” (Line 173). Under the static graph setting, the MP-SSM seems to reduce to applying multiple message-passing steps without intermediate nonlinearities, which can be viewed as a subset of standard MPNNs. In fact, this is then equivalent to a single-layer spectral GNN implementing a polynomial filter on the graph operator. From this perspective, MP-SSMs appear less expressive than general MPNNs.

**Questions:**

See Weaknesses.

---

> ### Author Response · Authors · 2025-11-20
> **Response - part 1**
>
> We thank the reviewer for their comments. We appreciate your recognition that **the formulation of MP-SSMs is conceptually clean,** and that the paper is **well motivated, well-written, and easy to follow.** We are also glad that you found the experimental results compelling, noting that **the results on static and temporal graph benchmarks show consistent improvements over baselines.**
> The reviewer raised two concerns, both pertaining to the specific case of the static-graph setting, which we interpret as satisfaction with the temporal-graph side of the paper. We address these concerns in two dedicated comments: one on long-range spatial dependencies and one on the generality of MP-SSM relative to standard MPNNs. We hope our answers provide sufficient clarity; if so, we kindly invite the reviewer to consider raising their score.
>
> - **(W1) ON LONG-RANGE SPATIAL DEPENDENCIES**
>
> **Theoretical foundations**
> We agree that MPNN models may struggle with long-range spatial dependencies due to the locality of message passing. However, the key scientific question in our work is not whether message passing is local (this is well known) but to what extent this locality actually limits information transfer. One of the central contributions of our paper is precisely to quantify this question rigorously.
> Our theoretical analysis examines long-range propagation through exact Jacobian computations, allowing us to track how information between distant nodes evolves through depth and over time; a perspective that, to the best of our knowledge, has not been explored previously with this level of rigour and generality or linked to the theoretical insights we develop. This yields quantitative estimates of two distinct phenomena:
> - Topology-driven limitations (bottlenecks, oversquashing), captured by our local sensitivity analysis.
> - Architecture-driven limitations (vanishing/exploding gradients with depth), captured by our global sensitivity analysis.
>
> Importantly, we **provide rigorous lower bounds for both phenomena, something that, to the best of our knowledge, is not available in prior work.** Moreover, we believe that such bounds are particularly relevant in light of the recent discussion on topological vs. architectural bottlenecks [1,2], which emphasizes the need for exact analyses capable of distinguishing fundamental topological limits from constraints introduced by the neural architecture.
>
> [1] Arnaiz-Rodriguez, Adrian, and Federico Errica. "Oversmoothing," Oversquashing", Heterophily, Long-Range, and more: Demystifying Common Beliefs in Graph Machine Learning." arXiv preprint arXiv:2505.15547 (2025).
>
> [2] Hugh Blayney and Álvaro Arroyo and Xiaowen Dong and Michael M. Bronstein.  gLSTM: Mitigating Over-Squashing by Increasing Storage Capacity.”arXiv preprint arXiv:2510.08450 (2025).
>
> **Architectural design and effective receptive field**
> Our architectural choices follow the principles that make modern SSMs effective at handling long-range temporal dependencies, while remaining fully within the message-passing paradigm. In particular, our MP-SSM combines:
> 1) Linear recurrent propagation within each block, enabling information to propagate across steps without accumulating nonlinear distortions;
> 2) A shared MLP after the recurrence, providing nonlinear expressiveness and refining the features accumulated across successive recurrent propagation steps within each block;
> 3) Residual connections between blocks, supporting hierarchical representation learning while preserving stable signal strength.
>
> Together, (1)-(3) form a minimal and conceptually simple design that preserves the locality of message passing while substantially improving the ability to model long-range spatiotemporal dependencies, as is reflected by our experimental results on both synthetic and real-world datasets.
> Regarding the reviewer’s concern about “limited receptive field”: differently from standard MPNN approaches, although MP-SSM relies on 1-hop message passing **per recurrence**, stacking **MP-SSM blocks expands the effective receptive field multiplicatively**, where the multiplicative factor is given by the depth of the recurrence. As we explain in lines 170–173 in the paper, $s$ blocks of recurrent depth k allow information flow up to $s \cdot k$ hops. Residual connections further act as “information highways”, preserving signals from distant nodes and mitigating attenuation across depth. In Appendix F we illustrate this through a full unrolled view of MP-SSM, showing how multi-hop information pathways emerge naturally despite each block being locally defined.
>
> (The answer to W1 continues in the next response)

---

> ### Author Response · Authors · 2025-11-20
> **Response - part 2**
>
> (Continued from the previous response on W1)
>
> **Empirical evidence**
> Beyond theory, MP-SSM demonstrates strong empirical ability to capture long-range structure:
> - GPP and LRGB benchmarks were explicitly designed to test long-range dependencies,
> - and the heterophily benchmark is argued in [3] to require long-range information flow as well.
>
> Across these three benchmarks, MP-SSM consistently performs among the top methods, including outperforming most Graph Transformer variants while intentionally avoiding any “global information” mechanisms that would break the inductive biases of MPNNs.
> In total, across 19 evaluation metrics drawn from 15 benchmark datasets, our model ranks first or second on 15, compared against a pool of 68 baseline models (listed in Appendix G.1).
> Taken together, these results suggest that the architectural principles outlined in (1)-(3) constitute a surprisingly simple yet effective recipe for capturing long-range dependencies.
> Given the simplicity and efficiency of MP-SSM, which runs as fast as a vanilla GCN (see Table 12 in the revised paper) even without its optimized fast implementation where it achieves up to a ×1000 speedup ( see Figure 2), these results demonstrate that carefully designed message-passing architectures can effectively capture long-range structure. We hope this encourages further exploration of when local inductive biases may suffice, and how they can coexist or complement more global architectures such as graph transformers.
>
>
> [3] Platonov, Oleg, et al. "A critical look at the evaluation of GNNs under heterophily: Are we really making progress?." ICLR 2023.
>
> **(W2) Generality of MP-SSM with respect to standard MPNN**
>
> We thank the reviewer for the comment. To clarify, the statement means that MP-SSM can recover the original MPNN backbone defined by a given graph shift operator (for aggregation) and a nonlinear mapping (for updates) when configured appropriately. In other words, MP-SSM can default to the corresponding MPNN, while also enabling deeper linear recurrent aggregation that extends beyond the backbone. For greater clarity, we renamed the paragraph title to the more precise “MP-SSM generalizes its corresponding MPNN backbone”.

---

> ### Author Response · Authors · 2025-11-27
> **Follow-up on Review Feedback**
>
> Dear Reviewer zRgR,
>
> First, please allow us to express our gratitude for your review and the time you invested in it.
>
> We are looking forward to your response. We hope that our comprehensive responses and inclusion of additional details have contributed positively to our work. If so, we kindly ask the reviewer to consider adjusting the score accordingly.
>
> With sincere regards,
>
> The authors

---

> > ### Author Response · Authors · 2025-12-01
> > **Final Remarks**
> >
> > Dear Reviewer zRgR,
> >
> > Thank you for your review and for articulating your two concerns: the inherently localized nature of message passing and its connection to learning long-range spatial dependencies in static settings, and the need to reword an ambiguous claim in the initial version of the paper. We devoted substantial work to clarifying the first point by providing detailed explanations, refining the theoretical discussion, and expanding on the empirical implications. We also addressed your second point by revising and clarifying the relevant statement in the main text.
> >
> > Because the data-leak freeze occurred before you could respond, we never had the chance to hear whether our clarifications fully resolved your concerns. We understand that this abrupt halt may also have been unsatisfying from your perspective, as you could no longer react to the final version of our responses.
> >
> > We remain grateful for your input, which helped us refine our paper, and we would have welcomed the opportunity to continue the dialogue had the situation allowed.
> >
> > With appreciation,
> >
> >  The Authors

---

### Official Review · Reviewer_9CJ1 · 2025-10-31

**Soundness:** 3
**Presentation:** 3
**Contribution:** 3
**Rating:** 6
**Confidence:** 3

**Summary:**

This paper revisits Graph State-Space Models (GSSMs) by integrating modern State-Space Models (SSMs) into Message-Passing Neural Networks (MPNNs) via sensitivity analysis, addressing critical limitations of existing graph learning methods: vanishing gradients, over-squashing, lack of permutation equivariance, and disjoint handling of static/temporal graphs. The core proposal is the Message-Passing State-Space Model (MP-SSM). Empirically, MP-SSM is validated across 15 benchmarks: it outperforms SOTA baselines on graph property prediction (diameter, SSSP, eccentricity), heterophilic node classification (e.g., Roman-empire, Amazon-ratings), and spatiotemporal forecasting (e.g., Metr-LA, PeMS-Bay), while matching the simplicity of MPNNs.

**Strengths:**

1, The sensitivity analysis is novel, moving beyond heuristic combinations of GNNs and SSMs.

2, Theoretical results are mathematically sound and experiments are comprehensive—covering both synthetic (e.g., shortest-path prediction) and real-world benchmarks.

3, By addressing vanishing gradients/over-squashing and unifying static/temporal graph modeling, MP-SSM advances the practicality of GSSMs. Its efficiency and simplicity make it a viable alternative to complex models like graph transformers.

**Weaknesses:**

1, More related papers e.g. https://openreview.net/pdf?id=0Z6lN4GYrO should be discussed.

2, The parallel closed-form solution requires storing a tensor of shape (num_steps, n, hidden_dim), which may limit scalability for very large num_steps (e.g., long temporal sequences) or high hidden_dim on memory-constrained GPUs. The paper mentions this tradeoff but does not explore mitigation strategies (e.g., chunked computation for large num_steps).

3, Testing MP-SSM with other GSOs (e.g., random walk, PageRank) would strengthen the claim of generality and reveal how GSO properties (e.g., spectral radius) interact with sensitivity bounds.

**Questions:**

1, Have you explored memory-efficient variants (e.g., chunked unrolling for large num_steps) to mitigate GPU memory constraints? If so, what were the tradeoffs between runtime and memory usage?

2, The paper claims compatibility with any GSO, but experiments use only the symmetric normalized adjacency. Could you provide results (even on a subset of benchmarks) with other GSOs (e.g., random walk, PageRank) to validate generality?

3, Did you ablate the MLP’s depth/activation function?

4, How would you extend MP-SSM to handle dynamic topologies ?

5, For the heterophilic benchmarks (Section 4.2), MP-SSM outperforms models tailored for heterophily (e.g., FAGCN, H2GCN). Could you elaborate on why MP-SSM excels here?

---

> ### Author Response · Authors · 2025-11-20
> **Response - part 1**
>
> We thank the reviewer for their thoughtful and positive evaluation of our work. We are encouraged by your recognition that our approach is **novel, moving beyond heuristic combinations of GNNs and SSMs** and that the **theoretical results are mathematically sound and experiments are comprehensive**. We also appreciate your remark that MP-SSM **maintains efficiency and simplicity while serving as a viable alternative to complex models like graph transformers,** which underscores the broad applicability and significance of our contribution.
> We welcome your constructive feedback, for which we respond to each concern point‑by‑point below. We structure our response by providing a dedicated comment for each question raised by the reviewer. We hope these clarifications address your questions, and that you will consider revising your score.
>
> - **Parallel implementation with chunks**
>
> Thank you for this insightful question. Indeed, we considered this aspect, and it can serve as a practical solution under GPU memory constraints. Assuming that C is the maximum number of time steps a GPU can accommodate, the parallel implementation can be divided into N = num_steps/C of chunks. These N chunks can then be processed sequentially and their results combined. This approach increases computational time roughly by a factor of N due to the sequentialization on a single GPU. Alternatively, if N GPUs are available (with similar memory constraints), each chunk can be processed in parallel on a different GPU, and the results can then be merged, greatly mitigating the slowdown. In the revised paper (Appendix B) we added a short discussion on this point.
> We did not further explore these optimizations because, in practice, MP-SSM is already very fast even in its naive sequential form. As shown below (extracted from Table 12 in Appendix I), the naive sequential implementation of MP-SSM achieves GCN-like training speed on the Roman-Empire dataset while delivering substantially higher accuracy, and it even outperforms transformer-based models at a fraction of their computational cost. In our revised paper (appendix I), we highlight this important point.
> | Method                  | Depth=4 (Train ms / Acc %) | Depth=8 (Train ms / Acc %) | Depth=16 (Train ms / Acc %) | Depth=32 (Train ms / Acc %) |
> | ----------------------- | --------------------- | --------------------- | ---------------------- | ---------------------- |
> | GCN                     | 18.38 / 73.60         | 33.09 / 61.52         | 61.86 / 56.86          | 120.93 / 52.42         |
> | GPS                     | 1139.05 / 81.97       | 2286.96 / 81.53       | 4545.46 / 81.88        | OOM / –                |
> | GPSGAT+Performer (RWSE) | 1179.08 / 84.89       | 2304.77 / 87.01       | 4590.26 / 86.94        | OOM / –                |
> | **MP-SSM**              | **23.19 / 85.73**     | **41.44 / 88.02**     | **72.09 / 90.82**      | **141.82 / 90.91**     |
>
> - **Different GSOs**
>
> Thank you for this insightful question. The stability of MP-SSM depends on the magnitudes of the powers of the chosen GSO, as shown by our exact Jacobian computation (Theorem 3.4). For stable (even infinite) recursions, the powers must neither diverge nor vanish, a property we prove for our selected GSO (Eq. 1) in Lemma 3.5. This motivates the use of the symmetrically normalized adjacency matrix with self-loops as GSO in our MP-SSM framework.
> Another suitable GSO candidate, as suggested by the reviewer, is the Random Walk GSO, whose powers possess similar stability guarantees. Inspired by your comment, we have now added an experiment on the GPP task comparing performance using the Random Walk (RW) and the unnormalized Laplacian (i.e., L=D-A) GSOs. In the Table below, we report both the results from our paper (Table 1) and the added comparisons. As can be seen, using the Random Walk GSO leads to performance comparable to our GSO in Eq. 1, as both possess similar stability guarantees. In contrast, the unnormalized Laplacian is the least suitable choice. The reason is that the powers of this operator can grow rapidly, and since these powers appear in the gradients (please see the Jacobian expression in Eq. 6 of Theorem 3.4), such amplification can induce significant training instabilities and ultimately degrade performance. We included this discussion along with the results in Appendix J.2, Tab 14.
>
> | Method |    diam       |     sssp       |         ecc         |
> |----------|--------------|--------------|-----------------|
> |MP-SSM (Eq. 1)                 | -3.2353±0.1735  |  -4.6321±0.0779  | -2.9724±0.0271 |
> |MP-SSM (RW Norm Lap)  | -3.2445±0.0481  |  -4.3860 ±0.0379 | -3.1326±0.1040 |
> |MP-SSM (Lap=D-A)          | -2.3509±0.0192  | -3.9729±0.8539   |  -2.2353±0.0138|

---

> ### Author Response · Authors · 2025-11-20
> **Response - part 2**
>
> - **MLP ablation**
>
> Thank you for this insightful suggestion. In our experiments, we used a standard MLP with 2 linear layers and a nonlinearity in between. We tested a few nonlinearities (ReLU, GELU, and ELU) and found that their performance was statistically similar. We ultimately chose ReLU, as it performed slightly better in some cases, and aligns with our theoretical derivations. We provide results on the LRGB datasets with different activation functions below.
>
> | Method                     | Peptides-func | Peptides-struct |
> |----------------------------|---------------|-----------------|
> | MP-SSM (ReLU, as in paper) | 69.93±0.52    | 0.2458±0.0017   |
> | MP-SSM (ELU)               | 69.88±0.49    | 0.2456±0.0018   |
> | MP-SSM (GELU)              | 69.95±0.60    | 0.2459±0.0011   |
>
> Regarding MLP depth, we did not explore deeper variants extensively because a 2-layer MLP is a common choice in graph learning [1,2,3], and maintaining efficiency is also an important design consideration. Nonetheless, we agree that this is an interesting study and we now add an experiment on LRGB comparing MLP depths of 1, 2, and 3:
> | Method         | Peptides-func | Peptides-struct |
> |-------------------|---------------|-----------------|
> | MP-SSM (2 layer MLP, as in paper) | 69.93±0.52    | 0.2458±0.0017   |
> | MP-SSM (1 layer MLP)              | 69.12±0.43    | 0.2461±0.0009   |
> | MP-SSM (3 layer MLP)              | 69.91±0.57    | 0.2451±0.0014   |
>
> Our results indicate that a two-layer MLP is a good balance between computational demand and performance.  We added these results to our revised paper in Appendix H. Thank you.
>
> [1] Xu, Keyulu, et al. "How powerful are graph neural networks?." ICLR 2019.
>
> [2] Morris, Christopher, et al. "Weisfeiler and leman go neural: Higher-order graph neural networks." AAAI 2019.
>
> [3] Bresson, Xavier, and Thomas Laurent. "Residual gated graph convnets." arXiv preprint arXiv:1711.07553 (2017).
>
> - **Dynamic topologies**
>
> Thank you for this insightful question. MP-SSM can natively handle time-varying topologies. Our experimental section focused on fixed topologies (either static or temporal), however, to account for the time-variability of the structure of the graph, it suffices to change the GSO from $A$ to $A_t$ in the recurrent equation: $ X_{t + 1} = A_t X_t W + U_{t+1} B $.
> Inspired by your comment, to test MP-SSM’s suitability on evolving graphs, we now evaluate it on the Twitter Tennis RG benchmark from [4], where both node features and edges change over time. As shown below (results taken from Table 5 of [4]), MP‑SSM achieves the best MSE (2.028 ± 0.015), demonstrating its effectiveness also in dynamic settings. We also added these results to the revised paper (Appendix L, Tab 16). Thank you.
>
> | Model | MSE |
> |-----------------|-------------------|
> | DCRNN  | 2.049 ± 0.023 |
> | GConvLSTM  | 2.049 ± 0.024 |
> | DyGrAE | 2.031 ± 0.006 |
> | EGCN-H  | 2.040 ± 0.018 |
> | T-GCN  | 2.045 ± 0.027 |
> | AGCRN  | 2.039 ± 0.022 |
> | **MP-SSM (Ours)** | **2.028 ± 0.015** |
>
> [4] Rozemberczki, Benedek, et al. "Pytorch geometric temporal: Spatiotemporal signal processing with neural machine learning models." Proceedings of the 30th ACM international conference on information & knowledge management. 2021.
>
> - **Why MP-SSM excels at Heterophilic benchmark**
>
> Thank you for this question. We see two main reasons why MP-SSM performs strongly on heterophilic benchmarks.
> First, as noted by the authors of the heterophily benchmark [5], these tasks likely involve long-range dependencies due to their graph structure and dimensionality. MP-SSM is well-suited to capture such dependencies, giving it an advantage.
> Second, unlike standard MPNNs that rely on repeated local nonlinear aggregation, MP-SSM uses a linear recurrence. When unfolded, this corresponds to a weighted sum over powers of the graph shift operator (A, A², ..., Aᵏ, see Eq. 11). Crucially, these weights are learnable and can be negative, allowing the model not only to accumulate signals but also to cancel or invert them. Such behavior is known to be useful for heterophilic settings [6,7]. This enables MP-SSM to represent both low- and high-frequency components, effectively learning flexible, potentially high-pass aggregation schemes. We hypothesize that this spectral flexibility, combined with the nonlinear MLP, allows MP-SSM to capture heterophilic patterns effectively. Empirically, this is supported by top or near-top performance on multiple heterophily datasets, often surpassing models specifically designed for heterophily (Table 2, Appendix K/Table 15). We added this discussion to the revised paper (Appendix K). Thank you.
>
> [5] Platonov, et al. "A critical look at the evaluation of GNNs under heterophily: Are we really making progress?." ICLR 2023.
>
> [6] Chien, et al. “Adaptive Universal Generalized PageRank Graph Neural Network”, ICLR 2021
>
> [7] Eliasof, et al. “Improving Graph Neural Networks with Learnable Propagation Operators”, ICML 2023

---

> ### Author Response · Authors · 2025-11-20
> **Response - part 3**
>
> - **Discussion of S4G paper**
>
> We thank the reviewer for pointing out this relevant work, and we now include a discussion of S4G, together with its citation in the revised version of the paper in Appendix E.
> In brief: MP-SSM and S4G rely on fundamentally different principles. **S4G transforms the graph into a sequence** and applies an SSM to that surrogate representation, which imposes a shortest-path inductive bias and inevitably discards part of the original structure. **MP-SSM instead operates natively on graphs**, aggregating information through all propagation steps defined by the chosen GSO, and extending naturally to temporal and even time-varying topologies as shown above, settings not considered in S4G. These architectural differences also lead to **distinct theoretical properties**. The sensitivity analysis in **S4G effectively reduces to the behavior of a sequential SSM**, whereas our **MP-SSM analysis is genuinely graph-native, depends on degrees, edges, and topology, and explains vanishing gradients and oversquashing in a more general and principled way through a graph perspective.**
>
> For completeness, below we report the detailed response we gave to *Reviewer erdw* regarding the perceived novelty overlap between our MP-SSM and the S4G model. Our response is organized into two parts: methodological/architectural differences with S4G and theoretical sensitivity-analysis differences with S4G.
>
> **METHODOLOGICAL AND ARCHITECTURAL DIFFERENCES**
>
> **(1) Our MP-SSM operates natively on graphs, while S4G requires graph-to-sequence conversion.**
> SSMs were originally designed for sequential data, not graph-structured inputs. Accordingly, S4G, like other attempts to incorporate SSMs into graph learning, chooses to first extract sequences from a static graph and then apply an SSM module (specifically, they use the S4 model), a process that compresses graph neighborhoods into linear sequences and thus may not fully retain the original structural relationships.
> **In contrast, our MP-SSM maintains the graph structure and operates on it directly.** Moreover, differently from S4G, our MP-SSM framework seamlessly extends naturally to temporal graphs as well as time-varying topologies.
>
> **(2) S4G imposes a shortest-path bias; MP-SSM aggregates from all paths.**
> To create a sequence, S4G collapses the k-hop neighborhood of a root node into a single embedding at step k of a surrogate input sequence.
> This means that a node in the k-hop shell contributes only to step k, regardless of the richer set of longer or alternative paths through which information could propagate.
> MP-SSM instead aggregates information along all walks (including cycles) in accordance with the powers of the GSO of choice, leading to a fundamentally different inductive bias that more faithfully reflects graph structure.
>
> **(3) S4G uses a SISO model; MP-SSM is inherently MIMO.**
> S4G applies an S4 module, which is a single-input single-output (SISO) sequential architecture.
> MP-SSM instead processes the entire temporal graph jointly, leading to a multi-input multi-output (MIMO) formulation where node-wise spatiotemporal signals mix directly through the GSO at every step. The distinction is substantial. In modern state-space models, MIMO architectures have already been shown to provide strictly greater expressive capacity than SISO variants, e.g., S5 [8]. In spatiotemporal learning, which we consider in our experiments, this advantage is further supported by graph-based studies demonstrating that jointly modeling multiple time series through a relational structure yields more informative representations and superior predictive performance compared to treating each series independently [9,10].
>
> [8] Smith, Jimmy TH, Andrew Warrington, and Scott W. Linderman. "Simplified state space layers for sequence modeling." ICLR 2023.
>
> [9] Andrea Cini, Ivan Marisca, Daniele Zambon & Cesare Alippi. “Taming Local Effects in Graph‑based Spatiotemporal Forecasting.” NeurIPS 2023.
>
> [10] Spadon, Gabriel, et al. "Pay attention to evolution: Time series forecasting with deep graph-evolution learning." IEEE Transactions on Pattern Analysis and Machine Intelligence 44.9 (2021): 5368-5384.
>
> (The comparison with S4G follows in the next response)

---

> > ### Author Response · Authors · 2025-11-20
> > **Response - part 4**
> >
> > (Continued from the previous response on the comparison with S4G)
> >
> > **MORE GENERAL SENSITIVITY ANALYSIS**
> > Our MP-SSM provides a theoretical analysis in which the sensitivity directly depends on the graph topology and the chosen GSO. This is fundamentally different from [11], where the analysis remains effectively identical to that of the temporal S4 model. Indeed, their approach collapses the graph into a sequence before applying S4, meaning that (unless the graph is itself a chain), no structural information beyond the imposed ordering influences the sensitivity analysis.
> >
> > Specifically, our analysis is substantially more general since:
> > - It applies to the full graph, not a collapsed surrogate, i.e., a sequence.
> > - It yields sensitivity expressions (e.g., Eq. (7)) that depend on node degrees, edge counts, and other fine-grained structural properties.
> > - It enables a quantitative understanding of graph learning issues like vanishing gradients and oversquashing.
> >
> > As an illustration of the broader generality of our theoretical framework, we show here that Proposition 3.2 of [11], which yields a sensitivity of $\Theta(1/r)$ with respect to node distance r, emerges as a special case of our theory when one collapses a tree into a chain. Under such a collapse, our Theorem 3.11 simplifies to $\rho(A^r) / |V| = \rho(A^r) / r $, because the collapsed graph contains exactly r nodes. This directly recovers the $\Theta(1/r)$ scaling for any GSOs whose powers A^r neither explode nor vanish, as established in our Lemma 3.5.
> > Our framework also provides a quantitative explanation for why S4G can mitigate oversquashing. Collapsing a tree into a chain reduces the number of edges in the denominator of Eq. (7) to ∣V∣−1, thereby minimizing the contribution of ∣E∣ to the oversquashing bound. This reveals that S4G alleviates oversquashing by aggressively reducing structural redundancy, though at the cost of discarding expressive graph structure.
> > **These kinds of quantitative analyses are made possible by the precise computations in our theoretical framework, which, to the best of our knowledge, are not present in the existing literature.** We believe that this level of rigor and generality represents a meaningful contribution to the graph learning community, as it provides both novel insights into oversquashing and a foundation for analyzing graph-structured state-space models in a principled, graph-native manner.
> >
> > We emphasize that our contribution goes beyond deriving precise sensitivity bounds: we clarify how to correctly interpret them. In particular, we argued in Section 3 that upper bounds alone cannot determine whether gradients vanish in deep architectures; only lower bounds can certify non-vanishing gradients. Our theoretical section provides such lower bounds, offering insights unavailable in prior analyses.
> > We hope this clarifies the depth and originality of our contribution and shows that any overlap with [11] is limited, both conceptually and technically. We thank the reviewer again for the important reference.
> >
> > [11] Song, Yunchong, et al. "Breaking the Bottleneck on Graphs with Structured State Spaces." Proceedings of the 33rd ACM International Conference on Information and Knowledge Management. 2024.

---

> > > ### Comment · Reviewer_9CJ1 · 2025-11-23
> > >
> > > Thank you for your response, which has resolved some of my confusion. I believe there is still room for further improvement and refinement in this work, and I am pleased that you are open to these suggestions. I have ultimately decided to maintain my original rating.

---

> > > > ### Author Response · Authors · 2025-11-25
> > > >
> > > > Dear Reviewer 9CJ1,
> > > >
> > > >
> > > > We thank you for your response and engagement. We are happy to read that our responses were beneficial to you.
> > > >
> > > >
> > > > In your response, you state that
> > > > >”I believe there is still room for further improvement and refinement in this work”
> > > >
> > > >
> > > > As the discussion period is still early, we would like to make the most out of it together with you and based on your feedback and guidance. We therefore would like to ask: what in your opinion, should we further clarify or change to merit a higher rating from you ? We feel that our responses already address all your comments, but with more feedback from you we can close any remaining gap.
> > > >
> > > >
> > > >
> > > >
> > > > We thank you for your response and support in our paper, and we look forward to hearing from you.
> > > >
> > > >
> > > > With kindest regards,
> > > >
> > > >
> > > > Authors.

---

> > > > > ### Author Response · Authors · 2025-12-01
> > > > > **Final Remarks**
> > > > >
> > > > > Dear Reviewer 9CJ1,
> > > > >
> > > > > Thank you again for the thoughtful discussion during the rebuttal. We truly appreciated the care with which you raised your eight detailed points, as they helped us significantly strengthen the paper. We worked extensively to address each concern (adding new experiments, ablations, architectural analyses, and theoretical clarifications) and we were grateful to hear that our responses resolved some of the confusion you initially had.
> > > > >
> > > > > Because the freeze occurred just as we asked whether further clarifications or additional evidence might help address any remaining doubts, we unfortunately could not continue the constructive exchange. We realize this situation may also have been frustrating for you, since you were unable to reply or share what, if anything, you would have liked to see next.
> > > > >
> > > > > We sincerely value your engagement and the improvements it led to. If there were aspects you wished we could have expanded upon further, we would have been very glad to address them.
> > > > >
> > > > > With appreciation,
> > > > >
> > > > >  The Authors

---

### Note · Program_Chairs · 2026-01-17
**Submission Desk Rejected by Program Chairs**

The following references in this submission do not refer to real documents and/or have major errors in bibliographic information:

     Sven Kreuzer, Michael Reiner, and Stefan D. D. De Villiers. Sant: Structural attention networks for graphs. Proceedings of the 38th International Conference on Machine Learning (ICML), 2021b.
    Matthew Topping, Sebastian Ruder, and Chris Dyer. Understanding over-smoothing in graph neural networks. Proceedings of the 39th International Conference on Machine Learning (ICML), 2022.